# Ocean biogeochemistry in the coupled ocean–sea ice–biogeochemistry model FESOM2.1–REcoM3

**Özgür Gürses**[1], **Laurent Oziel**[1], **Onur Karakuş**[1], **Dmitry Sidorenko**[1], **Christoph Völker**[1], **Ying Ye**[1], **Moritz Zeising**[1], **Martin Butzin**[1,2], **and Judith Hauck**[1]

[1]Marine Biogeosciences, Alfred-Wegener-Institut, Helmholtz-Zentrum für Polar- und Meeresforschung, 27570 Bremerhaven, Germany
[2]MARUM – Center for Marine Environmental Sciences, University of Bremen, 28334 Bremen, Germany TS1

**Correspondence:** Özgür Gürses (ozgur.gurses@awi.de)

**Abstract.** TS2 CE1 The cycling of carbon in the TS3 oceans is affected by feedbacks driven by changes in climate and atmospheric $CO_2$. Understanding these feedbacks is therefore an important prerequisite for projecting future climate. Marine biogeochemistry models are a useful tool but, as with CE2 any model, are a simplification and need to be continually improved. In this study, we coupled the Finite-volumE Sea ice–Ocean Model (FESOM2.1) to the Regulated Ecosystem Model version 3 (REcoM3). FESOM2.1 is an update of the Finite-Element Sea ice–Ocean Model (FESOM1.4) and operates on unstructured meshes. Unlike standard structured-mesh ocean models, the mesh flexibility allows for a realistic representation of small-scale dynamics in key regions at an affordable computational cost. Compared to the previous coupled model version of FESOM1.4–REcoM2, the model FESOM2.1–REcoM3 utilizes a new dynamical core, based on a finite-volume discretization instead of finite elements, and retains central parts of the biogeochemistry model. As a new feature, carbonate chemistry, including water vapour correction, is computed by mocsy 2.0. Moreover, REcoM3 has an extended food web that includes macrozooplankton and fast-sinking detritus. Dissolved oxygen is also added as a new tracer. In this study, we assess the ocean and biogeochemical state simulated with FESOM2.1–REcoM3 in a global set-up at relatively low spatial resolution forced with JRA55-do (Tsujino et al., 2018) atmospheric reanalysis. The focus is on the recent period (1958–2021) to assess how well the model can be used for present-day and future climate change scenarios on decadal to centennial timescales. A bias in the global ocean–atmosphere preindustrial $CO_2$ flux present in the previous model version (FESOM1.4–REcoM2) could be significantly reduced. In addition, the computational efficiency is 2–3 times higher than that of FESOM1.4–REcoM2. Overall, it is found that FESOM2.1–REcoM3 is a skilful tool for ocean biogeochemical modelling applications.

## 1 Introduction

There is an unequivocal consensus and concern about the effects of increasing greenhouse gases in the atmosphere due to human activities. Since the beginning of the industrial era (year 1750), the concentration of carbon dioxide ($CO_2$) in the air has substantially risen from 277 to 417.2 ppm (year 2022; Friedlingstein et al., 2022b). The ocean has taken up a remarkably constant fraction of 25 %– 30 % of human $CO_2$ emissions from fossil fuel burning and land use change throughout time (Crisp et al., 2022). For the recent decade, 2012–2021, the rate of ocean anthropogenic carbon uptake (including the effects of climate change) amounted to $2.9 \pm 0.4$ PgC yr$^{-1}$ (26 % of the total $CO_2$ emissions; Friedlingstein et al., 2022b). A similar proportion was taken up by the terrestrial biosphere, amounting to $3.1 \pm 0.6$ PgC yr$^{-1}$ (2012–2021), but the total air-to-land $CO_2$ flux is substantially lower because of emissions from land use change, mainly deforestation, that amounted to $1.2 \pm 0.7$ PgC yr$^{-1}$ (2012–2021). The ocean carbon sink has grown over the past few decades in response to the near-exponential rise in $CO_2$ emissions (Friedlingstein et al.,

2022b). While the global ocean carbon sink estimate is assigned an uncertainty of $0.4\,\mathrm{PgC\,yr^{-1}}$ and medium confidence, regional patterns of the sink differ more strongly. This points to the balance between physical and biological processes, which are more difficult to model, as also illustrated in model deficiencies in accurately representing the seasonal cycle of $p\mathrm{CO_2}$ and $\mathrm{CO_2}$ fluxes (Mongwe et al., 2018). Both climate change and rising atmospheric $\mathrm{CO_2}$ will feed back on the fraction of $\mathrm{CO_2}$ emissions that will end up in the ocean over the next century (Friedlingstein et al., 2003; Canadell et al., 2021). Models are important tools for estimating how large these feedbacks are.

The flux of $\mathrm{CO_2}$ between the atmosphere and ocean is controlled by two main mechanisms, namely the solubility pump and the biological pump. The solubility pump describes the air–sea $\mathrm{CO_2}$ exchange that occurs to satisfy a thermodynamic equilibrium and the subsequent transport of carbon from the surface to the deep ocean with the overturning circulation. This leads to $\mathrm{CO_2}$ uptake at mid- to high latitudes through high solubility in cold waters and large vertical motion in deep-water formation regions. In contrast, warm ocean regions in the tropics and subtropics and upwelling regions lose carbon to the atmosphere (Takahashi et al., 2009; Wanninkhof et al., 2013). The solubility pump is responsible for anthropogenic carbon uptake. The biological carbon pump comprises the fixation of $\mathrm{CO_2}$ into biomass by phytoplankton and the subsequent downward transfer of dead organic material (Boyd et al., 2019). The biological carbon pump is responsible for 75 % of the natural vertical carbon gradient and for the large interbasin gradient between the deep Pacific and Atlantic (Sarmiento and Gruber, 2006). Without the biological carbon pump, atmospheric $\mathrm{CO_2}$ would be higher by 200 ppm (Maier-Reimer et al., 1996), and perturbations thereof can have large effects on atmospheric $\mathrm{CO_2}$ (Kwon et al., 2009; Lauderdale and Cael, 2021), as also known from paleo evidence (Galbraith and Skinner, 2020).

Global ocean biogeochemistry models (GOBMs; Fennel et al., 2022) are used to assess the global ocean carbon sink (Hauck et al., 2020), its regional patterns (Fay and McKinley, 2021), and effects of climate change and variability on the ocean carbon sink (Le Quéré et al., 2010; Hauck et al., 2013; DeVries et al., 2019; Bunsen, 2022). Through their representation of pH, the marine oxygen cycle, and phytoplankton primary production as the base of the marine food web, they also offer information about the environmental conditions for marine life and how these will develop under climate change (Bopp et al., 2013; Laufkötter et al., 2015; Kwiatkowski et al., 2020). However, modelling the marine biogeochemistry is subject to several sources of uncertainties. First, GOBMs are expensive with respect to the computational cost, CE3 due to the advection of a large number of tracers, and therefore often demand low spatial resolution. This leads to deficiencies in the representation of significant physical processes such as (sub)mesoscale currents (McWilliams, 2016), which can have large impacts on trans-

port and mixing processes that strongly affect biological productivity (Lévy et al., 2018; Keerthi et al., 2022). Second, the descriptions of ecological interactions and of the physiology of primary and secondary producers in GOBMs are still mostly based on empirical or semi-empirical mathematical descriptions, such as the dependency of zooplankton grazing rates on prey abundance (Doney et al., 2001; Rohr et al., 2022). These contain a large number of parameters that are only partly constrained from observations, making it necessary to tune these parameters in GOBMs to some extent. Choices in these parameters can have strong effects on the biological carbon pump (e.g. Lauderdale and Cael, 2021). It has been demonstrated that the largest source of uncertainty for projections of net primary production (NPP; Tagliabue et al., 2021) comes from model uncertainty and not scenario uncertainty (Frölicher et al., 2016).

Ocean circulation models formulated on unstructured meshes have become an alternative to the existing structured global ocean models (Danilov, 2013). The Finite-Element Sea ice–Ocean Model (hereafter FESOM1.4; Wang et al., 2014) is one of the first global models with multiple resolutions designed to simulate the large-scale ocean circulation. While it has already been used in numerous applications (Sidorenko et al., 2015; Wekerle et al., 2017), another dynamical core, the Finite-volumE Sea ice–Ocean Model version 2.1 (FESOM2.1), has been developed (Danilov et al., 2017). The advantages of a finite-volume formulation are (a) better throughput and scalability as a result of a more efficient data structure (Koldunov et al., 2019), (b) the availability of clearly defined fluxes, and (c) the possibility to choose from a selection of transport algorithms, which was very limited before (Danilov et al., 2017). Furthermore, the arbitrary Lagrangian–Eulerian (ALE) vertical coordinate is introduced, which provides different types of vertical coordinates (Scholz et al., 2019). The Regulated Ecosystem Model (REcoM) is an ocean biogeochemistry model that describes the lower trophic levels of the marine ecosystem, using the plankton functional type approach. It bases its description of primary production on a physiological model for phytoplankton growth that takes into account the nutrient availability effects on photoacclimation (Geider et al., 1998) and, for diatoms, on the relative frustule weight (Hohn, 2009). One specificity of REcoM is the representation of flexible stoichiometry, which leads to a description of elemental fluxes that can deviate from the fixed Redfield ratios often used in models (Redfield et al., 1963).

Here, we document the ocean biogeochemistry in the Regulated Ecosystem Model version 3 (REcoM3), coupled to the ocean and sea ice model FESOM2.1, and assess its performance in reproducing carbon and nutrient biogeochemical fluxes and the distribution of phytoplankton and zooplankton. Our aim is to analyse the new set-up regarding the coupled model state under historical atmospheric $\mathrm{CO_2}$ forcing and the associated model bias and drift from the experiment with a constant preindustrial (PI) $\mathrm{CO_2}$ level. We thus focus on

evaluating model aspects with regard to the effects of climate change and $CO_2$ increase on carbon fluxes on century-scale timescales. We exclude in our analysis the deep-sea distribution of carbon and nutrients. This would require model runs over at least 500 to 2000 years (Séférian et al., 2020), which will be done in follow-up work.

## 2 Methods

### 2.1 Model description

We present the coupled ocean–sea ice–biogeochemistry model FESOM2.1–REcoM3. The previous model version (FESOM1.4–REcoM2) has been described by Schourup-Kristensen et al. (2014). Unlike its predecessor FESOM1.4, which uses a finite-element formulation, the ocean model is now based on a finite-volume discretization, which makes tracer conservation much easier to achieve. FESOM2.1 was described by Danilov et al. (2017) and evaluated in Scholz et al. (2019, 2022). The ocean biogeochemistry is simulated by the Regulated Ecosystem Model version 3 (REcoM3), which builds upon the previous version of REcoM2 (Hauck et al., 2013; Schourup-Kristensen et al., 2014). The advection and diffusion of 28 passive biogeochemical tracers is handled by FESOM2.1, whereas REcoM3 calculates the sources and sinks driven by biological interactions or biogeochemical exchange processes.

#### 2.1.1 Ocean model FESOM2.1

FESOM2.1 solves the hydrostatic primitive equations under the Boussinesq approximation (Danilov et al., 2017). When set in a differential form, this equation is discretized on a finite set of points (nodes). As a first step of the mesh generation, a 2-dimensional grid is created by combining these nodes into triangular shapes (elements). At this stage, the mesh resolution (i.e. the size of triangles) can be adjusted in areas of interest without requiring a nesting approach. A 3-dimensional mesh is produced by projecting the triangles in a vertical direction and forming prisms. The scalar quantities (tracers and pressure) are located at the nodes, while the horizontal velocities are defined at centroids of the elements (see Figs. 1 and 2 in Danilov et al., 2017). A pair of control volumes are defined, where the vector control volumes are the prisms based on elements, and the scalar control volumes are formed by connecting cell centroids with edge midpoints (Fig. 1). Integration is carried out on a staggered Arakawa B-type mesh (Scholz et al., 2019).

We use FESOM2.1, an updated version of FESOM2.0. The updated model features include several developments, such as parallel and asynchronous output writing. An important new feature that we applied is the kinematic backscatter parameterization. This method takes into account the scales at which energy is scattered back to the resolved flow by introducing a negative viscosity term (Juricke et al.,

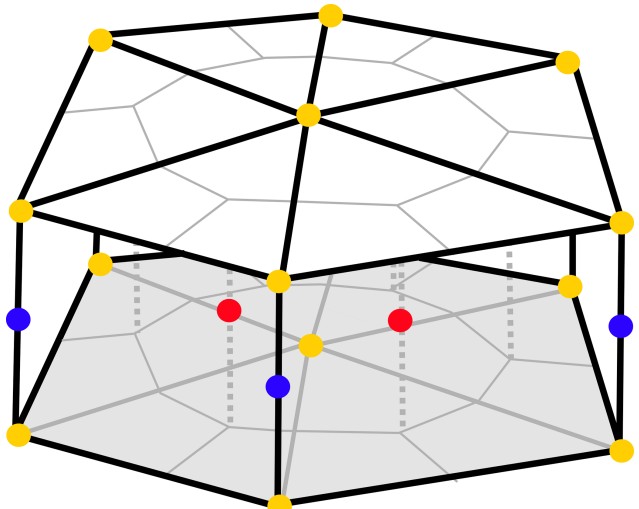

**Figure 1.** Scheme of the cell–vertex discretization in 3-dimensional space. Blue dots correspond to scalar quantities, including REcoM3 state variables, located at the mid-layer vertices of triangles. Red dots represent horizontal velocities located at the mid-layer cen-tres of the triangles. Yellow dots depict the vertical transfer veloci-ties, which are placed at the layer boundaries aligned with the scalar quantities in the vertical.

2020). This greatly improves the simulation of eddy effects in coarse-resolution mesh set-ups (Juricke et al., 2020). The model code also includes the representation of ice shelf cavities (Timmermann et al., 2012), which has been used in regional studies with FESOM1.4–REcoM2 (Nissen et al., 2022). Ice shelf cavities are, however, not used in this study. Isoneutral tracer diffusion (Redi, 1982) and the Gent-McWilliams (GM; Gent and McWilliams, 1990; Griffies, 1998) eddy stirring parameterization are applied. Both GM and the Redi (1982) scheme are scaled with horizontal resolution, with a maximum value of $2000\,\mathrm{m^2\,s^{-1}}$ at $100\,\mathrm{km}$ horizontal resolution. The scaling decreases linearly below a resolution of $40\,\mathrm{km}$ to reach CE4 0 at $30\,\mathrm{km}$ resolution, thus effectively switching the parameterization off. As a vertical mixing parameterization, the $K$-profile scheme is used (KPP; Large et al., 1994), with a background vertical diffusivity of $1 \times 10^{-4}\,\mathrm{m^2\,s^{-1}}$ for momentum and $1 \times 10^{-5}\,\mathrm{m^2\,s^{-1}}$ for tracers. Furthermore, the Monin–Obukhov length-dependent vertical mixing parameterization is applied in the surface boundary layer south of $50°\,\mathrm{S}$ (Timmermann and Beckmann, 2004).

Regarding the vertical discretization, FESOM2.1 is formulated with an arbitrary Lagrangian–Eulerian (ALE) scheme, which is a synthesis of different types of vertical coordinates. In the model configuration used here, we apply a full free-surface formulation and thus permit the vertical movement of the surface and of all other layers (referred to as zstar; Scholz et al., 2019). This drastically improves the tracer conservation properties (Campin et al., 2004). Partially filled cells are

used at the ocean floor, resulting in a smoother representation of the bathymetry.

The sea ice CE5 component (Finite-Element Sea Ice Model, FESIM version 2) solves for sea ice concentration, ice and snow thickness, and ice drift velocity (Danilov et al., 2015). It is discretized on the same unstructured horizontal mesh as the ocean model. The elastic viscous plastic solver and flux-corrected transport scheme are used for sea ice advection (Danilov et al., 2015). The formulation of sea ice thermodynamics follows the work of Timmermann et al. (2009).

### 2.1.2 Biogeochemistry model REcoM3

REcoM3 is a water column biogeochemistry and ecosystem model, which incorporates cycles of carbon and nutrients (nitrogen, iron, and silicon) with varying intracellular stoichiometry in phytoplankton, zooplankton, and detritus (see the Appendix for detailed description and equations). Starting from the work by Schartau et al. (2007), REcoM was first used to describe carbon overconsumption in mesocosm experiments. After being coupled to the ocean and sea ice model MITgcm (Marshall et al., 1997), the previous version (REcoM2) with two phytoplankton classes, one zooplankton and one detritus class, CE6 was applied to study the cycling of marine carbon on present-day (Hauck et al., 2013, 2018) and glacial timescales (Du et al., 2022; Völker and Köhler, 2013), as well as the marine iron cycle (e.g. Völker and Tagliabue, 2015; Tagliabue et al., 2016; Ye and Völker, 2017; Pagnone et al., 2019). Moreover, REcoM2 was employed in assessments on the efficiency of ocean alkalinity enhancement (Köhler et al., 2013; Hauck et al., 2016), in data assimilation studies (Pradhan et al., 2019), and as a test bed for model development; e.g. for the development of a parameterization of iron-ligand binding based on pH (Ye et al., 2020), among others. Simultaneously, REcoM2 was coupled to FESOM1.4 (Schourup-Kristensen et al., 2014). These coupled model set-ups were used either in a global configuration (e.g. Schourup-Kristensen et al., 2014; Hauck et al., 2020), with a regional focus on the Arctic or the Antarctic (Hauck et al., 2015; Schourup-Kristensen et al., 2018; Oziel et al., 2022; Nissen et al., 2022) or in regional configurations (Taylor et al., 2013; Losch et al., 2014). Recently, the model has matured to include two groups of each classes of phytoplankton, zooplankton, and detritus (REcoM3; Fig. 2).

Marine primary production is computed through representation of two phytoplankton functional types (PFTs), namely diatoms and small phytoplankton. The diverse group of small phytoplankton comprises a wide range of taxa, including, for instance, non-silicifying, calcifying, and non-calcifying haptophytes and green algae. The model allows PFTs to adapt their internal stoichiometry (C : N : Chl : CaCO$_3$ ratios for small phytoplankton and C : N : Chl : Si for diatoms) to nutrient levels, ambient light, and temperature, based on the photoacclimation model by Geider et al. (1998). Si uptake by diatoms is regulated as well, based on the internal Si : N quota, following Hohn (2009). This parameterization takes into account the strong decoupling between Si and N metabolism (e.g. Claquin et al., 2002) and prescribes the observed change in Si : N ratios under Fe and N limitation. The intracellular iron pool is derived from intracellular nitrogen with a fixed Fe : N ratio, based on the fact that intracellular iron is mostly associated with the photosynthetic electron transport chain and nitrogen metabolism (Geider and La Roche, 1994; Raven, 1988). REcoM3 also includes the photodamage parameterization by Álvarez et al. (2018). Calcium carbonate production is assumed to be linearly dependent on the gross small phytoplankton production. CaCO$_3$ dissolution is described by a depth-dependent dissolution rate.

Zooplankton is represented by two groups, namely small zooplankton and polar macrozooplankton (Karakuş et al., 2021), and each group has a carbon and nitrogen tracer. The small zooplankton group in the model is associated with relatively higher grazing rates compared to macrozooplankton and is widely spread in the global ocean. The polar macrozooplankton is mainly present in the Southern Ocean and northern high latitudes. The respiration rate is described mechanistically for macrozooplankton, taking into account the reduced metabolism in winter and increased metabolism at high grazing rates (Karakuş et al., 2021). For small zooplankton, respiration is calculated with a fixed respiration rate constant and biomass, which is in contrast to the previous version REcoM2, where respiration was used to drive zooplankton C : N back towards the Redfield ratio (Hauck et al., 2013; Schourup-Kristensen et al., 2014). Grazing is computed by applying a sigmoidal function, with variable preferences on both phytoplankton and detritus (Fasham et al., 1990).

Particulate organic matter (detritus) is split into two groups. The sinking speed of the first detritus group increases linearly with depth (from $20 \, \mathrm{m \, d^{-1}}$ at the surface to $192 \, \mathrm{m \, d^{-1}}$ at 6000 m depth; Kriest and Oschlies, 2008). The sinking speed of the second group (fast-sinking detritus) is constant throughout the water column ($200 \, \mathrm{m \, d^{-1}}$; Karakuş et al., 2021). Remineralization of carbon and nitrogen occurs in two steps. Detrital material is first degraded to dissolved organic matter and then remineralized to the inorganic forms (dissolved inorganic carbon and nitrogen). For iron, it is assumed that the organic form is directly bioavailable, so it enters the dissolved iron pool in one step.

REcoM3 comprises a single-layer sediment pool for nitrogen, silicon, dissolved inorganic carbon, and calcium carbonate. The sinking detritus and associated minerals are accumulated in this layer when they reach the ocean floor. This material is subsequently returned back to the water column to the pools of dissolved inorganic nitrogen, carbon, and silicon, as well as alkalinity, with a fixed remineralization rate. The release of iron to the bottom layer of the ocean is assumed to be proportional to the release of inorganic nitrogen (Elrod et al., 2004).

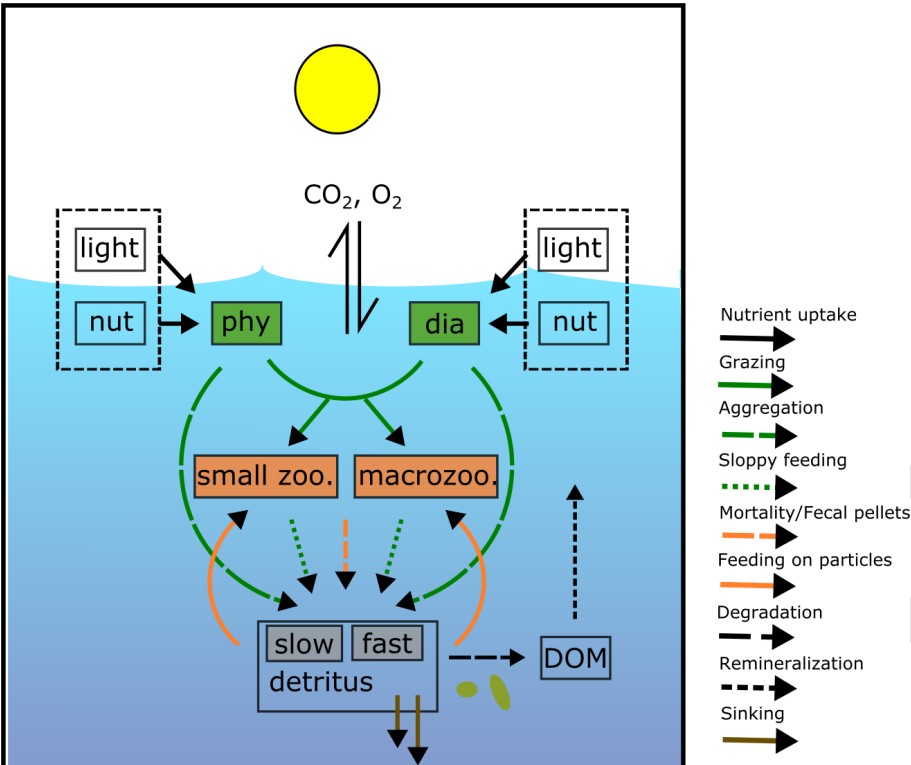

**Figure 2.** Schematic diagram of the components and interactions in the REcoM3 model. Small phytoplankton (phy) and diatoms (dia) take up inorganic nutrients (nut). Small zooplankton (small zoo.) and macrozooplankton (macrozoo.) consume phytoplankton and particles. Macrozooplankton feed on small zooplankton. Phytoplankton aggregation, zooplankton sloppy feeding, mortality, and fecal pellets generate sinking detritus (slow and fast). Sinking detritus degrades to dissolved organic carbon and nitrogen. Dissolved organic material (DOM) then remineralizes to dissolved inorganic carbon and nitrogen. The number of tracers and detailed processes is CE7 shown in the Appendix (Fig. A2).

### 2.1.3 Updates to previous REcoM version coupled to FESOM1.4

There are numerous improvements relative to the previously documented version of FESOM1.4–REcoM2 (Schourup-Kristensen et al., 2014), and the main changes are listed below.

**REcoM**

1. The routines for calculating carbonate chemistry and air–sea $CO_2$ exchange used in FESOM1.4–REcoM2, which followed the guidelines provided by the Ocean Carbon-Cycle Model Intercomparison Project (Orr, 1999), were replaced by the mocsy 2.0 scheme of Orr and Epitalon (2015). While both use the same thermodynamic equilibrium to calculate surface $p\mathrm{CO_2}$ and $CO_2$ flux, mocsy 2.0 uses the faster and more accurate algorithm SolveSAPHE (Munhoven, 2013). Among other differences, it follows best-practice guides and uses recommended equilibrium constants. The gas exchange formulation is updated to that of Wanninkhof (2014), which is largely equivalent to that of Ho et al.

(2006). The computed fluxes are scaled with the ice-free area.

2. Dissolved oxygen was added as a new tracer in REcoM3. The air–sea $O_2$ flux is calculated using the mocsy 2.0 routines (Orr and Epitalon, 2015). Photosynthesis, respiration, and remineralization change oxygen with a fixed $O_2 : C$ ratio, and remineralization does not depend on the $O_2$ levels in the current model version.

3. A second zooplankton group and a fast-sinking detritus class were added. The second zooplankton group represents a slow-growing polar macrozooplankton with a feeding preference on diatoms, which produces fast-sinking and carbon-rich fecal pellets (Karakuş et al., 2021).

4. The intracellular iron concentration is connected to intracellular nitrogen via a constant ratio $Fe : N$, leading to some variation in the $Fe : C$ ratio, as briefly presented in Tagliabue et al. (2016) and Pagnone et al. (2019).

5. A sedimentary release of iron was added to the model (Tagliabue et al., 2016); this was done in addition to the previously considered Fe input with dust deposition.

**FESOM**

Biogeochemical fluxes returned back to the ocean from the benthos are treated with a specific bottom boundary condition. Variable bottom topography leads to a smaller scalar control volume located at the lowermost level. This is because the scalar control volumes are obtained by connecting the areas from the elements to which they are attached (see Fig. 1 in Danilov et al., 2017). Therefore, the number of elements around a single surface node may vary with depths when it meets non-flat topography. We thus computed the control volume and associated fluxes for each node by considering all surrounding elements at different depth levels.

**Forcing**

Our simulation was forced by the atmospheric reanalysis JRA55-do data set (Tsujino et al., 2018) instead of the CORE-II data set (Large and Yeager, 2009) that was used in previous assessments (Schourup-Kristensen et al., 2014). JRA55-do is a blend of reanalysis data and satellite observations and has the advantage to provide regularly updated near-real-time data up to the present day, with a higher temporal (3 h) resolution. The freshwater supplied by rivers is a climatology and provided by Large and Yeager (2004) as part of the CORE-II forcing. Nutrient, carbon and alkalinity supply via river discharge is not included in the experiments described here.

**2.2 Experimental set-up and data**

In this study, we used a mesh with a nominal resolution of 1° as a background. The horizontal resolution is enhanced on the equatorial belt and in the region north of 50° N to match 1/3° and 25 km, respectively. The mesh has 48 unevenly spaced vertical layers, where the layer thickness ranges from 5 m at the surface to 250 m in the deep ocean (Scholz et al., 2019).

Initial fields for temperature and salinity were taken from the winter statistical fields of the Polar science center Hydrographic Climatology (PHC3.0; updated from Steele et al., 2001) that ingests observations from the period 1900–1994. Total alkalinity (Alk) and preindustrial dissolved inorganic carbon (DIC) were initialized from version 2 of the Global Ocean Data Analysis Project (GLODAPv2) climatology centred on the year 2002 (Lauvset et al., 2016) and based on data collected between 1972 and 2013. Dissolved inorganic nitrogen (DIN) and dissolved silicic acid (DSi) were started with values from the World Ocean Atlas climatology of 2013 (Garcia et al., 2014) that occupied the period between 1955 and 2012. We used the World Ocean Atlas climatology of

2018 for dissolved oxygen (Garcia et al., 2019a, see Table 2), based on data for the time span 1955–2017.

Due to scarcity of observations, the iron field (DFe) was initialized with output from the Pelagic Interactions Scheme for Carbon and Ecosystem Studies (PISCES) model (Aumont et al., 2003), which was corrected using observed profiles for the Southern Ocean (de Baar et al., 1999; Boye et al., 2001). Sensitivity tests indicated that high values stemming from a hydrothermal vent in the eastern equatorial Pacific led to unreasonably large values in the interior Pacific Ocean due to advective fluxes. Therefore, the region spanning the latitudes of 9.5° N–12.5° S and longitudes 72–106° W was masked to a maximum value of $0.3\,\mu\mathrm{mol\,m^{-3}}$ (below 2000 m). All other tracers were initialized with small values.

Iron was supplied to the ocean by dust deposition and from sediments. The sedimentary flux was assumed to scale with the organic nitrogen flux into the sediment, as found in Elrod et al. (2004). REcoM3 used monthly averages of dust deposition (Albani et al., 2014). We assumed that 3.5 % of the dust field consists of iron, of which 1.5 % dissolves into a bioavailable form when deposited on the ocean surface. We did not include aeolian nitrogen deposition in our simulations.

The atmospheric reanalysis data sets of JRA55-do v.1.5.0 (Tsujino et al., 2018) were used to force the model for the period 1958–2021 (hereafter JRA55-do). A single repeating annual cycle of all forcing fields (year 1961) was used to perform the spin-up simulations and a control experiment. This is referred to as repeat-year forcing (hereafter called RYF61). We have deliberately chosen the year 1961, as it had rather neutral El Niño–Southern Oscillation conditions and also contained a low amount of anthropogenic perturbation, compared to the years of 1990 and 1991 recommended by Stewart et al. (2020).

A series of experiments were carried out in a global set-up to investigate the performance of the coupled FESOM2.1–REcoM3 model. The experiments follow the definitions used in the Global Carbon Budget (Friedlingstein et al., 2022a) and in the RECCAP (REgional Carbon Cycle Assessment and Processes, https://reccap2-ocean.github.io/, last access: TS4) projects and are summarized in Table 1. Our first experiment was forced with varying climate from the JRA55-do data set and varying atmospheric $CO_2$ levels (hereafter referred to as A). Atmospheric $CO_2$ mixing ratio ($xCO_2$) values are taken from the Global Carbon Budget (Joos and Spahni, 2008; Ballantyne et al., 2012; Friedlingstein et al., 2022a). A second simulation was forced by RYF61 atmospheric reanalysis fields and a preindustrial atmospheric $CO_2$ mixing ratio of 278 ppm. This configuration, here termed B, is considered to be the control run. Our last simulation was forced with varying climate from the JRA55-do data set and a preindustrial atmospheric $CO_2$ mixing ratio of 278 ppm. This experiment is referred as D and is used to separate the effects of rising atmospheric $CO_2$ and of climate change on the DIC inventory. Using the simulations A and B, the global

**Table 1.** List of simulations performed in this study.

| Experiment | Period | Atmospheric $CO_2$ | Atmospheric forcing |
|---|---|---|---|
| Pre-spin-up | 1611–1799 | Constant (278 ppm) | RYF61 |
| $A_{spinup}$ | 1800–1957 | Increasing | RYF61 |
| $B_{spinup}$ | 1800–1957 | Constant (278 ppm) | RYF61 |
| A | 1958–2021 | Increasing | JRA55-do |
| B | 1958–2021 | Constant (278 ppm) | RYF61 |
| D | 1958–2021 | Constant (278 ppm) | JRA55-do |

ocean anthropogenic $CO_2$ sink was estimated by taking the model biases and drift from the control run into account. We used a coupled system spin-up (i.e. a direct strategy; Séférian et al., 2016). Before starting simulations A, B, and D, we performed spin-up experiments in two stages. In the first stage, a 189-year long (equivalent to three cycles of JRA55-do forcing) preindustrial spin-up simulation (named the pre-spin-up) was conducted using RYF61 atmospheric forcing and a preindustrial atmospheric $CO_2$ mixing ratio of 278 ppm until the air–sea $CO_2$ reached a quasi-equilibrium state. The $A_{spinup}$ and $B_{spinup}$ simulations are a continuation of the pre-spin-up simulation, with either increasing ($A_{spinup}$) or constant ($B_{spinup}$) atmospheric $CO_2$, and run from 1800–1957. From the spin-up simulations, A, B, and D were branched off in 1958 and run until the end of 2021. FESOM1.4–REcoM2 and FESOM2.1–RECOM3 reach a throughput of 6 simulated years per day (SYPD) and 16 SYPD using the same mesh configuration and the same experimental set-up (see Table 1) on 288 cores with time steps of 15 and 45 min, respectively. All modelled mean fields shown in this work are averaged over the period 2012–2021, unless stated otherwise.

## 3 Results and discussion

In this section, we assess the performance of FESOM2.1–REcoM3 in simulating the observed mean state of nutrients, chlorophyll $a$, net primary production, and export production in the near-surface ocean, as well as air–sea $CO_2$ flux primarily under elevating $CO_2$. Before assessing the biogeochemical variables, we analyse the key features of the ocean model.

### 3.1 Modelled hydrography, mixed layer, and Atlantic meridional overturning circulation

An extended analysis of analogous FESOM2.1 runs is presented in Scholz et al. (2019, 2022). Here we analyse only a few relevant diagnostics to prove the validity of the presented research. We start the analysis by inspecting the spatial distribution of the model bias in the surface hydrography, which is presented in Fig. 3 as the difference between modelled mean 2012–2021 and the PHC3.0 Climatology (Steele et al., 2001). For temperature and salinity, respectively, we found a global spatial correlation coefficient ($r$) of 0.99 and 0.99, with a root mean squared error (RMSE) of 0.82 °C and 0.43 psu. In the northern North Atlantic, the bias is expressed by cold ($\sim 4$ °C colder) and fresh ($\sim 1$ psu fresher) anomalies around Newfoundland, which is a typical bias for standalone and climate models at coarse resolutions (see, e.g., Scaife et al., 2011). Further south, the bias depicts a dipole anomaly associated with the Gulf Stream going too far north, which is a commonly addressed shortcoming for non-eddy-permitting models (see, e.g., Zhang and Vallis, 2007; Storkey et al., 2018). Similar issues are found in comparable current systems, such as the Kuroshio and Malvina systems. It is, however, surprising that in general FESOM is far too saline at the surface and is on average 0.3 psu saltier than the climatology. The reason for this bias could be imperfections in the river discharge from CORE-II forcing and the relatively low surface salinity restoring, which uses a piston velocity of 50 m/300 d in the simulations. In most of the ocean, the sea surface temperature (SST) and sea surface salinity (SSS) differences act in an opposite manner on buoyancy. Hence, the increase or decrease in SST is accompanied by an increase or decrease in SSS. The only exception is the Indian Ocean, where east and west in simulation A become less and more buoyant, respectively (Fig. 3).

In Fig. 4, we augment the diagnostic by inspecting the Atlantic meridional overturning circulation (AMOC), which provides the most general characteristic of water mass transformation and production. The mean AMOC in both runs is expressed with the basin-wide mid-depth cell, showing a maximum of $\sim 15$ Sv at ca 40° N. The bottom cell, induced by the circulation of the Antarctic bottom water, is also well reproduced, with a minimum of $\sim -5$ Sv. Even though the runs depict large differences in temperature and salinity from the observed climatology, the simulated AMOC shows the canonical picture known from other standalone ocean and coupled climate models (Griffies et al., 2009; Jungclaus et al., 2013; Danabasoglu et al., 2014). This indicates that although biases in the representation of water mass properties and ventilation mechanisms are present, they still result in a reasonable density distribution which maintains realistic transport.

The difference between simulations A and B shows that the mid-depth and bottom cells are stronger in simulation B. Consequently, the difference A − B is expressed by a basin-

**Table 2.** List of the observational data sets used to initialize the biogeochemistry model and assess its performance. TS5

| Data set | Variable name | Unit | Reference |
|---|---|---|---|
| Dissolved inorganic carbon | DIC | $mmol\,m^{-3}$ | Global Ocean Data Analysis Project version 2 (Lauvset et al., 2016) |
| Total alkalinity | Alk | $mmol\,m^{-3}$ | Global Ocean Data Analysis Project version 2 (Lauvset et al., 2016) |
| Dissolved inorganic nitrogen | DIN | $mmol\,m^{-3}$ | World Ocean Atlas (Garcia et al., 2014) |
| Dissolved inorganic silicon | DSi | $mmol\,m^{-3}$ | World Ocean Atlas (Garcia et al., 2014) |
| Oxygen | $O_2$ | $mmol\,m^{-3}$ | World Ocean Atlas (Garcia et al., 2019b) |
| Chlorophyll *a* concentration | Chl | $mg\,m^{-3}$ | OC-CCI- (Sathyendranath et al., 2019) and Southern-Ocean-specific data set (Johnson et al., 2013) |
| Net primary production | NPP | $mmol\,m^{-3}$ | CbPM (Westberry et al., 2008) and VGPM (Behrenfeld and Falkowski, 1997) |

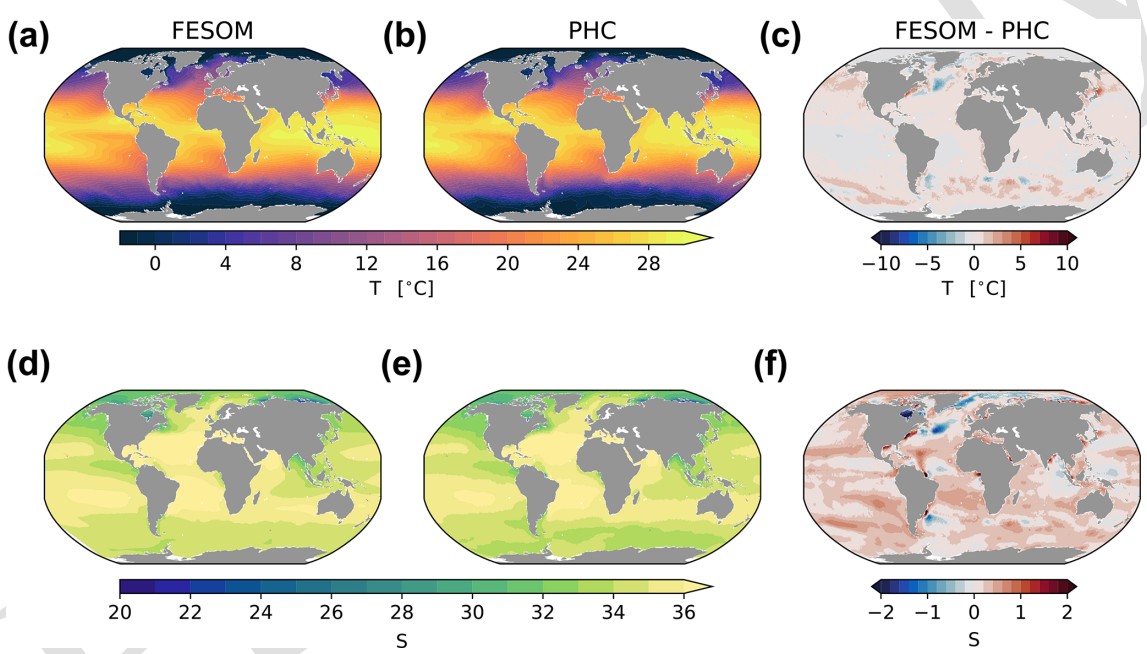

**Figure 3.** Maps of simulated FESOM2.1–REcoM3 (Sim A) surface temperature (°C) **(a)**, practical salinity (g kg$^{-1}$) **(d)**, with observations from the Polar science center Hydrographic Climatology (PHC3.0; updated from Steele et al., 2001), and corresponding differences **(c, f)**, averaged over the time period 2012–2021. TS6

wide positive anomaly, with a maximum of $\sim 3$ Sv. We also show the time series, for both runs, of AMOC maxima for the years 1958–2021 (Fig. 5). In simulation A, the time series depicts a multidecadal variability, with a minimum of $\sim 9.5$ Sv and a maximum of $\sim 13.5$ Sv. Concurrently, the reference simulation B depicts a nearly constant value, with a small increase between 9.5 and 10 Sv, which is a result of the repeat-year forcing.

Finally, in Fig. 6, we present the simulated and observed (Sallée et al., 2021; referred to as Atlas) annual maximum mixed layer depth (MLD) pattern for March and September, following the same methods (the depth at which the poten-

tial density referenced to the surface exceeds the density of the water by a threshold of $0.03\,kg\,m^{-3}$). Overall, the modelled MLD fits well with the observations, although some common discrepancies remained in the deep mixing areas. In the Northern Hemisphere, the deepest MLD ($> 1000\,m$) is found in the Labrador and Greenland–Iceland–Norwegian seas. The magnitude is larger than in Sallée et al. (2021) but is in the same range as in other modelling studies (Griffies et al., 2009; Sidorenko et al., 2011). In the Southern Hemisphere, winter deep mixing in high latitudes is also overestimated compared to the observations, especially in the Pacific sector of the Southern Ocean. From inspecting the model runs and

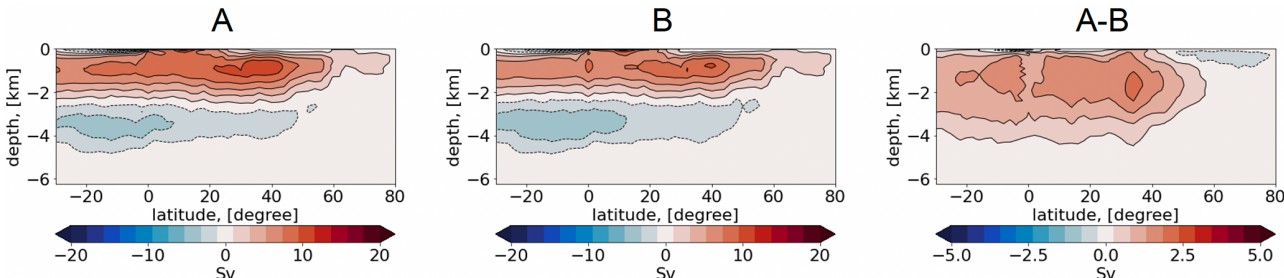

**Figure 4.** Vertical representation of the Atlantic meridional overturning circulation (Sv) in simulations A, B, and their difference (Sv).

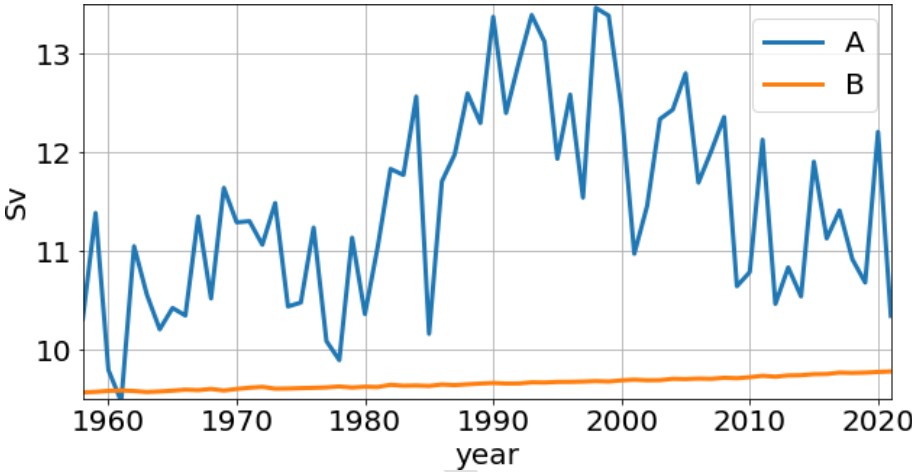

**Figure 5.** Time series of the annual mean Atlantic meridional overturning circulation (Sv) maxima in simulations A and B.

their differences, we conclude that FESOM2.1 simulated a reasonable ocean state which can be used for further analysis.

## 3.2 Nutrients, ocean productivity, and ecosystem

### 3.2.1 Modelled versus in situ nutrients

We first compared the spatial distribution of the surface (averaged over the top 100 m depth layer) ocean dissolved inorganic nitrogen (DIN) and dissolved silicate (DSi) from REcoM3 with the World Ocean Atlas 2018 (Garcia et al., 2019b) climatologies (Fig. 7). While simulated surface DIN concentrations were lower than observations in the subpolar regions, a large positive DIN bias of up to 20 mmol m$^{-3}$ was found in the subtropical South Pacific Ocean. The simulated DSi was overestimated in the Southern Ocean and underestimated in the northern Pacific. Exceptions are the Pacific and Atlantic sectors of the coastal Southern Ocean, where the modelled DSi concentrations are lower than the observations. These patterns were already present in FESOM1.4–REcoM2 (Schourup-Kristensen et al., 2014); however, two recent improvements should be noted. First, the large and positive DIN bias in the northern subtropical Pacific (Schourup-Kristensen et al., 2014) disappeared. This is caused by replacing the dust

deposition input forcing field from Mahowald et al. (2003) with Albani et al. (2014), which results in a more realistic (i.e. less strong) iron limitation. Second, the silicate bias in the Southern Ocean is reduced in magnitude and extent compared to Schourup-Kristensen et al. (2014). This is related to tuning experiments (not shown), which resulted in a larger share of diatoms in the Southern Ocean (Fig. 11) compared to Schourup-Kristensen et al. (2014), thus drawing down more silicic acid. Along with the increased share of diatoms, the Southern Ocean and global opal export also increased. For the global ocean, the opal export increased from 74.5 Tmol Si yr$^{-1}$ in Schourup-Kristensen et al. (2014) to 168 Tmol Si yr$^{-1}$ in the present study and is thus at the upper end instead of the lower end of the range of 69–185 Tmol Si yr$^{-1}$ (Dunne et al., 2007) and higher than the best estimate of Tréguer et al. (2021, Table 3). In the Southern Ocean, opal export increased from 21.5 Tmol Si yr$^{-1}$ in Schourup-Kristensen et al. (2014) to 85.5 Tmol Si yr$^{-1}$, which is higher than the range of 21–54 Tmol Si yr$^{-1}$ reported by Dunne et al. (2007). The silicic acid bias is rather insensitive to the formulation and parameter choice of opal dissolution but very sensitive to the share of diatoms in the Southern Ocean. The correlation coefficient ($r$) and root mean squared error (RMSE) between the simulated and ob-

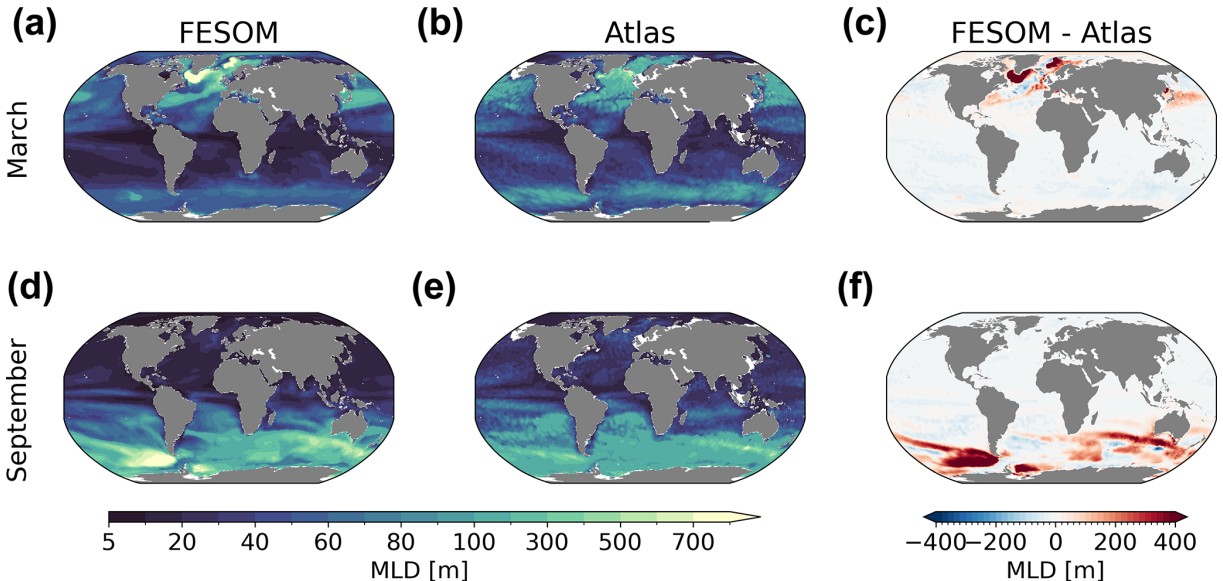

**Figure 6.** Maps of simulated FESOM2.1–REcoM3 (Sim A) maximum mixed layer depth (m) in March **(a)** and September **(d)**, averaged over the time period 2012–2021, using the observation-based maximum mixed layer depth from Sallée et al. (2021, **b** and **e**), which occupied the period between 1970 and 2018, and the corresponding differences **(c, f)**. TS7

served annual mean were 0.88 and 0.86 mmol m$^{-3}$ for DIN and 0.47 and TS8 0.54 mmol m$^{-3}$ for DSi. The correlation with observed DIN is higher than in Schourup-Kristensen et al. (2014, 0.75), which we relate to the disappearance of the DIN bias in the northern subtropical Pacific. The correlation with observed DSi is lower than in FESOM-1.4–REcoM2, despite the reduction in magnitude and extent in the Southern Ocean DSi bias. Moderately high silicic acid values in the northern high latitudes are not reproduced. This may be related to mixing that is too sluggish or to overly strong silicic acid drawdown by diatoms (Figs. 9 and A1), which is possibly linked to an iron limitation that may be too weak (Fig. 12).

Despite the enormous increase in the number of observations of dissolved iron with the GEOTRACES project, observations have not reached a global coverage that makes it possible to construct a global climatology. Therefore, the modelled dissolved iron is compared here to the global surface pattern of dissolved iron by Huang et al. (2022), which uses an artificial intelligence (AI) method (random forest) to construct a near-global iron field, based on the observations in the second intermediate GEOTRACES data product (Schlitzer et al., 2018), plus some older in situ iron observations compiled in Tagliabue et al. (2012), and on co-located hydrographic observations. The pattern of modelled dissolved iron (Fig. 8; averaged over the top 50 m) shows the expected pattern of high concentrations in regions with high dust deposition, mainly in the tropical Atlantic Ocean and the eastern part of the Arabian Sea and also to some extent in the southern subtropical Atlantic and Indian oceans. Concentrations are extremely low in the subpolar Southern Ocean and

in almost the whole equatorial and South Pacific. Iron concentrations are also low in the subpolar North Pacific, and – less so, but still noticeable – in the subpolar North Atlantic. Oceanic regions adjacent to extended shelves, especially in the Arctic, show somewhat elevated iron concentrations. If we compare this to the AI-generated global pattern of dissolved iron from Huang et al. (2022), then we find qualitatively similar patterns, like the elevated iron concentration in the equatorial and subtropical Atlantic and the Arabian Sea or the low concentrations in the subpolar Southern Ocean, the equatorial Pacific, and the subpolar North Pacific, but the amplitude of the patterns is smaller overall. The largest discrepancy in amplitude is found under the Saharan dust plume in the tropical Atlantic, where the model produces maximum dissolved iron values that are almost 3 times as high as the reconstruction from Huang et al. (2022). Direct observations in the tropical Atlantic also show dissolved iron concentrations that reach 1.2 nmol L$^{-1}$ (e.g. Hatta et al., 2015), while modelled maxima are $> 3$ nmol L$^{-1}$. A further important difference is that the distribution by Huang et al. (2022) shows slightly elevated iron concentrations in the centre of the subtropical South Pacific, where the model in contrast has extremely low values. This discrepancy causes an iron limitation that is too strong in this region in the model, probably explaining the overly high DIN concentrations in the model in the South Pacific. The fact that the amplitude of the patterns in modelled dissolved iron is too high, which is also found in other models, likely has a number of causes. The most important one is the assumption of a constant solubility in dust-deposited iron. Dust deposition close to the main source regions is on average coarser and has experienced less chemi-

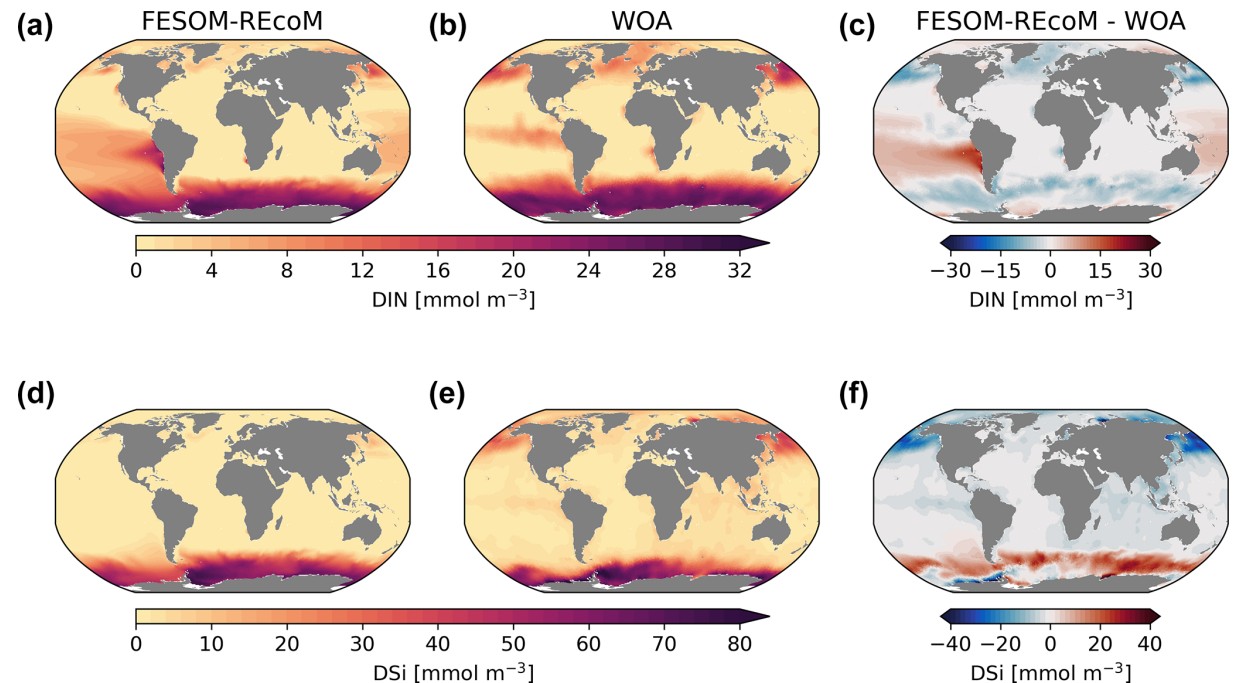

**Figure 7.** Maps of simulated FESOM2.1–REcoM3 (Sim A) surface (0–100 m) concentration of dissolved inorganic nitrogen (mmol m$^{-3}$; **a**), dissolved inorganic silicon (mmol m$^{-3}$; **d**), with observations from the World Ocean Atlas 2018 climatology (**b** and **e**; Garcia et al., 2019b), and the corresponding differences (**c, f**) averaged over the time period 2012–2021.

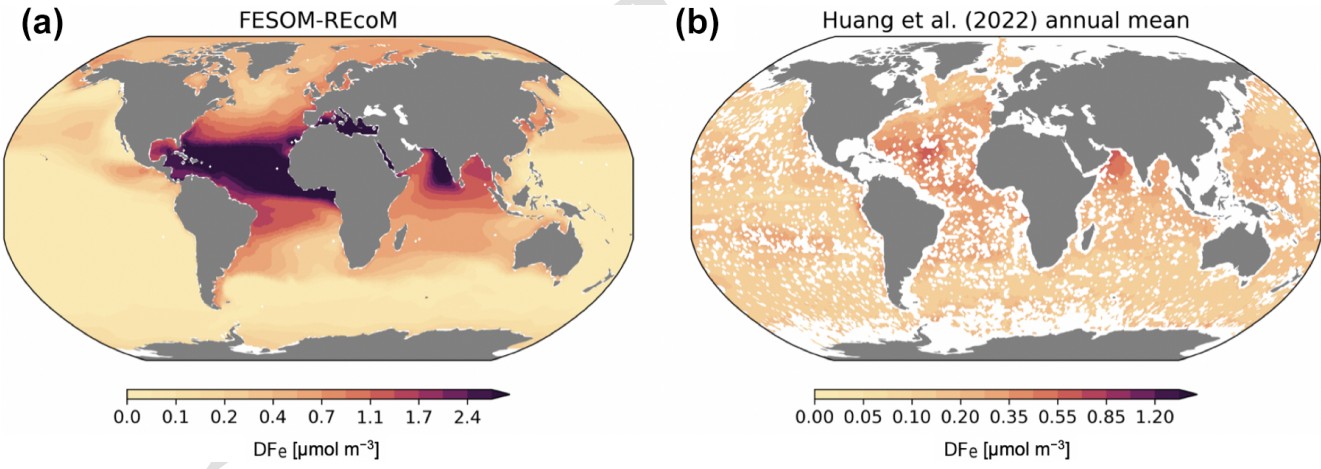

**Figure 8.** Maps of simulated FESOM2.1–REcoM3 (Sim A) surface (0–50 m) concentration of dissolved iron (µmol m$^{-3}$; **a**), and of the AI-based global reconstruction by Huang et al. (2022, **b**). Note the different colour scales for the two plots.

cal processing during its transport, both of which would lead to a lower solubility. The opposite is true for particles deposited far from the source regions, such as in the South Pacific. A second contribution might be the missing source from pyrogenic aerosols, which are far more soluble. Also, the effect of dust particles as iron scavengers, which has not been included in this simulation, has been shown to reduce the overly high dissolved iron concentrations often found in models in the main dust deposition regions (Ye and Völker,

2017; Pagnone et al., 2019). Furthermore, the intensity and extension of dust plumes vary between modelled dust deposition fluxes (e.g. Myriokefalitakis et al., 2018). The field of dust deposition by Albani et al. (2014), used in our model to calculate aeolian iron input, is within the range of modern estimates but surely contains some uncertainties. Despite the overall amplitude of the patterns in the dissolved iron being too strong, especially in the regions of high dust deposition, the model is able to reproduce the main regions in which iron

availability limits phytoplankton productivity (Moore et al., 2013), namely the subpolar Southern Ocean, the equatorial and North Pacific, and also to some extent the seasonal iron limitation in the subpolar North Atlantic (Nielsdóttir et al., 2009), but overestimates iron limitation in the subtropical South Pacific.

### 3.2.2 Modelled versus satellite-based phytoplankton biomass and productivity

We first evaluated the spatial distributions of the modelled chlorophyll $a$ concentration (Fig. 9), which is an indicator for phytoplankton biomass, and vertically integrated the net primary production (NPP, Fig. 10). We compared chlorophyll $a$ concentrations obtained from FESOM2.1–REcoM3 simulations, averaged from 2012 to 2021, with an ocean colour remote sensing merged data set (from the Ocean Colour Climate Change Initiative, OC-CCI; Sathyendranath et al., 2019), averaged from 1998 to 2019. We also compared the modelled NPP with satellite estimations, such as the Vertically Generalized Production Model (VGPM; Behrenfeld and Falkowski, 1997; Fig. 10) and the updated carbon-based productivity model (CbPM; Westberry et al., 2008; see Fig. A1 in the Appendix). VGPM is a chlorophyll-based algorithm that can be considered to be CE8 a standard NPP estimation from ocean colour for the last 20 years (Lee and Marra, 2022). CbPM uses spectrally resolved light attenuation and is based on a semi-analytical algorithm (Garver–Siegel–Maritorena, GSM; Maritorena et al., 2002). All data sets were also compared along a latitudinal distribution with an improved chlorophyll $a$ algorithm for the Southern Ocean (Johnson et al., 2013; Fig. 11).

The results for chlorophyll $a$ and NPP obtained here are comparable to those presented by Schourup-Kristensen et al. (2014). Over large parts of the global ocean, the mean surface chlorophyll $a$ concentrations are in agreement with the observations (Fig. 9c and d). Yet, there are regional differences. In temperate latitudes, the modelled chlorophyll $a$ concentrations are somewhat higher than observed, while the subtropical gyres show concentrations slightly lower than the observations. The comparison of modelled and observation-based satellite estimates of chlorophyll $a$ yielded a correlation of 0.66 and an RMSE of 0.38 mg m$^{-3}$. FESOM2.1–REcoM3 also shows a reasonably well-simulated latitudinal variation in chlorophyll $a$ compared to satellite estimations (Fig. 11a). The model underestimates chlorophyll $a$ concentrations in most of the coastal regions, especially in the high-latitude regions. In the southern high latitudes, FESOM2.1–REcoM3 follows the Southern-Ocean-adjusted chlorophyll $a$ data set (Johnson et al., 2013) quite well, except for the coastal regions close to Antarctica (approximately south of 70° S). In the Arctic Ocean, the model strongly underestimates the chlorophyll $a$ concentrations, which is driven by negative biases reaching up to 3 mg chlorophyll $a$ m$^{-3}$ on the continental shelves (Fig. 9). Although FESOM2.1–

REcoM3 did reproduce the NPP distribution at low latitudes well (Figs. 10 and 11), it also strongly underestimated the NPP at higher latitudes when compared to VGPM ($r = 0.43$; RMSE $= 0.34$ mgC m$^{-2}$ d$^{-1}$), in particular in productive areas north of 50° N and coastal areas (Fig. 10). For regional applications, further analysis and possibly tuning may be needed. When compared with VGPM, the model simulation generally underestimated the remotely sensed NPP estimations (Table 3), especially in the subtropical Pacific. Yet, with a value of 35.9 PgC yr$^{-1}$, the modelled global total NPP is slightly above the range of the earlier modelling studies (23.7–30.7 PgC yr$^{-1}$; Schneider et al., 2008) and within the range of recent Earth system models (24.5–57.3 PgC yr$^{-1}$; Séférian et al., 2020). It is lower than other satellite-based estimates of 47.3 PgC yr$^{-1}$ (Behrenfeld and Falkowski, 1997), 52 PgC yr$^{-1}$ (Westberry et al., 2008), and 48.7–52.5 PgC yr$^{-1}$ reported by Kulk et al. (2020).

Both simulated chlorophyll $a$ concentrations and NPP from FESOM2.1–REcoM3 seemed to be underestimated in coastal regions. Primary production and chlorophyll $a$ levels that are too low were particularly evident in coastal regions, which could be linked to deficiencies in either the chlorophyll $a$ data set and/or in the model. For the former, the chlorophyll $a$ OC-CCI data set and the CbPM primary production data set uses the GSM algorithm. GSM tries to distinguish the optical signatures from phytoplankton, particles, and dissolved organic matter but still requires regional tuning in coastal regions, where the presence of non-biotic optically active material (i.e. yellow substances and sediments) makes chlorophyll $a$ retrieval challenging (Blondeau-Patissier et al., 2014). The overestimation of chlorophyll $a$ in coastal waters is even more pronounced with the use of standard global chlorophyll algorithms in the VGPM primary production data set, such as OC4 that are only adapted to CASE-I waters (low influence of dissolved organic matter and non-algal particles). Therefore, both remotely sensed NPP estimations carry uncertainties related to the global algorithms. For example, turbid waters over the Arctic shelves are known to artificially increase both chlorophyll $a$ and NPP estimates from remote sensing (Matsuoka et al., 2012; Mitchell, 1992; Mustapha et al., 2012). Some recent advances used local parameterizations with in situ data, which resulted in much lower productivity levels in those coastal areas (Lewis et al., 2020; Lewis and Arrigo, 2020). Generally, the NPP and chlorophyll differences compared to satellite-based estimates could also be linked to model deficiencies, such as a coarse model resolution and associated weak upwelling, missing complexity in the simulated phytoplankton classes, and also the so-far unconsidered nutrient input from terrigenous sources.

The low values of primary production could be caused by several top-down and/or bottom-up effects. The nutrient dynamics that partly control NPP are the result of a delicate balance between physical (mixing, stratification, and upwelling systems) and biogeochemical processes. To investi-

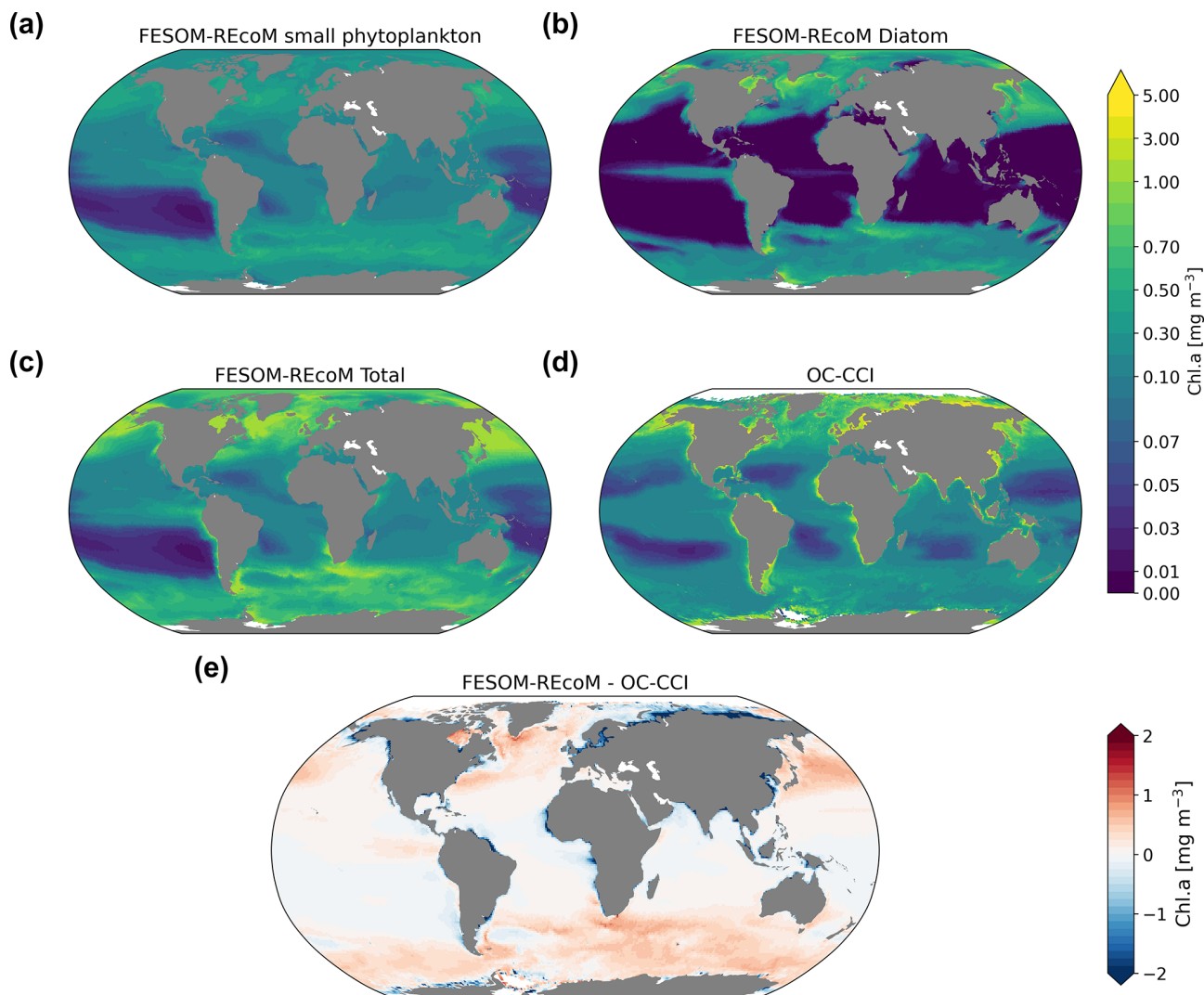

**Figure 9.** Maps of simulated FESOM2.1–REcoM3 (Sim A) surface ($\log_{10}$ transformed) chlorophyll *a* concentration (mg Chl m$^{-3}$) of small phytoplankton **(a)**, diatoms **(b)**, and the sum of both phytoplankton groups **(c)**. The satellite-based merged data set OC-CCI (Sathyendranath et al., 2019) is shown in panel **(d)**, with the corresponding differences between FESOM2.1–REcoM3 and OC-CCI shown in panel **(e)**. Note the different time periods for the simulation (2012–2021) and OC-CCI (1998–2019).

gate bottom-up controls on regional NPP dynamics, we derived the most limiting factor (either light or nutrients) of the growth of diatom and small phytoplankton. This factor ranges between zero (most limiting) and one (no limitation) and is based on the nutrient uptake Michaelis–Menten kinetics of REcoM. The Michaelis–Menten coefficient (MM) is computed as $\mathrm{MM} = [\mathrm{Nut}] / ([\mathrm{Nut}] + K\mathrm{Nut})$, with [Nut] being the nutrient concentration, and KNut a nutrient and phytoplankton-dependent half-saturation constant. The light limitation is defined as the carbon-specific photosynthesis rate divided by the maximum photosynthetic rate. We derived maps showing the most limiting factor (the factor closest to zero, with either the nutrients of DIN, DSi, or DFe or light) in the annual mean (Fig. 12).

The spatial distribution of the dominant growth-limiting factor for diatoms and small phytoplankton over the time period 2012–2021 is shown in Fig. 12. Over large areas of the South Pacific and almost the entire Southern Ocean, diatoms were limited by iron availability. Elsewhere, except for the Arctic Ocean, where light was the most limiting factor, diatom growth was controlled by the abundance of dissolved silicic acid. Nutrient uptake of small phytoplankton was limited by iron in the South Pacific, DIN within the band of 45° S–45° N in the Atlantic and Indian oceans, and insufficient light at high latitudes (south of 45° S and north of 45° N).

The large-scale patterns of limitation were in general agreement with observations (Moore et al., 2013) and other

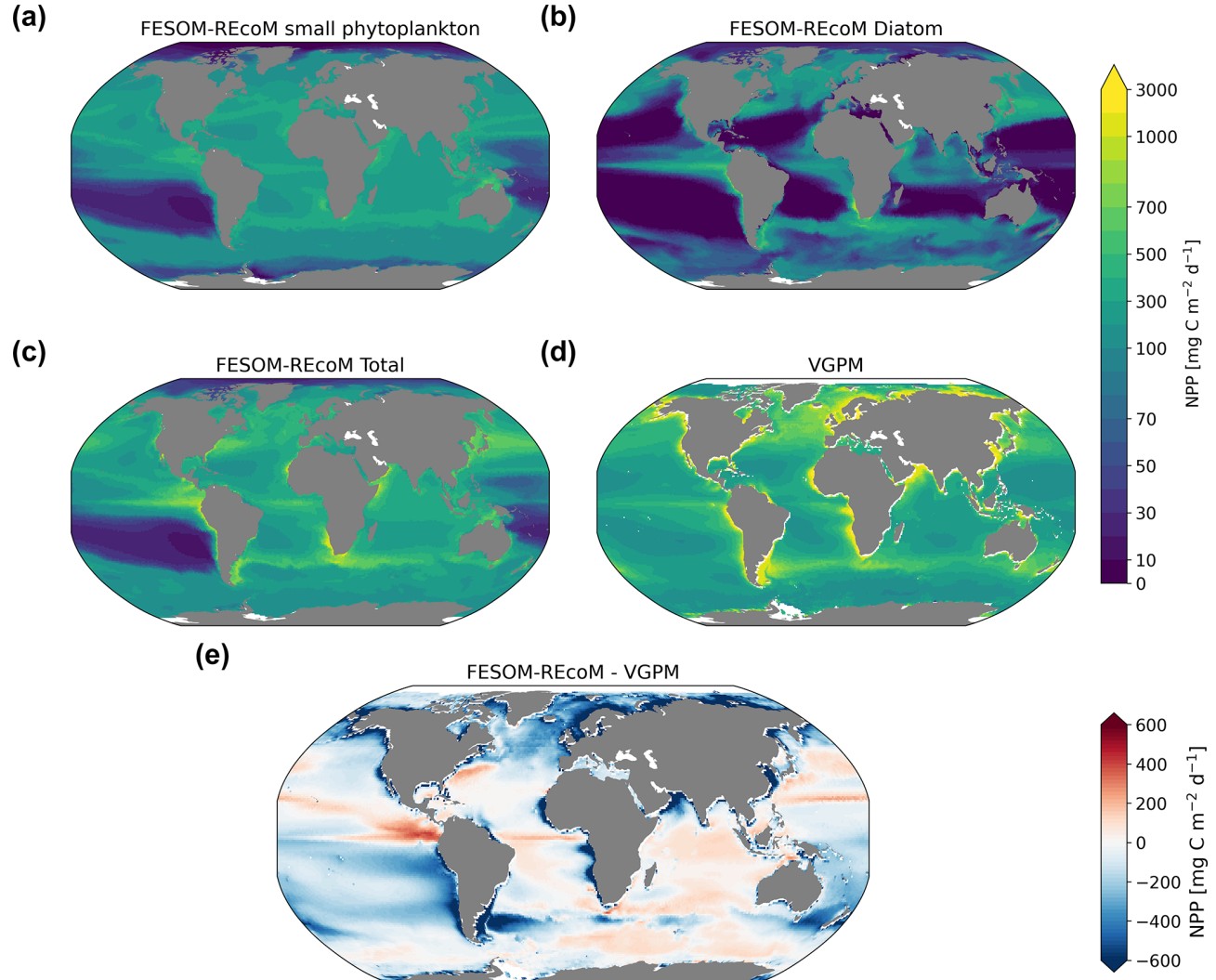

**Figure 10.** Maps of simulated FESOM2.1–REcoM3 (Sim A) vertically integrated net primary production ($mgC\,m^{-2}\,d^{-1}$) of small phytoplankton **(a)**, diatoms **(b)**, and the sum of both phytoplankton groups **(c)**. The satellite-based Vertically Generalized Production Model (VGPM; Behrenfeld and Falkowski, 1997) is shown in panel **(d)**, with the corresponding differences between FESOM2.1–REcoM3 and VGPM shown in panel **(e)**. All fields are averaged over the time period from 2012 to 2021.

modelling studies (Long et al., 2021a), although the degree of silicic acid limitation for diatoms (outside the iron-limited Southern Ocean) varied across models (Laufkötter et al., 2015). The severe iron limitation in most of the Pacific might contribute to the lower productivity levels than observed in the same regions (Fig. 10).

In addition to bottom-up explanations, the formulation and parameter choices for zooplankton grazing may be a reason for low primary production (Anderson et al., 2010; Prowe et al., 2012; Karakuş et al., 2021). In fact, Karakuş et al. (2022) demonstrated that a separation of the small zooplankton group in REcoM into micro- and mesozooplankton leads not only to a 25 % increase in NPP but also to a reduction in the overly strong iron limitation in the South Pacific, due to nutrient recycling by zooplankton. Further-

more, REcoM does not explicitly represent picophytoplankton (e.g. non-$N_2$-fixing cyanobacteria such as *Synechococcus* and *Prochlorococcus*) and nitrogen fixers, and this might contribute to an underestimation of NPP.

### 3.2.3 Modelled versus MAREDAT zooplankton biomass

In REcoM3, the small zooplankton group is widely spread in the global ocean and the highest biomass occurs in high-productivity regions (Fig. 13a). The macrozooplankton is present in the high latitudes (Fig. 13b), since it is parameterized as a polar macrozooplankton group (Karakuş et al., 2021). We compare the latitudinal distribution of the integrated modelled zooplankton biomass with gridded global

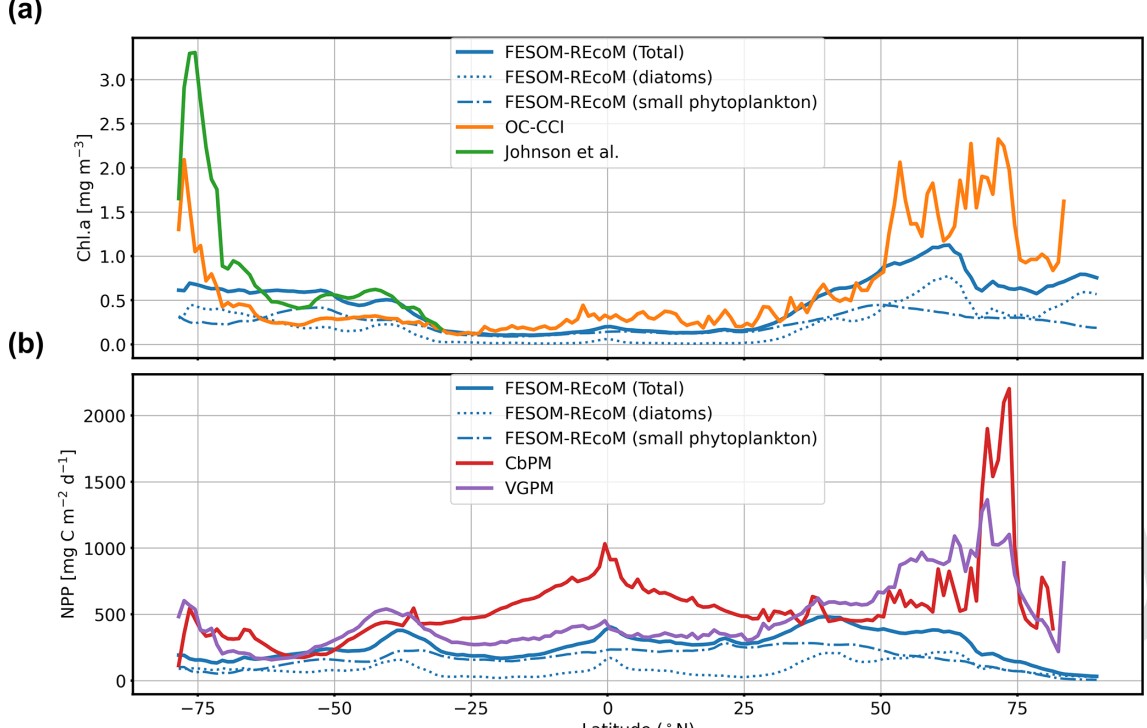

**Figure 11.** Latitudinal distribution of vertically integrated and zonally averaged **(a)** chlorophyll $a$ (mg Chl m$^{-3}$) and **(b)** net primary production (mg C m$^{-2}$ d$^{-1}$) simulated by FESOM2.1–REcoM3 (blue line). The satellite-based merged chlorophyll $a$ data sets of OC-CCI (Sathyendranath et al., 2019; orange line) and the improved chlorophyll $a$ algorithm for the Southern Ocean (Johnson et al., 2013; green line) are shown in panel **(a)**. The satellite-based data set of the Vertically Generalized Production Model (VGPM; Behrenfeld and Falkowski, 1997; purple line) and the carbon-based productivity model (CbPM; Westberry et al., 2008; red line) are shown in panel **(b)**.

zooplankton biomass data from MAREDAT (Buitenhuis et al., 2010; Moriarty et al., 2013; Moriarty and O'Brien, 2013). The simulated biomass of small and total zooplankton reproduces MAREDAT-derived biomass reasonably well in low to midlatitudes but underestimates biomass in the polar regions (Fig. 13c). The underestimation of zooplankton biomass in the northern high latitudes may be related to an underestimation of primary production in the same region. In agreement with the MAREDAT data set (Moriarty et al., 2013), macrozooplankton is not present in low latitudes.

### 3.2.4 Synthesis

The modelled biogeochemical fluxes were compared to the previous version FESOM1.4–REcoM2 and observational studies (Table 3). Modelled global NPP is higher in FESOM2.1–REcoM3 than in FESOM1.4–REcoM2 but still lower than in satellite-based estimates. The estimate is comparable to other global modelling studies (Schneider et al., 2008; Séférian et al., 2020). Export production (EP) is slightly higher in FESOM2.1–REcoM3 than in the previous version and falls within the observational range previously documented in the literature for both the global ocean and the Southern Ocean. For the global ocean, FESOM2.1–

REcoM3 NPP and EP estimations remained at the lower end of the range, despite a slight increase in NPP. A more detailed description of zooplankton results in more efficient nutrient recycling and can thus increase NPP by 25 % (see also explanation in Sect. 3.2.2; Karakuş et al., 2022). In the Southern Ocean, estimations of NPP and EP remained very close to observation-based estimates. Maybe the most noticeable change between the two model versions is the substantial increase in opal export, which increased by a factor of 4 in the Southern Ocean, passing from the lower to the higher end of the observational range of an earlier review (Dunne et al., 2007), and lies above an updated estimate (Tréguer et al., 2021). This is due to an increase in the relative contribution of diatoms to the total NPP in high latitudes (Fig. 11).

### 3.3 Carbon cycle

#### 3.3.1 Dissolved inorganic carbon and alkalinity

Insight into the carbonate system can be obtained by inspecting surface maps of modelled dissolved inorganic carbon and alkalinity and the corresponding observational Global Ocean Data Analysis Project (GLODAPv2) climatologies (Fig. 14). Global patterns of simulated concentra-

**Table 3.** Global and Southern Ocean net primary production (NPP) and export production (EP) in FESOM2.1–REcoM3 and estimates from the literature. The Southern Ocean is here considered to be the region south of 50° S. The numbers for VGPM and CbPM are recalculated after interpolation to the model mesh over the years 2012–2019.

| | Unit | FESOM1.4–REcoM2 (Sim. A) | FESOM2.1–REcoM3 (Sim. A) | Range from the literature |
|---|---|---|---|---|
| NPP global | PgC yr$^{-1}$ | 32.5 | 35.8 | 50.5 (VGPM; this study) <br> 68.9 (CbPM; this study) <br> 47.3 (Behrenfeld and Falkowski, 1997) <br> 52 (Westberry et al., 2008) <br> 23.7–30.7 (Schneider et al., 2008) <br> 48.7–52.5 (Kulk et al., 2020) <br> 24.5–57.3 (CMIP6; Séférian et al., 2020) |
| EP global | PgC yr$^{-1}$ | 6.1 | 6.3 | 9.6 (Schlitzer, 2004) <br> 5.8–13 (Dunne et al., 2007) <br> 5 (Henson et al., 2011) <br> 5.9 (Siegel et al., 2014) |
| Opal export global | Tmol Si yr$^{-1}$ | 74.5 | 168 | 69–185 (review in Dunne et al., 2007) <br> 112 (Tréguer et al., 2021) |
| CaCO$_3$ export global | PgC yr$^{-1}$ | 1.2 | 0.89 | 0.1–4.7 (Lee, 2001; Jin et al., 2006; Gangstøet al., 2008; Berelson et al., 2007; Dunne et al., 2007; Battaglia et al., 2016; Gehlen et al., 2006 TS9) |
| NPP Southern Ocean | PgC yr$^{-1}$ | 3.1 | 3.2 | 3.48 (VGPM; this study) <br> 3.92 (CbPM; this study) <br> 1.1–4.9 (Carr et al., 2006) <br> 5.7 (Behrenfeld and Falkowski, 1997) |
| EP Southern Ocean | PgC yr$^{-1}$ | 1.1 | 1.5 | 1.0 (Schlitzer, 2002; Nevison et al., 2012) |
| Opal export Southern Ocean | Tmol Si yr$^{-1}$ | 21.5 | 85.5 | 21–54 (Dunne et al., 2007) |
| CaCO$_3$ export Southern Ocean | PgC yr$^{-1}$ | | 0.31 | 0.018 (Dunne et al., 2007) |

tions resemble the observed fields reasonably well ($r = 0.99$; RMSE $= 36.5$ mmol m$^{-3}$; calculated from annual means), with the highest DIC values in the subtropical gyres of the Atlantic and south Pacific and the subpolar North Atlantic and the Southern Ocean. Similar to GLODAP, the highest alkalinity values are found in the subtropical gyres of the Atlantic and south Pacific, with a good agreement with global spatial features ($r = 0.99$; RMSE $= 33.9$ mmol m$^{-3}$). Yet, simulated surface DIC and alkalinity concentrations were slightly overestimated in most of the global ocean. Two major exceptions are the Arctic Ocean and the North Atlantic, where the concentrations were underestimated, and the tropical upwelling regions and the northern Indian Ocean for DIC. The bias patterns differ relative to FESOM-1.4–REcoM (which was too low for DIC and alkalinity in the tropics and subtropics and too high in the high latitudes; not shown), which indicates that different realizations in circulation or mixing may drive these bias patterns. This is in line with an overestimation of surface salinity in most of the global ocean, with the exception of the North Atlantic and the Arctic Ocean (see Fig. 3). Also, surface alkalinity biases are generally attributed to a dominant physical (preformed) signal, with a smaller contribution from the calcium carbonate cycle and a

negligible contribution from organic matter remineralization (Koeve et al., 2014). However, tuning the model to result in a higher CaCO$_3$ production could possibly also counteract the positive alkalinity bias. Similarly, a higher NPP in the South Pacific could regionally ameliorate the high DIC bias. A positive bias in alkalinity at constant atmospheric CO$_2$ in the spin-up (not shown) and simulation A (Fig. 14) leads to a positive bias in DIC, as surface water with a higher alkalinity can hold more CO$_2$ in equilibrium than a low-alkalinity surface ocean. The range of biases is similar ($\pm 100$ mmol m$^{-3}$) to other ocean biogeochemical models (e.g. Tjiputra et al., 2020; Long et al., 2021a).

### 3.3.2  Surface ocean $p\mathrm{CO_2}$ and air–sea CO$_2$ flux

We compare the pattern of the temporal mean (2012–2021) surface ocean partial pressure of CO$_2$ ($p$CO$_2$; Fig. 15) and air–sea CO$_2$ flux (Fig. 16) to the $p$CO$_2$-based data product of Chau et al. (2022), with a seamless coverage from open ocean to the coasts (Fig. 16). Different $p$CO$_2$ products largely agree with each other in terms of spatial patterns, although they differ with respect to amplitude and timing of variability in the regionally or globally integrated fluxes (Fay

**(a)**

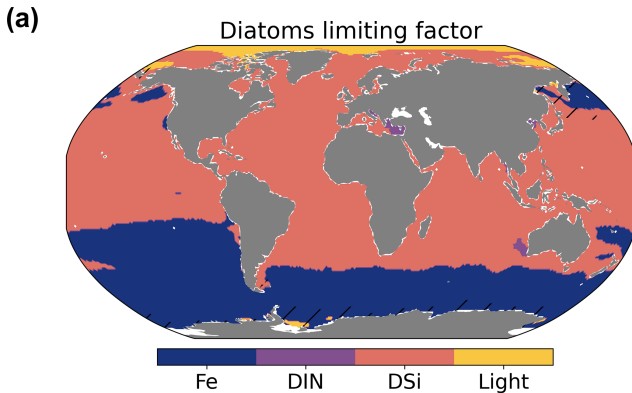

**(b)**

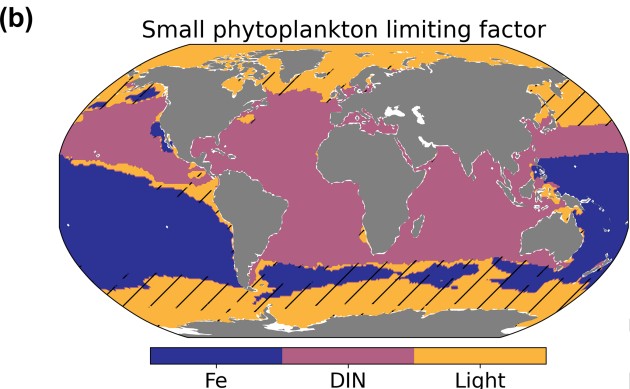

**Figure 12.** Maps showing the simulated spatial distribution of the most limiting factor in the surface water for diatoms **(a)** and small phytoplankton **(b)**. Hatching denotes areas of weak limitation (all limiting factors > 0.5). Note that Fe is for iron, DIN is for dissolved inorganic nitrogen, and DSi is for dissolved silicic acid.

et al., 2021; Fay and McKinley, 2021). Therefore, we chose one product (Chau et al., 2022) and focus on the comparison of the spatial pattern with our model. We further evaluate the temporal evolution of $pCO_2$ in FESOM2.1–REcoM3 with a direct comparison to surface ocean $pCO_2$ observations from the Surface Ocean $CO_2$ Atlas (SOCAT; Bakker et al., 2016), where we subsampled the model output for spatiotemporal locations where observations exist, following Hauck et al. (2020) and Friedlingstein et al. (2022b) in Fig. 17.

The large-scale spatial patterns of $pCO_2$ are well reproduced (Fig. 15) with high values in the tropics that are typically higher than atmospheric values (red colours) and lower values in the subpolar Southern and Pacific oceans and the high-latitude North Atlantic ($r = 0.75$ and RMSE = 29.2 µatm; ice-free areas). However, compared to the $pCO_2$ product of Chau et al. (2022), the model $pCO_2$ values are overestimated in the subtropical gyres (Fig. 15c). Furthermore, the North Atlantic $pCO_2$ is on average lower than the $pCO_2$ product, and the two data sets also differ in terms of the over- versus undersaturation of $pCO_2$ relative to the atmosphere in the polar Southern Ocean (higher values in FESOM2.1–REcoM3). The latter may well be explained by a known summer bias in the Southern Ocean $pCO_2$ observations (e.g. Metzl et al., 2006; Gregor et al., 2019). FESOM2.1–REcoM3 also simulates very high $pCO_2$ values on the Russian shelves in the Arctic, where hardly any observations exist. Similarly high $pCO_2$ values were reported for this region by Anderson et al. (2009), but missing repeat observations prevent a conclusion on whether this is a robust signal and what its extent in time and space is.

FESOM2.1–REcoM3 reproduced the temporal evolution of surface ocean $pCO_2$ reasonably well compared to SOCAT when accounting for where and when the $pCO_2$ sampling took place (Fig. 17). The annual correlation coefficient and RMSE between the simulated and observed global mean $pCO_2$ are 0.93 and 4.6 µatm, respectively. The subsampled model follows the SOCAT time series closely, including its variability due to a sampling distribution in space and time. The global mismatch with SOCAT $pCO_2$ as measured by the RMSE is comparable to or slightly below the value for FESOM-1.4–REcoM2 (see Fig. S9 in the Supplement of Hauck et al., 2020, for 1985–2018) and comparable to but at the high end of the range of other models in GCB 2022 (1990–2021; Friedlingstein et al., 2022b). On a monthly scale, the RMSE is higher (38 µatm), as the models capture the large-scale patterns better than smaller-scale variability, according to a previous assessment (Hauck et al., 2020). An analysis of large-scale regional patterns (north, tropics, and south; Fig. 17) reveals that the model reproduced the trend well but overestimates the mean $pCO_2$ in the tropics and underestimates $pCO_2$ in the northern extratropics and to a lesser extent in the southern extratropics in recent decades, as also indicated in the maps (Fig. 15).

The air–sea $CO_2$ flux spatial pattern was reasonably reproduced by FESOM2.1–REcoM3, with $CO_2$ uptake in the subpolar regions of both hemispheres and outgassing in the tropics and north Pacific (Fig. 16; $r = 0.72$; RMSE = 1.45 mol C m$^{-2}$ yr$^{-1}$). Generally, the $CO_2$ flux patterns mirror the $pCO_2$ patterns (Fig. 15) but with the additional imprint of spatial variability in the wind speed. Hence, the $CO_2$ uptake in the subpolar Southern Ocean may appear large compared to $pCO_2$, which is not as strongly undersaturated in the South Atlantic as in the North Atlantic. Regions of mean outgassing in the Southern Ocean are seen to a smaller extent in the model than in the $pCO_2$ product. While it is well established that the outgassing of $CO_2$ in the polar Southern Ocean occurs in winter (e.g. Bakker et al., 1997), its magnitude and timing varies between estimates and is being debated (Gruber et al., 2009; Lenton et al., 2013; Gray et al., 2018; Bushinsky et al., 2019; Sutton et al., 2021; Long et al., 2021b). The misfit between the annual mean modelled $CO_2$ flux and the $pCO_2$-based data product generally mimics $pCO_2$ biases and thus shows small positive differences (less uptake or more outgassing) in the subtropical gyres and small negative differences (stronger uptake or less outgassing) in the equatorial Pacific and the Southern Ocean (Fig. 16; bottom panel). The strongest biases were

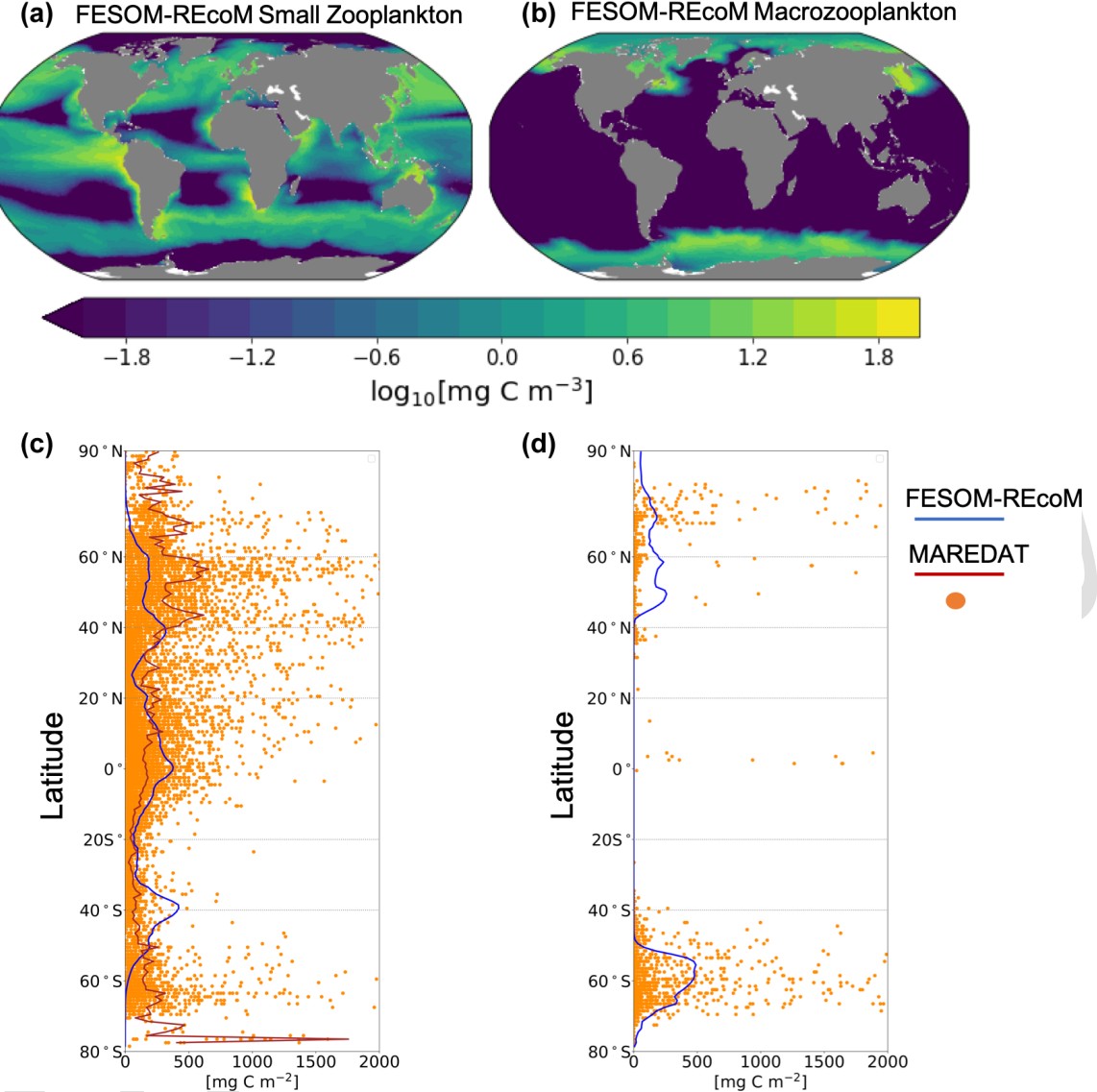

**Figure 13.** Maps of annual mean surface **(a)** small zooplankton and **(b)** macrozooplankton concentrations in FESOM2.1–REcoM3 (simulation A). Latitudinal distribution of vertically integrated zooplankton biomass (mg C m$^{-2}$) of **(c)** modelled small zooplankton (solid blue line) and the sum of microzooplankton and mesozooplankton from MAREDAT (orange dots for individual observations and solid brown line for the zonal mean of the observations; Buitenhuis et al., 2010; Moriarty and O'Brien, 2013) and **(d)** modelled macrozooplankton (solid blue line) and macrozooplankton from MAREDAT (orange dots; Moriarty et al., 2013). The modelled zooplankton biomass is averaged over the time period from 2012 to 2021. The zonal mean of macrozooplankton is not calculated due to the low number of observations.

found in the northern high latitudes (negative bias) and the upwelling zone of the eastern tropical Pacific (positive bias). FESOM2.1–REcoM3 also generally captures the large-scale patterns of coastal $CO_2$ fluxes with $CO_2$ uptake in the mid- and high latitudes (poleward of 25° N/S) and outgassing in the tropical coastal ocean, as described in a recent synthesis based on low- and high-resolution models and $pCO_2$ products (Resplandy et al., 2023). The large mismatch in $pCO_2$ on the Siberian shelves does not show up in the $CO_2$ flux, as sea ice prevents $CO_2$ outgassing throughout most of the year.

We continue our investigation with the analysis of the global ocean–atmosphere $CO_2$ flux time series (Fig. 18). In 1800, the first year of spin-up after the first 189 years of pre-spin-up of simulation B, the global ocean–atmosphere $CO_2$ flux was already relatively stable and converged towards a value close to zero. Under the assumption that the ocean and atmosphere were in equilibrium at constant preindustrial $CO_2$ values and without riverine carbon being transported into the ocean (Aumont et al., 2001; Resplandy et al., 2018; Regnier et al., 2022), an equilibrium flux of zero is expected for simulation B. Any deviation from this can be

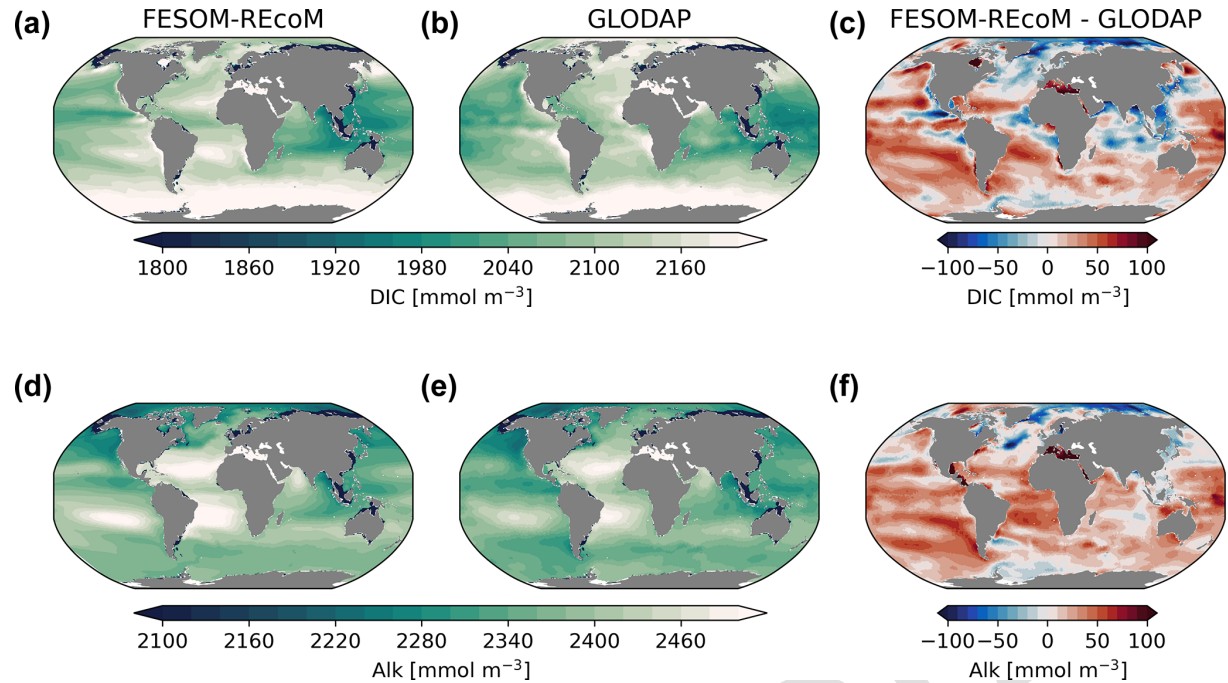

**Figure 14.** Maps of simulated FESOM2.1–REcoM3 (simulation A) surface (0–100 m) concentration of dissolved inorganic carbon (mmol m$^{-3}$; **a**) and alkalinity (mmol m$^{-3}$; **d**). Corresponding data from GLODAPv2 (**b**, **e**; Lauvset et al., 2016) and model–data differences (**c**, **f**) are shown. The comparison is for the period 1998–2006, which is centred on the year 2002 to be comparable to GLODAP.

considered a bias (Hauck et al., 2020). The global bias of the annual air–sea $CO_2$ flux in the FESOM2.1–REcoM3 control simulation amounts to $-0.12$ PgC yr$^{-1}$. The control simulation conducted with the older model version FESOM1.4 had a larger bias, with a positive flux of around $0.4$ PgC yr$^{-1}$ at the end of the simulation. In addition to the bias, the drift is reduced from $0.00264$ PgC yr$^{-2}$ in FESOM1.4–REcoM2 to $-0.00011$ PgC yr$^{-2}$ in FESOM2.1–REcoM3 with a longer spin-up. Despite different spin-up procedures (FESOM1.4 has a shorter spin-up period), simulation A with both FE-SOM2.1 and FESOM1.4 reveals similar $CO_2$ fluxes under interannually varying forcing after 1980, which indicates a dominance of the forcing over the initial conditions. This also questions the common assumption that the same bias occurs in the control and historical simulations.

We next assess the model performance of the interannually varying simulation A by comparing it with the Global Carbon Budget's ensemble of $p$CO$_2$-based data products and other ocean biogeochemistry models (Fig. 19). Note that all model time series shown in Fig. 19 are referenced relative to their control simulations, with a constant atmospheric $CO_2$ concentration and without climate change forcing (simulation B). Although being consistent with the interannual variability, air–sea $CO_2$ fluxes of FESOM1.4 are at the lower end of the range compared to other global ocean biogeochemistry models and $p$CO$_2$-based estimates. In contrast, starting from the mid-1960s, FESOM2.1 shows a higher $CO_2$ flux in comparison to FESOM1.4. Considering the fact that both model

versions do not differ much from each other in simulation A, the increase in the net $CO_2$ flux is mostly attributed to the level of $CO_2$ fluxes in their control simulations (Fig. 18).

After accounting for the bias in simulation B, the simulated ocean carbon sink (1990–1999) is $1.74 \pm 0.11$ and $2.17 \pm 0.13$ PgC yr$^{-1}$ for FESOM1.4–REcoM2 and FESOM2.1–REcoM3, respectively. Hence, FESOM2.1–REcoM3 is closer to the best estimate for the 1990s ($2.2 \pm 0.4$ PgC yr$^{-1}$; IPCC; based on seven different methodologies; Denman et al., 2007; Ciais et al., 2014) than FESOM1.4–REcoM2. The cumulative uptake over the period of 1959–2019 amounts to $93.4$ PgC (FESOM1.4–REcoM2) and $116.6$ PgC (FESOM2.1–REcoM3), which is a $25\%$ increase in $CO_2$ flux. Yet, the FESOM2.1–REcoM3 $CO_2$ fluxes have been lower than the mean of the $p$CO$_2$-based data products since about 2008 and thus affirm the growing discrepancy between global ocean biogeochemistry models and $p$CO$_2$ products (Friedlingstein et al., 2022a). It is likely that the models underestimate the mean ocean carbon uptake (Friedlingstein et al., 2022a), which is linked to biases in ventilation (Goris et al., 2018; Terhaar et al., 2021, 2022; Bourgeois et al., 2022) and surface ocean buffer capacity (Vaittinada Ayar et al., 2022; Terhaar et al., 2022), and it is thus encouraging that FESOM2.1–REcoM3 has a comparatively high mean flux (Fig. 19). The $p$CO$_2$ products are statistical models that interpolate and extrapolate sparse $p$CO$_2$ observations and have substantial uncertainties themselves. In particular, they are sensitive to sparse and unevenly dis-

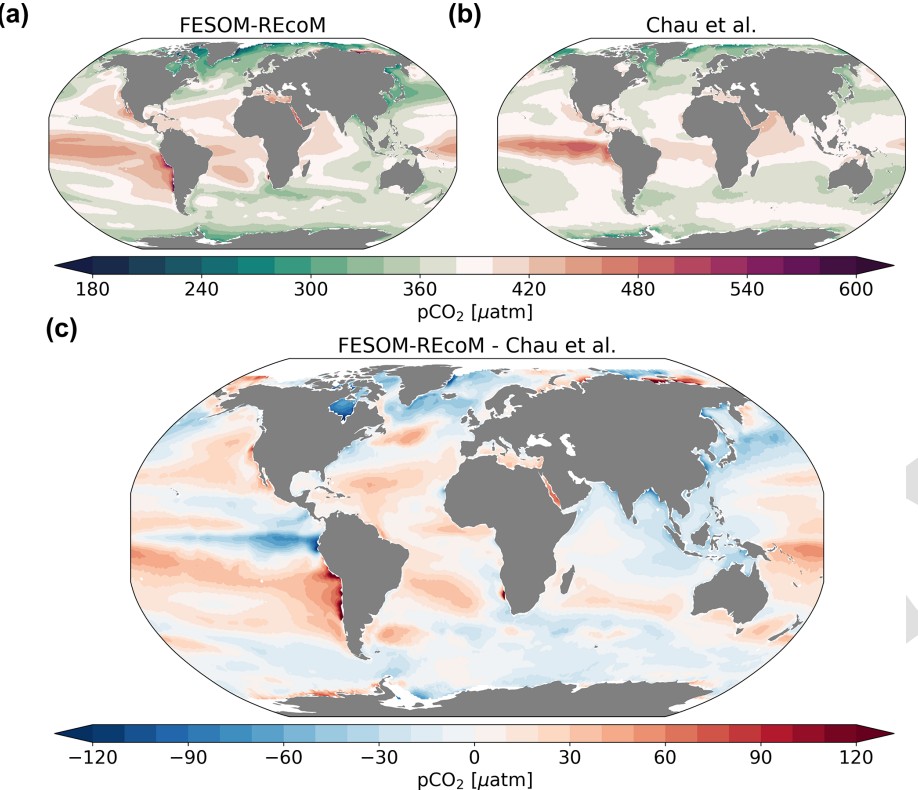

**Figure 15.** Maps of surface ocean $p$CO$_2$ (µatm). The top row compares the **(a)** simulated FESOM2.1–REcoM3 (simulation A) surface partial pressure of CO$_2$ to the **(b)** $p$CO$_2$-based data product (Chau et al., 2022) and both are averaged over 2012–2020. Panel **(c)** shows model–data differences. Note that simulation and observations are masked every month with the sea ice concentration > 15 %.

tributed observations (e.g. Gloege et al., 2021; Hauck et al., 2023), and it was shown that two of these methods overestimate the decadal CO$_2$ flux trend (2000–2018) by 20 %–35 %, based on the current $p$CO$_2$ observation distribution using a synthetic data set (Hauck et al., 2023).

### 3.3.3 DIC inventory changes

The interior ocean DIC inventory in FESOM2.1–REcoM3 amounts to about 38 200 PgC, which is in the reported range of 37 200 ± 200 to 39 000 PgC (Sundquist, 1985; Keppler et al., 2020). The DIC inventory is thought to change primarily in response to the rise in atmospheric CO$_2$, and the resulting DIC inventory change is often referred to as anthropogenic carbon. Effects of climate change (warming and circulation changes) are thought to be 1 `CE10` order of magnitude smaller, have the opposite sign, and affect both the natural carbon cycle and the anthropogenic carbon uptake and inventory (see Hauck et al., 2020; Friedlingstein et al., 2022a; Crisp et al., 2022, for more details on the different simulations and carbon components). The observation-based estimates of the anthropogenic DIC inventory change use back-calculation techniques to separate anthropogenic carbon changes from the vast natural carbon reservoir. Thus, we here analyse the DIC inventory change in FESOM–REcoM

for Sim A minus B, which quantifies the total DIC inventory change while accounting for model drift. In addition, to derive the comparable DIC inventory change component as in observation-based studies, we make use of a third simulation (called simulation D), which is forced by interannually varying climate and preindustrial atmospheric CO$_2$. Quantifying the DIC inventory change over a specific period from simulation A minus D is then coherent with the anthropogenic carbon definition used in Gruber et al. (2019).

The DIC inventory grew over time, in accordance with observation-based estimates (Table 4; Sabine et al., 2004; Gruber et al., 2019). The simulated anthropogenic DIC inventory change 1800–1994 (119 PgC) is in good agreement with the observation-based anthropogenic DIC inventory change (118 ± 19 PgC). For the total DIC inventory change, FESOM2.1–REcoM3 estimates a somewhat higher number (121 PgC; compared to 111 ± 21 PgC) but is well within the reported uncertainty in the observation-based estimate. For the period 1994–2007, the total DIC increase is 29.9 PgC (simulation A minus simulation B; i.e. drift corrected) and thus slightly lower than the estimate by Gruber et al. (2019). However, Gruber et al. (2019) only quantify the ocean anthropogenic DIC inventory increase (34 ± 4 PgC) and neglect the counter-effect of climate change. When consider-

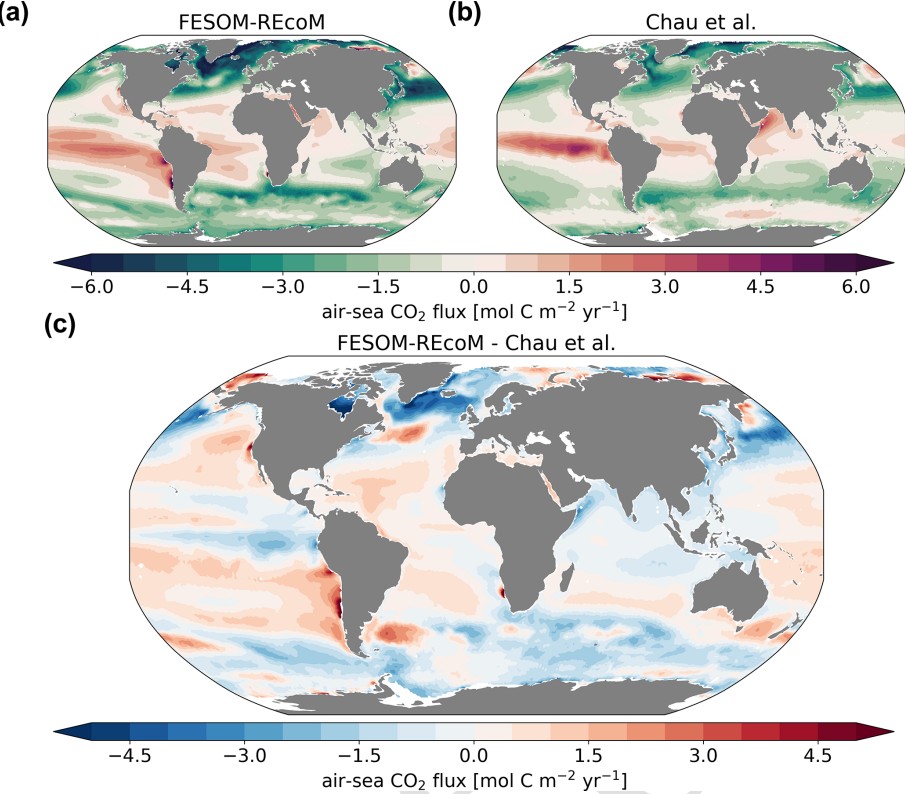

**Figure 16.** Maps of air–sea $CO_2$ fluxes (mol C m$^{-2}$ yr$^{-1}$). The top row compares the **(a)** simulated FESOM2.1–REcoM3 (simulation A) $CO_2$ flux to the **(b)** $pCO_2$-based data product (Chau et al., 2022) and both are averaged over 2012–2020. Panel **(c)** shows model–data differences. Negative numbers indicate a flux into the ocean. Note that simulation and observations are masked every month with the sea–ice concentration > 15 %.

ing the poorly constrained response of the natural carbon inventory to climate change, the model estimate falls within the uncertainty range of Gruber et al. (2019, $29 \pm 5$ PgC). Alternatively, estimating the anthropogenic DIC inventory change from simulations A minus D leads to 30.9 PgC, which is within the uncertainty range of Gruber et al. (2019)'s $34 \pm 4$ PgC. In GCB 2022, only four models simulated an anthropogenic DIC inventory change (1994–2007; simulation A minus D) $\geq 30$ PgC (i.e. within the Gruber et al., 2019, uncertainty range). The other six models ranged between 25.5 and 28.3 PgC, and the model ensemble mean was $28.3 \pm 2.6$ PgC (Friedlingstein et al., 2022a). FESOM2.1–REcoM3 is thus one of the few ocean biogeochemistry models that falls within the range of interior ocean anthropogenic carbon accumulation that is also supported by $O_2/N_2$ ratios (Tohjima et al., 2019) and atmospheric inversions (also see the discussion in Friedlingstein et al., 2022b). Notably, FESOM2.1–REcoM3 reproduces the latitudinal distribution of anthropogenic carbon accumulation in 1994–2007, with the maximum in the tropics (30° S–30° N), followed by the Southern Ocean (south of 30° S), and the north (north of 30° N). However, it also underestimates the accumulation in the tropics, as most other models do (Friedlingstein et al.,

2022a). If the observation-based assessment of the DIC inventory changes in the north, tropics, and south is correct, then this may indicate transport of anthropogenic carbon from the Southern Ocean into the tropics that is too weak or an air–sea $CO_2$ flux in the tropics with too little ocean uptake (or too much release) of $CO_2$.

## 3.4 Oxygen

The simulated distributions of global $O_2$ concentration at the surface ocean and intermediate depths was consistent with the observed patterns in the World Ocean Atlas (WOA) 2018 (Fig. 20; with $r = 0.98/0.91$ and RMSE $= 19.6/38.4$ mmol m$^3$ for surface (0–10 m) and intermediate depths (300–500 m), respectively). The model successfully reproduced the typical spatial patterns (Schmidtko et al., 2017), including (1) oxygen minimum zones in the western boundary upwelling systems, where old deoxygenated waters are brought to the surface, (2) high concentrations in the high-latitude regions, where cold temperature increases oxygen solubility (Arctic and Southern oceans), and (3) moderate oxygen concentrations in the more stratified tropical gyres. Nevertheless, there were regional discrepancies. At the surface, the model slightly un-

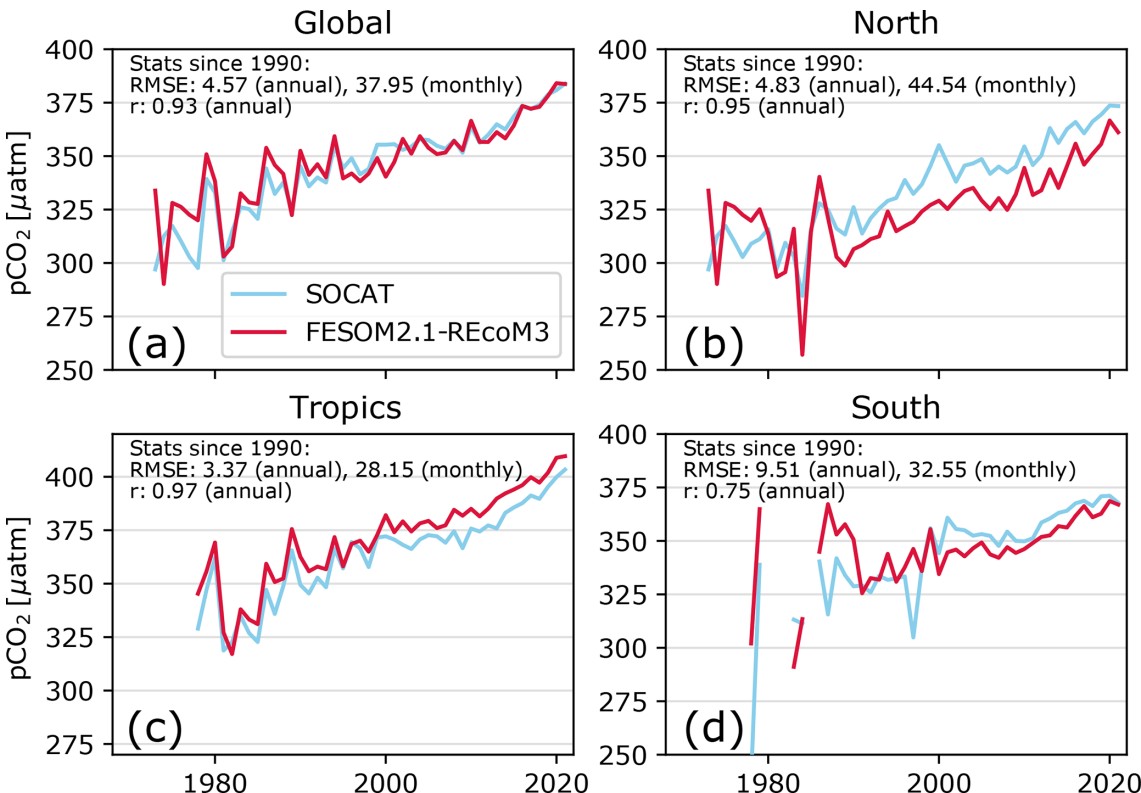

**Figure 17.** Comparing the annual mean $p$CO$_2$ (µatm) from FESOM2.1–REcoM3 (simulation A; subsampled for spatiotemporal locations of observations in SOCAT; red) with observations from SOCATv2022 (light blue; updated from Bakker et al., 2016). Results are shown as spatially averaged for **(a)** the global ocean, **(b)** the north ($> 30°$ N), **(c)** the tropics ($30°$ S–$30°$ N), and **(d)** the south ($< 30°$ S). The time series are shown for all observations in SOCAT (since 1970), but the correlation coefficient $r$ (unitless) and root mean squared error (RMSE; µatm) are indicated in the panels for the time period 1990–2021.

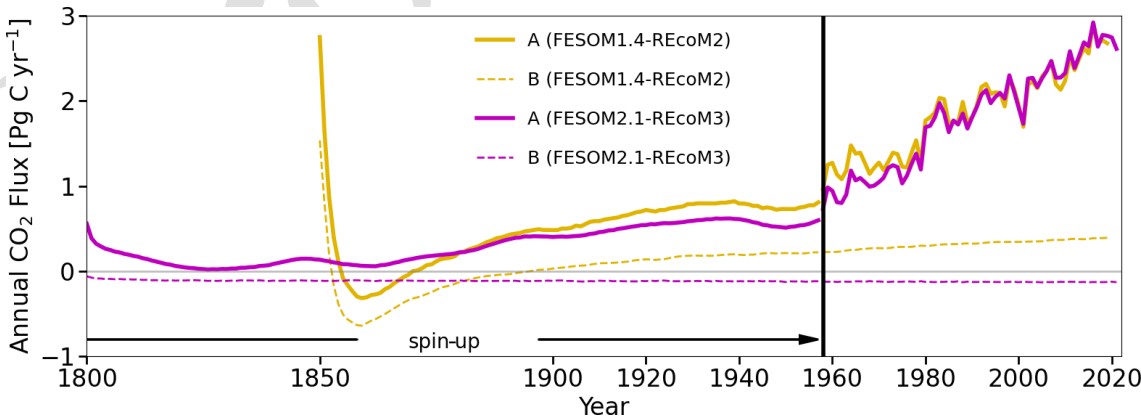

**Figure 18.** Time series of simulated annual mean global ocean–atmosphere CO$_2$ flux (in PgC yr$^{-1}$) in the experiments conducted in this study. FESOM2.1–REcoM3 spin-up was conducted for 347 years (including 189 years of pre-spin-up; not shown in the plot) under repeat-year forcing taken from the year 1961 (RYF61). Here we show the spin-up since 1800 that has been continued as the control simulation B after 1958 for FESOM-1.4–REcoM2 (yellow) and FESOM2.1–REcoM3 (magenta), with a constant CO$_2$ concentration of 278 ppm (dashed lines) and the spin-up under increasing CO$_2$ that has been continued as simulation A after 1958 (solid lines). The control simulation B started in the year 1958 and was conducted for 64 years with RYF61 (dashed lines). Simulation A also started in 1958 and was forced with interannual varying forcing JRA55-do-1.5.0 (solid lines). Please note that the spin-up period for FESOM1.4–REcoM2 and FESOM2.1–REcoM3 differs CE9 from each other, with the latter being longer than the former.

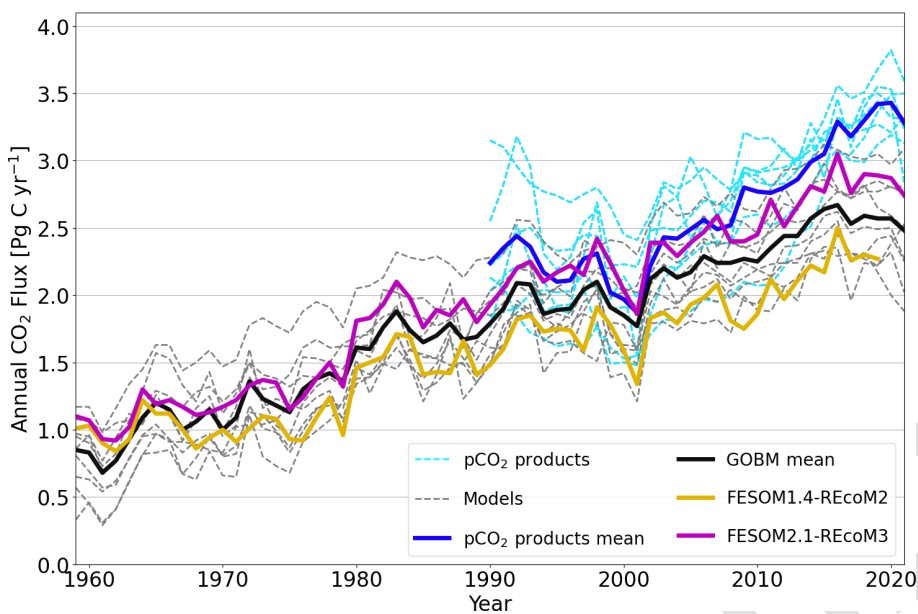

**Figure 19.** Globally integrated annual air–sea $CO_2$ flux from 10 global ocean biogeochemistry models (GOBMs) and seven $pCO_2$-based data products used in the Global Carbon Budget 2022 (Table 4 in Friedlingstein et al., 2022a), namely after applying bias correction to the models and river flux adjustment of $0.65\,\mathrm{PgC\,yr^{-1}}$ (Regnier et al., 2022) to the $pCO_2$ products (Wright et al., 2021; Schwinger et al., 2016; Lacroix et al., 2021; Berthet et al., 2019; Hauck et al., 2020; Liao et al., 2020; Doney et al., 2009; Aumont et al., 2015; Nakano et al., 2011; Urakawa et al., 2020; Long et al., 2021a; Landschützer et al., 2016; Rödenbeck et al., 2022; Chau et al., 2022; Gloege et al., 2021; Zeng et al., 2014; Iida et al., 2015; Gregor and Gruber, 2021). The thick black line indicates the model ensemble mean, and the thick blue line shows the mean of the $pCO_2$ product ensemble. Thin dashed lines are from individual GOBMs and $pCO_2$ products. FESOM2.1–REcoM3 (magenta) shows the ocean carbon flux for the period of 1959–2021, whereas FESOM1.4–REcoM2 (yellow) covers the period from 1959–2019. Positive numbers indicate a flux into the ocean.

**Table 4.** FESOM2.1–REcoM3 DIC inventory for simulation A (in PgC) in 1994 and change in DIC inventory between 1800–1994 and 1994–2007 calculated from simulation A minus simulation B to account for model drift. Thus, the FESOM2.1–REcoM3 numbers encompass anthropogenic carbon cycle processes and the effect of climate change on the natural carbon cycle. Gruber et al. (2019) estimate the anthropogenic carbon inventory change, which is equivalent to simulation A minus simulation D (constant atmospheric $CO_2$ and variable climate). We have given the Gruber et al. (2019) anthropogenic plus the back-of-the-envelope natural carbon inventory changes in parenthesis, which is roughly comparable to simulation A minus simulation B (only available for the global ocean).

|  | Year | Global (PgC) | North (PgC) | Tropics (PgC) | South (PgC) |
|---|---|---|---|---|---|
| Total DIC inventory |  |  |  |  |  |
| FESOM2.1–REcoM3 (Sim A) | 1994 | 38 167.4 | 5259.8 | 21 108.1 | 11 799.5 |
| DIC inventory change |  |  |  |  |  |
| FESOM2.1–REcoM3 (Sim A minus Sim B) | 1800 to 1994 | 121 | 19.5 | 54.4 | 47.2 |
| FESOM2.1–REcoM3 (Sim A minus Sim D) | 1800 to 1994 | 119 | 22.3 | 52.5 | 44.3 |
| Sabine et al. (2004); Gruber et al. (2019) | 1800 to 1994 | $118 \pm 19\ (111 \pm 21)$ | 25.1 | 46.6 | 48.0 |
| FESOM2.1–REcoM3 (Sim A minus Sim B) | 1994 to 2007 | 29.9 | 5.4 | 12.6 | 12.0 |
| FESOM2.1–REcoM3 (Sim A minus Sim D) | 1994 to 2007 | 30.9 | 5.8 | 13.2 | 11.9 |
| Gruber et al. (2019) | 1994 to 2007 | $34 \pm 4\ (29 \pm 5)$ | 5.9 | 17.5 | 10.4 |

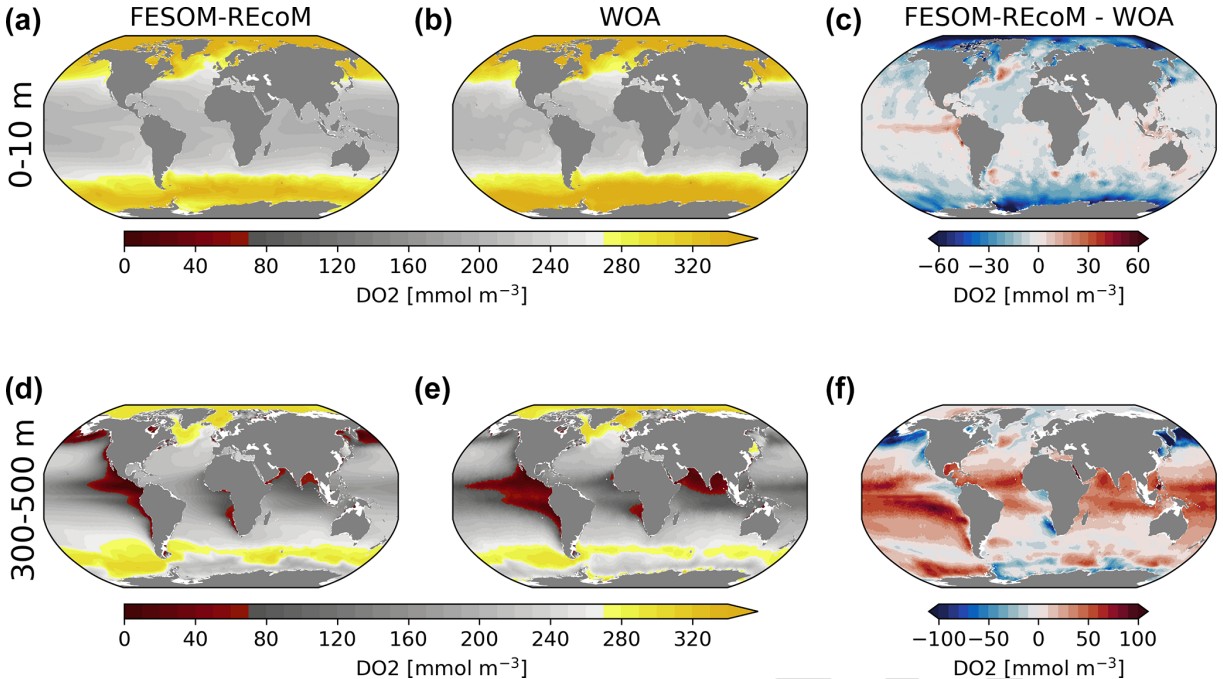

**Figure 20.** Maps of surface (0–10 m; **a–c**) and intermediate depth (300–500 m; **d–f**) concentration of simulated FESOM2.1–REcoM3 (simulation A) dissolved $O_2$ (mmol m$^{-3}$; **a, d**), World Ocean Atlas 2018 climatology of dissolved $O_2$ (**b, e**; Garcia et al., 2019b) and corresponding differences (**c, f**) over the time period from 2012–2021.

derestimated $O_2$ concentrations in the high-latitude surface ocean. At intermediate depth, the model generally overestimated the oxygen levels, especially in the Pacific Ocean and the subpolar Southern Ocean, with biases exceeding 100 mmol m$^{-3}$. Within the 100–600 m layer, FESOM2.1–REcoM3 performed remarkably well, with simulated values of about $160 \pm 105$ mmol m$^{-3}$, which is very close to the observations from WOA 2018 ($158 \pm 103$ mmol m$^3$). A previous model intercomparison study of oxygen concentrations within the 100–600 m layer (Cocco et al., 2013) showed that such performances are common, as all evaluated models fell within the error range of observations.

## 4 Conclusions and outlook

We have presented the new coupled ocean biogeochemistry model FESOM2.1–REcoM3. Building upon finite volumes for the ocean component improves the numerical efficiency and leads to higher numerical throughput of the coupled model (Danilov et al., 2017). Furthermore, the biogeochemistry component was extended to incorporate state-of-the-art carbonate chemistry routines, a second zooplankton and detritus group, and simulate the cycling of oxygen in the ocean. In its present configuration, the overall realism of FESOM2.1–REcoM3 in simulating the observed mean biogeochemical state is comparable to that of most GOBMs, while being among the more realistic models for estimating

the global ocean anthropogenic carbon uptake. There are still a number of model shortcomings, such as a lower simulated NPP and regional misfit between the annual mean $CO_2$ flux of the model simulation and the $pCO_2$-based data product that will be addressed in the future.

This model set-up provides the basis for further model development, e.g. CE11 the inclusion of coccolithophores as an additional phytoplankton functional type and the sensitivity of phytoplankton growth to rising $CO_2$ (Seifert et al., 2022) and the separation of the generic small zooplankton group into micro- and mesozooplankton that reduces model biases in nutrient fields, increases net primary production, and better captures the top-down control on phytoplankton bloom phenology (Karakuş et al., 2022). We further plan to incorporate more detailed iron biogeochemistry, as developed in REcoM coupled to MITgcm (e.g. Ye et al., 2020), and the explicit representation of the effects of viscosity and ballasting on the particle sinking speed, as well as oxygen-dependent remineralization, following Cram et al. (2018) to address knowledge gaps in carbon export and transfer to depth (Henson et al., 2022). Other on-going work addresses the role of rivers for carbon and nutrient transport into the ocean and the remineralization timescale of this river-derived organic material (Aumont et al., 2001; Lacroix et al., 2020; Regnier et al., 2022) and thus tackles a major uncertainty in the ocean carbon cycle and that complicates the comparison of ocean carbon sink estimates based on $pCO_2$ products and ocean biogeochemistry models (e.g. Hauck et al., 2020).

## Appendix A: Equations

This appendix provides an overview of the underlying model equations and lists all biogeochemical variables of FESOM2.1–REcoM3. Changes in state variables in REcoM3 are controlled by biological and chemical processes, in addition to the changes induced by ocean circulation, mixing, diffusion, and advection computed by FESOM2.1. While some variables exchange across the ocean surface and/or the sea floor, others, like dead organic matter (detritus), sink through the water column. The concentration change for a state variable $S$ is formulated as follows:

$$\frac{\partial S}{\partial t} = -\boldsymbol{U} \cdot \nabla S + \nabla \cdot (\kappa \cdot \nabla S) + \mathrm{SMS}(S), \tag{A1}$$

where $S$ is the volumetric concentration of a state variable, $\boldsymbol{U}$ TS10 is the 3-dimensional advection velocity, and $\kappa$ is the diffusivity. The term $\mathrm{SMS}(S)$ represents the biogeochemical sources minus sinks. The slow-sinking detritus class is assumed to sink with a velocity, which increases linearly with depth as a first-order description of the shift to larger and faster-sinking particles with increasing depth (Kriest and Oschlies, 2008). A constant sinking rate is applied to the fast-sinking detritus class. REcoM3 has 28 oceanic and four explicit benthic state variables (Tables A1 and A2).

## A1  Sources minus sinks

### A1.1  Nutrients

### Dissolved inorganic nitrate (DIN)

The simulated DIN conceptually represents the concentrations of nitrate, nitrite, and ammonia, while in practice only nitrate is considered. The concentration of DIN in the water column rises when dissolved organic nitrogen (DON) is remineralized and diminishes as a consequence of assimilation by small phytoplankton and diatoms as follows:

$$\mathrm{SMS}(\mathrm{DIN}) = \underbrace{\rho_{\mathrm{DON}} \cdot f_{\mathrm{T}} \cdot \mathrm{DON}}_{\text{DON remineralization}}$$
$$- \underbrace{V_{\mathrm{small}}^{\mathrm{N}} \cdot \mathrm{PhyC}_{\mathrm{small}}}_{\text{N assimilation, small phytoplankton}} - \underbrace{V_{\mathrm{dia}}^{\mathrm{N}} \cdot \mathrm{PhyC}_{\mathrm{dia}}}_{\text{N assimilation, diatoms}}. \tag{A2}$$

The state variables DON, $\mathrm{PhyC}_{\mathrm{small}}$, and $\mathrm{PhyC}_{\mathrm{dia}}$ are listed in Table A1. The value of the remineralization rate constant ($\rho_{\mathrm{DON}}$) is given in Table A8. The temperature dependency of remineralization ($f_{\mathrm{T}}$) is calculated in Eq. (A43). See Sect. A3.4 for details on the carbon-specific nitrogen assimilation rates $V_{\mathrm{small}}^{\mathrm{N}}$ and $V_{\mathrm{dia}}^{\mathrm{N}}$ (Table A5).

### Dissolved silicic acid (DSi)

Silicon assimilation (Si assimilation) increases when biogenic silica from one of the two detritus classes dissolves.

$$\mathrm{SMS}(\mathrm{DSi}) = \underbrace{\rho_{\mathrm{Si}}^{\mathrm{T}} \cdot \mathrm{DetSi}}_{\text{Remineralization, slow-sinking detritus}}$$
$$+ \underbrace{\rho_{\mathrm{Si}}^{\mathrm{T}} \cdot \mathrm{DetZ2Si}}_{\text{Remineralization, fast-sinking detritus}} - \underbrace{V^{\mathrm{Si}} \cdot \mathrm{PhyC}_{\mathrm{dia}}}_{\text{Si assimilation, diatoms}} \tag{A3}$$

The state variables $\mathrm{PhyC}_{\mathrm{dia}}$, DetSi, and DetZ2Si are listed in Table A1. The temperature-dependent remineralization rate of silicon ($\rho_{\mathrm{Si}}^{\mathrm{T}}$) and the carbon-specific Si assimilation rate ($V^{\mathrm{Si}}$) are calculated in Eqs. (A45) and (A51), respectively (Table A5).

### Dissolved iron (DFe)

Excretion of phyto- and zooplankton and remineralization of detritus release iron with a fixed iron : nitrate ratio ($q^{\mathrm{Fe:N}}$). Unlike for nitrogen, which is released as dissolved organic nitrogen and needs to be remineralized further to become available as nutrient again, the released iron is put directly into the dissolved pool iron, basically assuming that all dissolved iron is ultimately bio-available. Iron assimilation (again assumed to be proportional to nitrogen assimilation; hereafter N assimilation) by both phytoplankton classes lower the level of dissolved iron. In addition, free inorganic iron Fe$'$ is scavenged onto sinking particles, with a rate that is proportional to particle concentration. We take detrital carbon as a proxy for the mass of sinking particles.

$$\mathrm{SMS}(\mathrm{DFe}) = q^{\mathrm{Fe:N}} \cdot (\underbrace{\epsilon_{\mathrm{phy}}^{\mathrm{N}} \cdot f_{\mathrm{lim, small}}^{\mathrm{N:Cmax}} \cdot \mathrm{PhyN}_{\mathrm{small}}}_{\text{Excretion, small phytoplankton}} + \underbrace{\epsilon_{\mathrm{phy}}^{\mathrm{N}} \cdot f_{\mathrm{lim, dia}}^{\mathrm{N:Cmax}} \cdot \mathrm{PhyN}_{\mathrm{dia}}}_{\text{Excretion, diatoms}}$$
$$+ \underbrace{\rho_{\mathrm{DetN}} \cdot f_{\mathrm{T}} \cdot \mathrm{DetN}}_{\text{Remineralization, slow-sinking detritus}} + \underbrace{\rho_{\mathrm{DetN}} \cdot f_{\mathrm{T}} \cdot \mathrm{DetZ2N}}_{\text{Remineralization, fast-sinking detritus}}$$
$$+ \underbrace{\epsilon_{\mathrm{zoo}}^{\mathrm{N}} \cdot \mathrm{ZooN}}_{\text{Excretion, small zooplankton}} + \underbrace{\epsilon_{\mathrm{zoo2}}^{\mathrm{N}} \cdot \mathrm{Zoo2N}}_{\text{Excretion, macrozooplankton}}$$
$$- \underbrace{V_{\mathrm{small}}^{\mathrm{N}} \cdot \mathrm{PhyC}_{\mathrm{small}}}_{\text{N assimilation, small phytoplankton}} - \underbrace{V_{\mathrm{dia}}^{\mathrm{N}} \cdot \mathrm{PhyC}_{\mathrm{dia}}}_{\text{N assimilation, diatom}})$$
$$- \underbrace{\kappa_{\mathrm{Fe}} \cdot \mathrm{DetC} \cdot \mathrm{Fe}'}_{\text{Scavenging, slow-sinking detritus}} - \underbrace{\kappa_{\mathrm{Fe}} \cdot \mathrm{DetZ2C} \cdot \mathrm{Fe}'}_{\text{Scavenging, fast-sinking detritus}}$$
$$\tag{A4}$$

The state variables $\mathrm{PhyC}_{\mathrm{small}}$, $\mathrm{PhyC}_{\mathrm{dia}}$, $\mathrm{PhyN}_{\mathrm{small}}$, $\mathrm{PhyN}_{\mathrm{dia}}$, DetC, DetN, DetZ2C, DetZ2N, ZooN, and Zoo2N are listed in Table A1. The intracellular Fe : N ratio ($q^{\mathrm{Fe:N}}$) and scavenging rate of iron ($\kappa_{\mathrm{Fe}}$) are given in Table A4. Excretion rates ($\epsilon_{\mathrm{phy}}^{\mathrm{N}}$, $\epsilon_{\mathrm{zoo}}^{\mathrm{N}}$, and $\epsilon_{\mathrm{zoo2}}^{\mathrm{N}}$) and the degradation rate for detritus N ($\rho_{\mathrm{DetN}}$) are listed in Table A8. The temperature dependency ($f_{\mathrm{T}}$) is calculated in Eq. (A43). The limitation of intracellular nitrogen ($f_{\mathrm{lim, small}}^{\mathrm{N:Cmax}}$; $f_{\mathrm{lim, dia}}^{\mathrm{N:Cmax}}$) is described in Eq. (A55). Scavenging is calculated following Parekh et al. (2004). The total concentration of dissolved iron (Fe$_{\mathrm{T}}$) is separated into free iron (Fe$'$) and iron complexed with organic

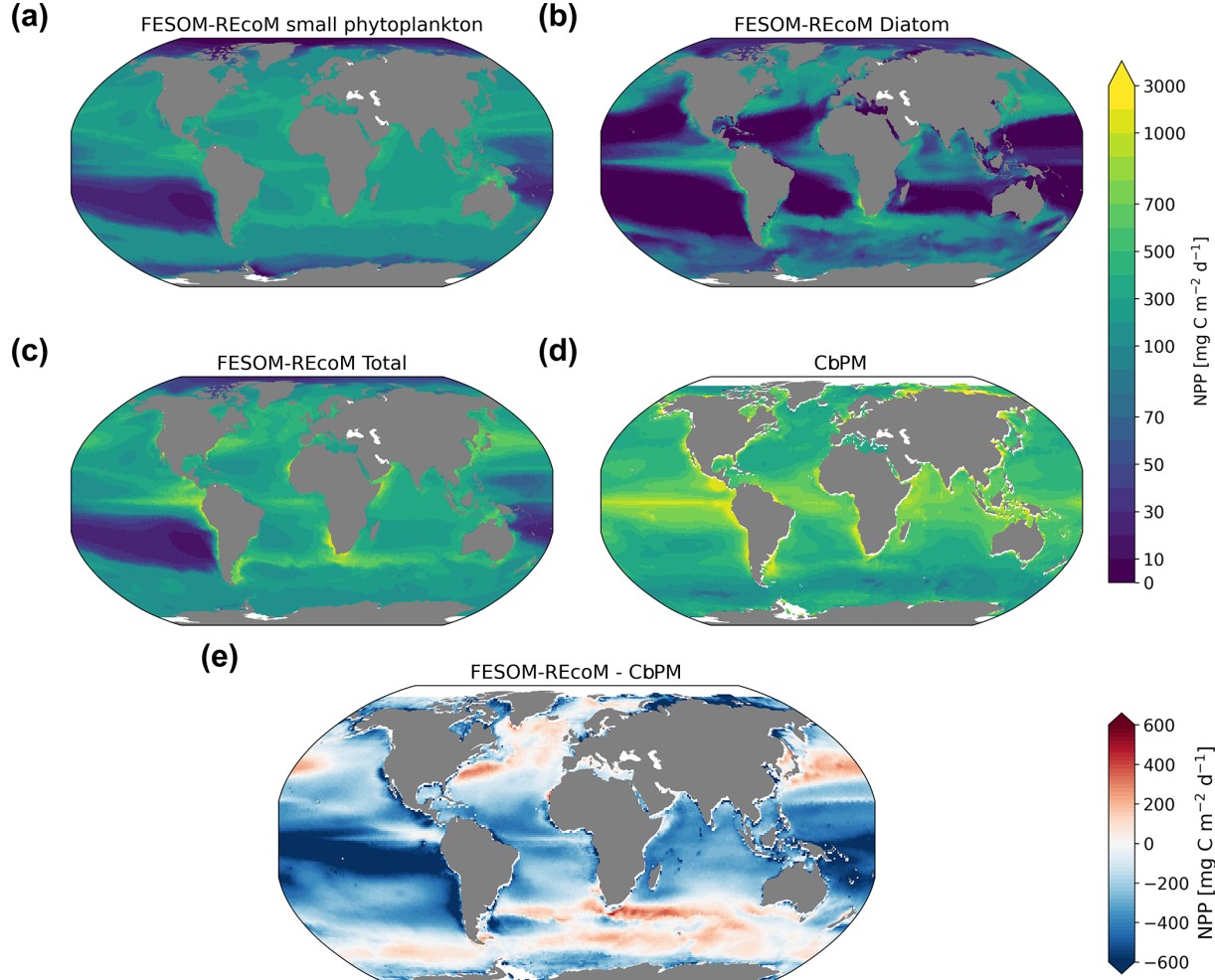

**Figure A1.** Maps of simulated FESOM2.1–REcoM3 (simulation A) vertically integrated net primary production ($mgC\,m^{-2}\,d^{-1}$) of small phytoplankton **(a)**, diatoms **(b)** and the sum of both phytoplankton groups **(c)**. The satellite-based carbon-based productivity model (CbPM) is shown (**d**; Westberry et al., 2008), with corresponding differences between FESOM2.1–REcoM3 and VGPM **(e)**. All fields are averaged over the time period from 2012 to 2021.

ligands (Fe$_L$), which is not scavenged. Complexation reactions are fast (Tagliabue and Völker, 2011), so we assume instantaneous equilibrium between free iron and free ligand (L'), which is computed using a constant $K_{Fe_L} = \frac{[Fe']\cdot[L']}{[Fe_L]}$, by solving the following:

$$Fe_T = Fe' + Fe_L \qquad L_T = Fe_L + L'. \tag{A5}$$

For simplicity, we assume here a constant total ligand concentration L$_T$, unlike in Völker and Tagliabue (2015). Variable ligand concentration, like in Misumi et al. (2011) or Völker and Tagliabue (2015), or variable ligand binding strength, like in Ye et al. (2020), will be explored in the future. The values for $K_{Fe_L}$ and L$_T$ are listed in Table A4.

## A1.2 Carbon cycle

### Dissolved inorganic carbon (DIC)

DIC concentration increases with respiration of phyto- and zooplankton, remineralization of semi-labile dissolved organic carbon, dissolution of calcitic detritus, and dissolution of CaCO$_3$ in zooplankton guts. Loss terms are carbon fixation by primary producers and the formation of calcium carbonate. In addition, the sea–air flux of CO$_2$ leads to an exchange of carbon with the atmosphere, depending on the partial pressure difference in the CO$_2$ between ocean and atmosphere. This exchange is treated separately as a boundary condition. The partial pressure of surface ocean CO$_2$ is com-

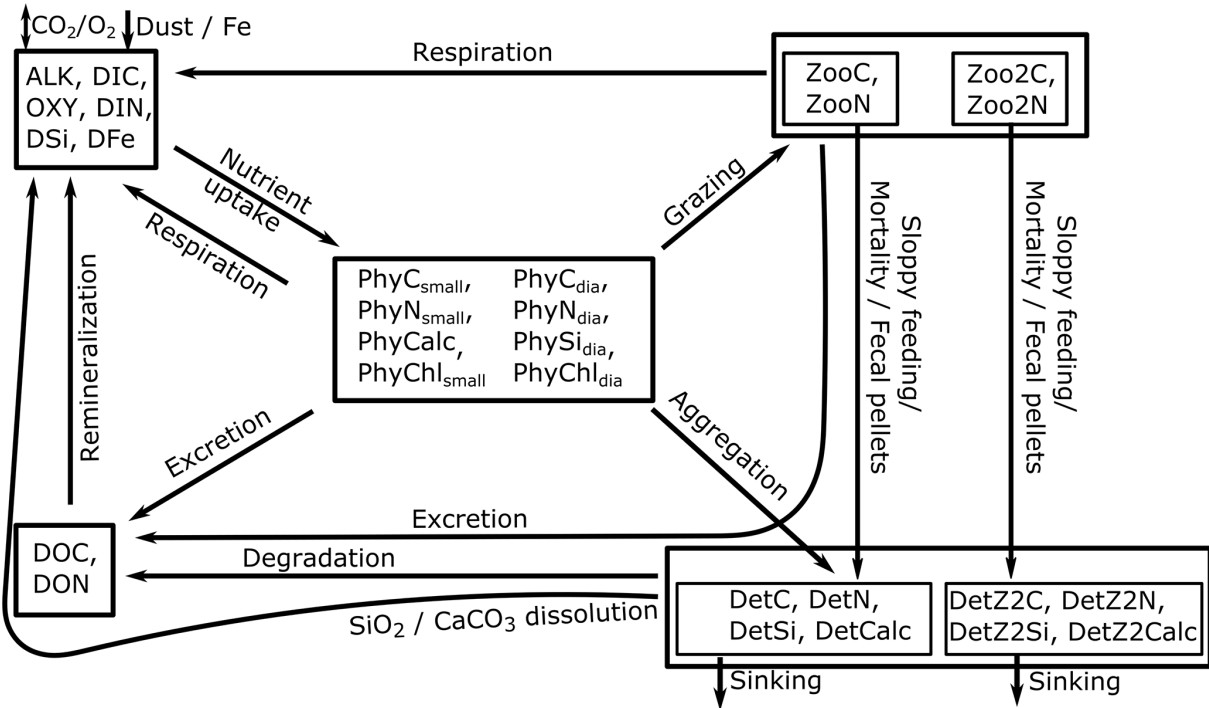

**Figure A2.** Conceptual diagram of the ocean biogeochemical model REcoM3. The 28 tracers can be grouped (indicated by boxes) into dissolved nutrients, carbonate system parameters and oxygen (upper left), phytoplankton functional types (centre), zooplankton functional types (upper right), two detritus classes (lower right), and dissolved organic material (lower left). Source and sink terms are depicted by arrows. For reasons of diagrammatic clarity, connections of dissolved oxygen (Oxy) to other state variables are omitted here. Similarly, the release of alkalinity, dissolved inorganic nutrients, and organic matter from the sediment are not shown.

puted using the mocsy 2.0 routines (Orr and Epitalon, 2015).

$$
\begin{aligned}
\text{SMS(DIC)} = & \underbrace{(r_{\text{small}} - P_{\text{small}}) \cdot \text{PhyC}_{\text{small}}}_{\text{Net respiration, small phytoplankton}} + \underbrace{(r_{\text{dia}} - P_{\text{dia}}) \cdot \text{PhyC}_{\text{dia}}}_{\text{Net respiration, diatom}} \\
& + \underbrace{\rho_{\text{DOC}} \cdot f_{\text{T}} \cdot \text{DOC}}_{\text{Remineralization of DOC}} + \underbrace{r_{\text{zoo}} \cdot \text{ZooC}}_{\text{Respiration, small zoo.}} \\
& + \underbrace{r_{\text{zoo2}} \cdot \text{Zoo2C}}_{\text{Respiration, macrozoo.}} + \underbrace{\text{Diss}_{\text{calc}} \cdot \text{DetCalc}}_{\text{Calcite dissolution, slow-sinking detritus}} \\
& + \underbrace{G_{\text{small}}^{\text{zoo}} \cdot q_{\text{small}}^{\text{CaCO}_3:\text{N}} \cdot \text{Diss}_{\text{calc\_guts}}}_{\text{CaCO}_3 \text{ dissolution in guts, small zoo.}} - \underbrace{\psi \cdot P_{\text{small}} \cdot \text{PhyC}_{\text{small}}}_{\text{Calcification}} \\
& + \underbrace{\text{Diss}_{\text{calc2}} \cdot \text{DetZ2Calc}}_{\text{Calcite dissolution, fast-sinking detritus}} \\
& + \underbrace{G_{\text{small}}^{\text{zoo2}} \cdot q_{\text{small}}^{\text{CaCO}_3:\text{N}} \cdot \text{Diss}_{\text{calc\_guts}}}_{\text{CaCO}_3 \text{ dissolution in guts, macrozoo.}}
\end{aligned} \tag{A6}
$$

The state variables PhyC$_{\text{small}}$, PhyC$_{\text{dia}}$, DOC, ZooC, Zoo2C, DetCalc, and DetZ2Calc are listed in Table A1. Respiration rate constants of small phytoplankton ($r_{\text{small}}$), diatoms ($r_{\text{dia}}$), and zooplankton groups ($r_{\text{zoo}}$ and $r_{\text{zoo2}}$) are computed in Sects. A3.2 and A4.1, respectively. Photosynthesis terms ($P_{\text{small}}$ and $P_{\text{dia}}$) are calculated in Eq. (A46). The remineralization rate constant ($\rho_{\text{DOC}}$) is listed in Table A8, and the temperature dependency ($f_{\text{T}}$) is given in Eq. (A43). Calcite dissolution by detritus (Diss$_{\text{calc}}$; Diss$_{\text{calc2}}$) is calculated in Eq. (A38). The constant for the dissolution of calcium

carbonate in zooplankton guts (Diss$_{\text{calc\_guts}}$) is listed in Table A5. $G_{\text{small}}^{\text{zoo}}$ and $G_{\text{small}}^{\text{zoo2}}$ are grazing terms and explained in Sect. A4.2. The value of the calcite production ratio ($\psi$) is given in Table A3.

**Total alkalinity (Alk)**

The balance of alkalinity is affected by primary production, remineralization of dissolved organic matter, dissolution of calcitic detritus, and dissolution of CaCO$_3$ in zooplankton guts. Alkalinity increases when nitrogen is assimilated and when CaCO$_3$ is dissolved (Wolf-Gladrow et al., 2007). Simultaneously, it is reduced by the calcification and remineralization of dissolved organic nitrogen. The effect of phosphate assimilation and remineralization onto alkalinity is taken into account, assuming a constant N : P Redfield ratio (16 : 1).

$$
\text{SMS(Alk)} = (1 + 1/16) \cdot \underbrace{V_{\text{small}}^{\text{N}} \cdot \text{PhyC}_{\text{small}}}_{\text{N assimilation, small phytoplankton}}
$$

$$
+ (1 + 1/16) \cdot \underbrace{V_{\text{dia}}^{\text{N}} \cdot \text{PhyC}_{\text{dia}}}_{\text{N assimilation, diatom}}
$$

$$
- (1 + 1/16) \cdot \underbrace{\rho_{\text{DON}} \cdot f_{\text{T}} \cdot \text{DON}}_{\text{Remineralization of DON}}
$$

$$
- 2 \cdot \underbrace{\psi \cdot P_{\text{small}} \cdot \text{PhyC}_{\text{small}}}_{\text{Calcification}} \tag{A7}
$$

$$
+ 2 \cdot \underbrace{\text{Diss}_{\text{calc}} \cdot \text{DetCalc}}_{\text{Calcite dissolution, slow-sinking detritus}}
$$

$$
+ 2 \cdot \underbrace{G_{\text{small}}^{\text{zoo}} \cdot q_{\text{small}}^{\text{CaCO}_3:\text{N}} \cdot \text{Diss}_{\text{calc\_guts}}}_{\text{CaCO}_3 \text{ dissolution in guts, small zoo.}}
$$

$$
+ 2 \cdot \underbrace{\text{Diss}_{\text{calc2}} \cdot \text{DetZ2Calc}}_{\text{Calcite dissolution, fast-sinking detritus}}
$$

$$
+ 2 \cdot \underbrace{G_{\text{small}}^{\text{zoo2}} \cdot q_{\text{small}}^{\text{CaCO}_3:\text{N}} \cdot \text{Diss}_{\text{calc\_guts}}}_{\text{CaCO}_3 \text{ dissolution in guts, macrozoo.}} \tag{A8}
$$

The **TS11** state variables PhyC$_{\text{small}}$, PhyC$_{\text{dia}}$, DON, DetCalc, and DetZ2Calc are listed in Table A1. The N assimilation ($V_{\text{small}}^{\text{N}}$ and $V_{\text{dia}}^{\text{N}}$) is calculated in Sect. A3.4. The remineralization rate constant ($\rho_{\text{DON}}$) is given in Table A8. The temperature dependency ($f_{\text{T}}$) is calculated in Eq. (A43). The value of the calcite production ratio ($\psi$) is given in Table A3. The photosynthesis term ($P_{\text{small}}$) is calculated in Eq. (A46). The calcite dissolution by detritus (Diss$_{\text{calc}}$, Diss$_{\text{calc2}}$) is calculated in Eq. (A38). Dissolution of calcium carbonate in guts (Diss$_{\text{calc\_guts}}$) is listed in Table A5. $G_{\text{small}}^{\text{zoo}}$ and $G_{\text{small}}^{\text{zoo2}}$ are grazing terms and explained in Sect. A4.2.

### A1.3 Phytoplankton

#### Nitrogen

The phytoplankton nitrogen pools increase through N assimilation. The assimilation process is assumed to be proportional to carbon biomass, with a carbon-specific uptake rate that depends on the C : N ratio of phytoplankton and the external DIN concentration (Geider et al., 1998). Excretion of biogenic nitrogen to semi-labile DON drains the pool. At a high intracellular C : N ratio, the excretion is downregulated. Aggregation and grazing by the two zooplankton groups transfer nitrogen to the zooplankton and detritus pools.

$$
\text{SMS(PhyN}_{\text{small}}) = \underbrace{V_{\text{small}}^{\text{N}} \cdot \text{PhyC}_{\text{small}}}_{\text{N assimilation}} \tag{A9}
$$

$$
- \underbrace{\epsilon_{\text{phy}}^{\text{N}} \cdot f_{\text{lim, small}}^{\text{N:Cmax}} \cdot \text{PhyN}_{\text{small}}}_{\text{DON excretion}} - \underbrace{\text{Agg} \cdot \text{PhyN}_{\text{small}}}_{\text{Aggregation loss}}
$$

$$
- \underbrace{G_{\text{small}}^{\text{zoo}}}_{\text{Grazing loss by small zoo.}} - \underbrace{G_{\text{small}}^{\text{zoo2}}}_{\text{Grazing loss by macrozoo.}} \tag{A10}
$$

$$
\text{SMS(PhyN}_{\text{dia}}) = \underbrace{V_{\text{dia}}^{\text{N}} \cdot \text{PhyC}_{\text{dia}}}_{\text{N assimilation}} \tag{A11}
$$

$$
- \underbrace{\epsilon_{\text{phy}}^{\text{N}} \cdot f_{\text{lim, dia}}^{\text{N:Cmax}} \cdot \text{PhyN}_{\text{dia}}}_{\text{DON excretion}} - \underbrace{\text{Agg} \cdot \text{PhyN}_{\text{dia}}}_{\text{Aggregation loss}}
$$

$$
- \underbrace{G_{\text{dia}}^{\text{zoo}}}_{\text{Grazing loss by small zoo.}} - \underbrace{G_{\text{dia}}^{\text{zoo2}}}_{\text{Grazing loss by macrozoo.}} \tag{A12}
$$

The **TS12** state variables PhyC$_{\text{small}}$, PhyN$_{\text{small}}$, PhyC$_{\text{dia}}$, and PhyN$_{\text{dia}}$ are listed in Table A1. The N assimilation ($V_{\text{small}}^{\text{N}}$ and $V_{\text{dia}}^{\text{N}}$) is explained in Sect. A3.4. The constant excretion rate constant ($\epsilon_{\text{phy}}^{\text{N}}$) is given in Table A8. When the C : N ratio of the cells becomes too high, excretion of DON is downregulated by the limiter function ($f_{\text{lim, small}}^{\text{N:Cmax}}$; $f_{\text{lim, dia}}^{\text{N:Cmax}}$) that is described in Eq. (A55). Phytoplankton aggregation (Agg) defines the transfer of nitrogen into the detritus pools, which depends quadratically on detritus and phytoplankton concentrations (Eq. A52). Grazing loss terms ($G_{\text{small}}^{\text{zoo}}$, $G_{\text{small}}^{\text{zoo2}}$, $G_{\text{dia}}^{\text{zoo}}$, and $G_{\text{dia}}^{\text{zoo2}}$) are explained in Sect. A4.2.

#### Carbon

The carbon biomass of small phytoplankton and diatoms increases as a result of carbon assimilation during photosynthesis. Loss terms include excretion of DOC, which is limited by the availability of proteins as in the nitrogen pool, respiration, aggregation, and grazing.

$$
\text{SMS(PhyC}_{\text{small}}) = \underbrace{(P_{\text{small}} - r_{\text{small}}) \cdot \text{PhyC}_{\text{small}}}_{\text{Net photosynthesis}}
$$

$$
- \underbrace{\text{Agg} \cdot \text{PhyC}_{\text{small}}}_{\text{Aggregation loss}} - \underbrace{\epsilon_{\text{phy}}^{\text{C}} \cdot f_{\text{lim, small}}^{\text{N:Cmax}} \cdot \text{PhyC}_{\text{small}}}_{\text{Excretion of DOC}}
$$

$$
- \underbrace{q_{\text{small}}^{\text{C:N}} \cdot G_{\text{small}}^{\text{zoo}}}_{\text{Grazing loss by small zoo.}} - \underbrace{q_{\text{small}}^{\text{C:N}} \cdot G_{\text{small}}^{\text{zoo2}}}_{\text{Grazing loss by macrozoo.}} \tag{A13}
$$

$$
\text{SMS(PhyC}_{\text{dia}}) = \underbrace{(P_{\text{dia}} - r_{\text{dia}}) \cdot \text{PhyC}_{\text{dia}}}_{\text{Net photosynthesis}}
$$

$$
- \underbrace{\text{Agg} \cdot \text{PhyC}_{\text{dia}}}_{\text{Aggregation loss}} - \underbrace{\epsilon_{\text{phy}}^{\text{C}} \cdot f_{\text{lim, dia}}^{\text{N:Cmax}} \cdot \text{PhyC}_{\text{dia}}}_{\text{Excretion of DOC}}
$$

$$
- \underbrace{q_{\text{dia}}^{\text{C:N}} \cdot G_{\text{dia}}^{\text{zoo}}}_{\text{Grazing loss by small zoo.}} - \underbrace{q_{\text{dia}}^{\text{C:N}} \cdot G_{\text{dia}}^{\text{zoo2}}}_{\text{Grazing loss by macrozoo.}} \tag{A14}
$$

The state variables PhyC$_{\text{small}}$ and PhyC$_{\text{dia}}$ are listed in Table A1. The photosynthesis terms ($P_{\text{small}}$ and $P_{\text{dia}}$) are calculated in Eq. (A46). The rates of respiration by small phytoplankton ($r_{\text{small}}$) and diatoms ($r_{\text{dia}}$) are explained in

Sect. A3.2. The constant for DOC excretion rate of phytoplankton ($\epsilon_{\text{phy}}^{\text{C}}$; Table A8) is downregulated by the limiter factor ($f_{\text{lim, small}}^{\text{N:Cmax}}$; $f_{\text{lim, dia}}^{\text{N:Cmax}}$) when the N : C ratio becomes too high (Eq. A55). Phytoplankton aggregation (Agg) is calculated in Eq. (A52). Grazing terms ($G_{\text{small}}^{\text{zoo}}$, $G_{\text{small}}^{\text{zoo2}}$, $G_{\text{dia}}^{\text{zoo}}$, and $G_{\text{dia}}^{\text{zoo2}}$) are explained in Sect. A4.2. $q^{\text{C:N}} = \text{PhyC}/\text{PhyN}$, is used to convert the grazing units from millimoles of N to millimoles of C.

### CaCO$_3$

The formation of biogenic calcium carbonate in our model is limited to coccolithophores only, which are assumed to form a constant fraction of the non-diatom phytoplankton. Formation of $CaCO_3$ by heterotrophs, such as foraminifera or pteropods is neglected. Biogenic $CaCO_3$ produced by coccolithophores is transformed into detritus $CaCO_3$ with all forms of organic carbon loss, including organic matter excretion, respiration, aggregation, and grazing. Calcifiers are assumed to comprise a certain fraction of the total small phytoplankton concentration, which is specified by the parameter $\psi$ (Table A3), thus tying the calcite production of calcifiers to the growth of small phytoplankton.

$$\text{SMS(PhyCalc)} = \underbrace{\psi \cdot P_{\text{small}} \cdot \text{PhyC}_{\text{small}}}_{\text{Calcification}} - \underbrace{r_{\text{small}} \cdot \text{PhyCalc}}_{\text{Respiration}}$$
$$- \underbrace{G_{\text{small}}^{\text{zoo}} \cdot q_{\text{small}}^{\text{CaCO}_3\text{:N}}}_{\text{Grazing loss, small zoo.}} - \underbrace{G_{\text{small}}^{\text{zoo2}} \cdot q_{\text{small}}^{\text{CaCO}_3\text{:N}}}_{\text{Grazing loss, macrozoo.}}$$
$$- \underbrace{\epsilon_{\text{phy}}^{\text{C}} \cdot f_{\text{lim, small}}^{\text{N:Cmax}} \cdot \text{PhyCalc}}_{\text{Excretion loss}} - \underbrace{\text{Agg} \cdot \text{PhyCalc}}_{\text{Aggregation loss}} \quad \text{(A15)}$$

The state variables PhyC$_{\text{small}}$ and PhyCalc are listed in Table A1. The value of the calcite production ratio ($\psi$) is given in Table A3. The constant excretion rate ($\epsilon_{\text{phy}}^{\text{C}}$; Table A8) is downregulated by the limiter factor $f_{\text{lim, small}}^{\text{N:Cmax}}$ (Eq. A55) when the N : C ratio becomes too high. Photosynthesis ($P_{\text{small}}$), respiration ($r_{\text{small}}$), and the aggregation of phytoplankton (Agg) rates are calculated in Eqs. (A46), (A48), and (A52), respectively. Grazing terms ($G_{\text{small}}^{\text{zoo}}$ and $G_{\text{small}}^{\text{zoo2}}$) are explained in Sect. A4.2. $q_{\text{small}}^{\text{CaCO}_3\text{:N}} = \text{PhyCalc}/\text{PhyN}_{\text{small}}$ is used to convert the grazing units from millimoles of N to millimoles of $CaCO_3$.

### Diatom silicon

The silica frustule of diatoms is built through Si assimilation, which we assume to be carbon specific and regulated by cellular quotas (see below). Any decrease in N biomass through excretion, grazing, or aggregation leads to a corresponding transfer of silica to the detritus silica pool.

$$\text{SMS(PhySi)} = \underbrace{V^{\text{Si}} \cdot \text{PhyC}_{\text{dia}}}_{\text{Diatom Si assimilation}}$$
$$- \underbrace{\epsilon_{\text{phy}}^{\text{N}} \cdot f_{\text{lim, dia}}^{\text{N:Cmax}} \cdot \text{PhySi}_{\text{dia}}}_{\text{Excretion to detritus}} - \underbrace{\text{Agg} \cdot \text{PhySi}_{\text{dia}}}_{\text{Aggregation loss}}$$
$$- \underbrace{G_{\text{dia}}^{\text{zoo}} \cdot q^{\text{Si:N}}}_{\text{Grazing loss, small zoo.}} - \underbrace{G_{\text{dia}}^{\text{zoo2}} \cdot q^{\text{Si:N}}}_{\text{Grazing loss, macrozoo.}}$$
$$\text{(A16)}$$

The state variables PhyC$_{\text{dia}}$ and PhySi$_{\text{dia}}$ are described in Table A1. Si assimilation ($V^{\text{Si}}$) and aggregation rates (Agg) are calculated in Eqs. (A51) and (A52), respectively. The constant excretion rate ($\epsilon_{\text{phy}}^{\text{N}}$; Table A8) is downregulated by the limiter factor $f_{\text{lim, dia}}^{\text{N:Cmax}}$ (Eq. A55) when the N : C ratio becomes too high. Grazing terms ($G_{\text{dia}}^{\text{zoo}}$ and $G_{\text{dia}}^{\text{zoo2}}$) are explained in Sect. A4.2. The intracellular ratio between diatom silicon and nitrate is defined as $q^{\text{Si:N}} = \text{PhySi}_{\text{dia}}/\text{PhyN}_{\text{dia}}$.

### Chlorophyll $a$

Chlorophyll $a$ synthesis is structured as a function of irradiance and of N assimilation, following Geider et al. (1998). Chlorophyll $a$ is degraded at a light-dependent rate (see Álvarez et al., 2018) and lost via aggregation and grazing. The grazing losses in terms of nitrogen biomass are converted to chlorophyll loss using the intracellular Chl : N ratio.

$$\text{SMS(PhyChl}_{\text{small}}) = \underbrace{S_{\text{small}}^{\text{chl}} \cdot \text{PhyC}_{\text{small}}}_{\text{Chlorophyll } a \text{ synthesis}} \quad \text{(A17)}$$
$$- \underbrace{G_{\text{small}}^{\text{zoo}} \cdot q_{\text{small}}^{\text{Chl:N}}}_{\text{Grazing loss, small zoo.}} - \underbrace{G_{\text{small}}^{\text{zoo2}} \cdot q_{\text{small}}^{\text{Chl:N}}}_{\text{Grazing loss, macrozoo.}}$$
$$- \underbrace{\text{deg}_{\text{small}}^{\text{chl}} \cdot \text{PhyChl}_{\text{small}}}_{\text{Degradation loss}} - \underbrace{\text{Agg} \cdot \text{PhyChl}_{\text{small}}}_{\text{Aggregation loss}} \quad \text{(A18)}$$

$$\text{SMS(PhyChl}_{\text{dia}}) = \underbrace{S_{\text{dia}}^{\text{chl}} \cdot \text{PhyC}_{\text{dia}}}_{\text{Chlorophyll } a \text{ synthesis}} \quad \text{(A19)}$$
$$- \underbrace{G_{\text{dia}}^{\text{zoo}} \cdot q_{\text{dia}}^{\text{Chl:N}}}_{\text{Grazing loss, small zoo.}} - \underbrace{G_{\text{dia}}^{\text{zoo2}} \cdot q_{\text{dia}}^{\text{Chl:N}}}_{\text{Grazing loss, macrozoo.}}$$
$$- \underbrace{\text{deg}_{\text{dia}}^{\text{chl}} \cdot \text{PhyChl}_{\text{dia}}}_{\text{Degradation loss}} - \underbrace{\text{Agg} \cdot \text{PhyChl}_{\text{dia}}}_{\text{Aggregation loss}}$$
$$\text{(A20)}$$

The state variables PhyC$_{\text{small}}$, PhyC$_{\text{dia}}$, PhyChl$_{\text{small}}$, and PhyChl$_{\text{dia}}$ are listed in Table A1. The chlorophyll $a$ synthesis ($S_{\text{small}}^{\text{chl}}$; $S_{\text{dia}}^{\text{chl}}$) and the aggregation (Agg) terms are calculated in Eqs. (A49) and (A52), respectively. The degradation parameters ($\text{deg}_{\text{small}}^{\text{chl}}$; $\text{deg}_{\text{dia}}^{\text{chl}}$) are given in Table A8. Grazing terms ($G_{\text{small}}^{\text{zoo}}$, $G_{\text{small}}^{\text{zoo2}}$, $G_{\text{dia}}^{\text{zoo}}$, and $G_{\text{dia}}^{\text{zoo2}}$) are explained in Sect. A4.2. The conversion factor from millimoles of N to milligrams of Chl $a$ is defined as $q^{\text{Chl:N}} = \text{PhyChl}/\text{PhyN}$.

## A1.4  Zooplankton

### Nitrogen

Both zooplankton classes increase their nitrogen biomass via grazing on phytoplankton and detritus, while mortality and excretion of DON reduce it. Macrozooplankton further feeds on small zooplankton and releases nitrogen via fecal pellet production.

$$\text{SMS(ZooN)} = \underbrace{\gamma_{\text{zoo}} \cdot G_{\text{tot}}^{\text{zoo}}}_{\text{Grazing}} - \underbrace{G_{\text{zoo}}}_{\text{Grazing loss, macrozoo.}}$$

$$- \underbrace{m_{\text{zoo}} \cdot \text{ZooN}^2}_{\text{Mortality}} - \underbrace{\epsilon_{\text{zoo}}^{\text{N}} \cdot \text{ZooN}}_{\text{Excretion of DON}} \qquad (A21)$$

$$\text{SMS(Zoo2N)} = \underbrace{\gamma_{\text{zoo2}} \cdot G_{\text{tot}}^{\text{zoo2}}}_{\text{Grazing}} - \underbrace{m_{\text{zoo2}} \cdot \text{Zoo2N}^2}_{\text{Mortality}}$$

$$- \underbrace{\epsilon_{\text{zoo2}}^{\text{N}} \cdot \text{Zoo2N}}_{\text{Excretion of DON}} - \underbrace{f_{\text{n}} \cdot G_{\text{tot}}^{\text{zoo2}}}_{\text{Fecal pellet}} \qquad (A22)$$

The state variables ZooN and Zoo2N are listed in Table A1. Only a fraction of the grazed phytoplankton ($\gamma_{\text{zoo}}$ and $\gamma_{\text{zoo2}}$; Table A3) enters the zooplankton biomass. The rest is transferred to detritus due to sloppy feeding. The grazing terms ($G_{\text{tot}}^{\text{zoo}}$ and $G_{\text{tot}}^{\text{zoo2}}$) are calculated in Sect. A4.2. The mortality parameter ($m_{\text{zoo}}$ and $m_{\text{zoo2}}$) and fecal pellet production rate constant ($f_{\text{n}}$) are listed in Table A3. The DON excretion terms ($\epsilon_{\text{zoo}}^{\text{N}}$ and $\epsilon_{\text{zoo2}}^{\text{N}}$) are given in Table A8.

### Carbon

The zooplankton carbon biomass increases with carbon uptake via grazing and decreases through carbon losses through mortality, respiration, and carbon excretion to the semi-labile DOC pool. Macrozooplankton further gains carbon by grazing on small zooplankton and loses it via fecal pellet production.

$$\text{SMS(ZooC)} = \underbrace{\gamma_{\text{zoo}} \cdot (G_{\text{small}}^{\text{zoo}} \cdot q_{\text{small}}^{\text{C:N}} + G_{\text{dia}}^{\text{zoo}} \cdot q_{\text{dia}}^{\text{C:N}})}_{\text{Grazing on phytoplankton}}$$

$$+ \underbrace{\gamma_{\text{zoo}} \cdot (G_{\text{det}}^{\text{zoo}} \cdot q_{\text{det}}^{\text{C:N}} + G_{\text{detZ2}}^{\text{zoo}} \cdot q_{\text{detZ2}}^{\text{C:N}})}_{\text{Grazing on detritus}}$$

$$- \underbrace{G_{\text{zoo}} \cdot q_{\text{zoo}}^{\text{C:N}}}_{\text{Grazing loss by macrozoo.}} \qquad (A23)$$

$$- \underbrace{m_{\text{zoo}} \cdot \text{ZooN}^2 \cdot q_{\text{zoo}}^{\text{C:N}}}_{\text{Zooplankton mortality}} - \underbrace{r_{\text{zoo}} \cdot \text{ZooC}}_{\text{Respiration loss}}$$

$$- \underbrace{\epsilon_{\text{zoo}}^{\text{C}} \cdot \text{ZooC}}_{\text{Excretion of DOC}} \qquad (A24)$$

$$\text{SMS(Zoo2C)} = \underbrace{\gamma_{\text{zoo2}} \cdot (G_{\text{small}}^{\text{zoo2}} \cdot q_{\text{small}}^{\text{C:N}} + G_{\text{dia}}^{\text{zoo2}} \cdot q_{\text{dia}}^{\text{C:N}})}_{\text{Grazing on phytoplankton}}$$

$$+ \underbrace{\gamma_{\text{zoo2}} \cdot (G_{\text{det}}^{\text{zoo2}} \cdot q_{\text{det}}^{\text{C:N}} + G_{\text{detZ2}}^{\text{zoo2}} \cdot q_{\text{detZ2}}^{\text{C:N}})}_{\text{Grazing on detritus}}$$

$$+ \underbrace{\gamma_{\text{zoo2}} \cdot (G_{\text{zoo}} \cdot q_{\text{zoo}}^{\text{C:N}})}_{\text{Grazing on small zoo.}} \qquad (A25)$$

$$- \underbrace{m_{\text{zoo2}} \cdot \text{Zoo2N}^2 \cdot q_{\text{zoo2}}^{\text{C:N}}}_{\text{Zooplankton mortality}} - \underbrace{r_{\text{zoo2}} \cdot \text{Zoo2C}}_{\text{Respiration loss}}$$

$$- \underbrace{\epsilon_{\text{zoo2}}^{\text{C}} \cdot \text{Zoo2C}}_{\text{Excretion of DOC}} - \underbrace{f_{\text{c}} \cdot G_{\text{cflux}}}_{\text{Fecal pellet}} \qquad (A26)$$

The TS13 state variables ZooN, ZooC, Zoo2N, and Zoo2C are listed in Table A1. A fraction of the grazed phytoplankton ($\gamma_{\text{zoo}}$ and $\gamma_{\text{zoo2}}$; Table A3) is kept in the zooplankton biomass, while the remainder is returned back to detritus pool as a consequence of sloppy feeding. Grazing terms ($G_{\text{small}}^{\text{zoo}}$, $G_{\text{dia}}^{\text{zoo}}$, $G_{\text{det}}^{\text{zoo}}$, $G_{\text{detZ2}}^{\text{zoo}}$, $G_{\text{small}}^{\text{zoo2}}$, $G_{\text{dia}}^{\text{zoo2}}$, $G_{\text{det}}^{\text{zoo2}}$, $G_{\text{detZ2}}^{\text{zoo2}}$, and $G_{\text{zoo}}$) are calculated in Sect. A4.2. The respiration terms of zooplankton ($r_{\text{zoo}}$ and $r_{\text{zoo2}}$) are calculated in Eqs. (A60) and (A61). Mortality parameters ($m_{\text{zoo}}$ and $m_{\text{zoo2}}$) are listed in Table A3. The DOC excretion terms ($\epsilon_{\text{zoo}}^{\text{C}}, \epsilon_{\text{zoo2}}^{\text{C}}$) are in Table A8. The grazing flux in terms of nitrogen biomass is converted to carbon biomass using the respective intracellular C : N ratios ($q_{\text{small}}^{\text{C:N}}$, $q_{\text{dia}}^{\text{C:N}}$, $q_{\text{det}}^{\text{C:N}}$, $q_{\text{detZ2}}^{\text{C:N}}$, $q_{\text{zoo}}^{\text{C:N}}$, and $q_{\text{zoo2}}^{\text{C:N}}$), where $q_{\text{small}}^{\text{C:N}} = \text{PhyC}_{\text{small}}/\text{PhyN}_{\text{small}}$, $q_{\text{dia}}^{\text{C:N}} = \text{PhyC}_{\text{dia}}/\text{PhyN}_{\text{dia}}$, $q_{\text{det}}^{\text{C:N}} = \text{DetC}/\text{DetN}$, $q_{\text{detZ2}}^{\text{C:N}} = \text{DetZ2C}/\text{DetZ2N}$, $q_{\text{zoo}}^{\text{C:N}} = \text{ZooC}/\text{ZooN}$, and $q_{\text{zoo2}}^{\text{C:N}} = \text{Zoo2C}/\text{Zoo2N}$. Total grazed carbon biomass ($G_{\text{cflux}}$) and the fecal pellet production rate constant ($f_{\text{c}}$; Table A3) together determine the fraction of carbon being lost to the large detritus carbon pool via fecal pellets.

## A1.5 Detritus

### Nitrogen

Detrital nitrogen pool increases as a result of sloppy feeding and mortality. Sloppy feeding is outlined as a function of grazing fluxes and grazing efficiency of macrozooplankton. In other words, the grazed phytoplankton partly goes to the macrozooplankton biomass, depending on the grazing efficiency. The phytoplankton aggregation contributes only to slow-sinking detritus. Fecal pellet production is defined only for macrozooplankton group. Detritus is degraded to DON, based on temperature and a remineralization rate.

$$
\text{SMS(DetN)} = \underbrace{(G_{\text{small}}^{\text{zoo}} + G_{\text{dia}}^{\text{zoo}}) \cdot (1 - \gamma_{\text{zoo}})}_{\text{Sloppy feeding}} + \underbrace{m_{\text{zoo}} \cdot \text{ZooN}^2}_{\text{Zooplankton mortality}}
$$
$$
- \underbrace{\gamma_{\text{zoo}} \cdot (G_{\text{det}}^{\text{zoo}} + G_{\text{detZ2}}^{\text{zoo}})}_{\text{Grazing loss, small zoo.}}
$$
$$
+ \underbrace{\text{Agg} \cdot (\text{PhyN}_{\text{small}} + \text{PhyN}_{\text{dia}})}_{\text{Phytoplankton aggregation}} - \underbrace{\rho_{\text{DetN}} \cdot f_{\text{T}} \cdot \text{DetN}}_{\text{Degradation to DON}} \quad (A27)
$$

$$
\text{SMS(DetZ2N)} = \underbrace{(G_{\text{small}}^{\text{zoo2}} + G_{\text{dia}}^{\text{zoo2}} + G_{\text{zoo}}) \cdot (1 - \gamma_{\text{zoo2}})}_{\text{Sloppy feeding}}
$$
$$
- \underbrace{\gamma_{\text{zoo2}} \cdot (G_{\text{det}}^{\text{zoo2}} + G_{\text{detZ2}}^{\text{zoo2}} \cdot)}_{\text{Grazing loss, macrozoo.}} \quad (A28)
$$
$$
+ \underbrace{m_{\text{zoo2}} \cdot \text{Zoo2N}^2}_{\text{Mortality}} + \underbrace{f_{\text{n}} \cdot G_{\text{tot}}}_{\text{Fecal pellet}}
$$
$$
- \underbrace{\rho_{\text{DetN}} \cdot f_{\text{T}} \cdot \text{DetZ2N}}_{\text{Degradation to DON}} \quad (A29)
$$

The TS14 state variables PhyN$_{\text{small}}$, PhyN$_{\text{dia}}$, ZooN, DetN, Zoo2N, and DetZ2N are listed in Table A1. The grazing efficiency ($\gamma_{\text{zoo}}$ and $\gamma_{\text{zoo2}}$), mortality ($m_{\text{zoo}}$ and $m_{\text{zoo2}}$), and fecal pellet production rate constant ($f_{\text{n}}$) are listed in Table A3. Grazing terms ($G_{\text{small}}^{\text{zoo}}$, $G_{\text{dia}}^{\text{zoo}}$, $G_{\text{det}}^{\text{zoo}}$, $G_{\text{detZ2}}^{\text{zoo}}$, $G_{\text{small}}^{\text{zoo2}}$, $G_{\text{dia}}^{\text{zoo2}}$, $G_{\text{det}}^{\text{zoo2}}$, $G_{\text{detZ2}}^{\text{zoo2}}$, and $G_{\text{zoo}}$) are calculated in Sect. A4.2. The remineralization rate constant of DON ($\rho_{\text{DetN}}$) is listed in Table A8. The temperature dependency $f_{\text{T}}$ is calculated in Eq. (A43). The aggregation (Agg) term is calculated in Eq. (A52).

### Carbon

Detrital carbon sources are associated with sloppy feeding, aggregation of phytoplankton, mortality of small zooplankton, and fecal pellet production by macrozooplankton. Degradation of DetC and DetZ2C to DOC is the only loss term.

$$
\text{SMS(DetC)} = \underbrace{(G_{\text{small}}^{\text{zoo}} \cdot q_{\text{small}}^{\text{C:N}} + G_{\text{dia}}^{\text{zoo}} \cdot q_{\text{dia}}^{\text{C:N}}) \cdot (1 - \gamma_{\text{zoo}})}_{\text{Sloppy feeding}}
$$
$$
+ \underbrace{m_{\text{zoo}} \cdot \text{ZooN}^2 \cdot q_{\text{zoo}}^{\text{C:N}}}_{\text{small zoo. mortality}}
$$
$$
- \underbrace{\gamma_{\text{zoo}} \cdot (G_{\text{det}}^{\text{zoo}} \cdot q_{\text{det}}^{\text{C:N}} + G_{\text{detZ2}}^{\text{zoo}} \cdot q_{\text{detZ2}}^{\text{C:N}})}_{\text{Grazing loss by macrozoo.}}
$$
$$
+ \underbrace{\text{Agg} \cdot (\text{PhyC}_{\text{small}} + \text{PhyC}_{\text{dia}})}_{\text{Phytoplankton aggregation}} - \underbrace{\rho_{\text{DetC}} \cdot f_{\text{T}} \cdot \text{DetC}}_{\text{Degradation to DOC}} \quad (A30)
$$

$$
\text{SMS(DetZ2C)} = \underbrace{\substack{(G_{\text{small}}^{\text{zoo2}} \cdot q_{\text{small}}^{\text{C:N}} + G_{\text{dia}}^{\text{zoo2}} \cdot q_{\text{dia}}^{\text{C:N}} + G_{\text{zoo}} \cdot q_{\text{zoo}}^{\text{C:N}}) \\ \cdot (1 - \gamma_{\text{zoo2}})}}_{\text{Sloppy feeding}}
$$
$$
- \underbrace{\gamma_{\text{zoo2}} \cdot (G_{\text{det}}^{\text{zoo2}} \cdot q_{\text{det}}^{\text{C:N}} + G_{\text{detZ2}}^{\text{zoo2}} \cdot q_{\text{detZ2}}^{\text{C:N}})}_{\text{Grazing loss by macrozoo.}} \quad (A31)
$$
$$
+ \underbrace{m_{\text{zoo2}} \cdot \text{Zoo2N}^2 \cdot q_{\text{zoo2}}^{\text{C:N}}}_{\text{Mortality}} + \underbrace{f_{\text{c}} \cdot G_{\text{cflux}}}_{\text{Fecal pellet}}
$$
$$
- \underbrace{\rho_{\text{DetC}} \cdot f_{\text{T}} \cdot \text{DetZ2C}}_{\text{Degradation to DOC}} \quad (A32)
$$

The TS15 state variables PhyC$_{\text{small}}$, PhyC$_{\text{dia}}$, ZooN, DetC, Zoo2N, and DetZ2C are listed in Table A1. The grazing efficiency ($\gamma_{\text{zoo}}$ and $\gamma_{\text{zoo2}}$) and mortality ($m_{\text{zoo}}$ and $m_{\text{zoo2}}$) parameters are listed in Table A3. Grazing terms ($G_{\text{small}}^{\text{zoo}}$, $G_{\text{dia}}^{\text{zoo}}$, $G_{\text{det}}^{\text{zoo}}$, $G_{\text{detZ2}}^{\text{zoo}}$, $G_{\text{small}}^{\text{zoo2}}$, $G_{\text{dia}}^{\text{zoo2}}$, $G_{\text{det}}^{\text{zoo2}}$, $G_{\text{detZ2}}^{\text{zoo2}}$, and $G_{\text{zoo}}$) are calculated in Sect. A4.2. The remineralization rate of DOC ($\rho_{\text{DetC}}$) is listed in Table A8. Temperature dependency $f_{\text{T}}$ is calculated in Eq. (A43). The aggregation (Agg) term is calculated in Eq. (A52). The total grazed carbon biomass ($G_{\text{cflux}}$) and the fecal pellet production rate constant ($f_{\text{c}}$; Table A3) together determine the fraction of carbon being lost to the large detritus carbon pool via fecal pellets. The quotas $q_{\text{small}}^{\text{C:N}} = \text{PhyC}_{\text{small}}/\text{PhyN}_{\text{small}}$, $q_{\text{dia}}^{\text{C:N}} = \text{PhyC}_{\text{dia}}/\text{PhyN}_{\text{dia}}$, $q_{\text{zoo}}^{\text{C:N}} = \text{ZooC}/\text{ZooN}$, $q_{\text{zoo2}}^{\text{C:N}} = \text{Zoo2C}/\text{Zoo2N}$, $q_{\text{det}}^{\text{C:N}} = \text{DetC}/\text{DetN}$, and $q_{\text{detZ2}}^{\text{C:N}} = \text{DetZ2C}/\text{DetZ2N}$ are used to convert the units from mmol N to mmol C.

### Silica

Biogenic detrital silica increases with excretion fluxes from diatoms to detritus, aggregation, and grazing and decreases with silica dissolution from DetSi and DetZ2Si.

$$
\text{SMS(DetSi)} = \underbrace{(\epsilon_{\text{phy}}^{\text{N}} \cdot f_{\text{lim, dia}}^{\text{N:Cmax}}}_{\text{Diatom excretion}} + \underbrace{\text{Agg})}_{\text{Aggregation}} \cdot \text{DiaSi}
$$
$$
+ \underbrace{G_{\text{dia}}^{\text{zoo}} \cdot q^{\text{Si:N}}}_{\text{Sloppy feeding}} - \underbrace{\rho_{\text{Si}}^{\text{T}} \cdot \text{DetSi}}_{\text{Remineralization to DSi}} \quad (A33)
$$

$$
\text{SMS(DetZ2Si)} = \underbrace{G_{\text{dia}}^{\text{zoo2}} \cdot q^{\text{Si:N}}}_{\text{Sloppy feeding}} - \underbrace{\rho_{\text{Si}}^{\text{T}} \cdot \text{DetZ2Si}}_{\text{Remineralization to DSi}} \quad (A34)
$$

The state variables DiaSi, DetSi, and DetZ2Si are listed in Table A1. The constant excretion rate ($\epsilon_{\text{phy}}^{\text{N}}$; Table A8)

is downregulated by the limiter factor $f_{\text{lim, dia}}^{\text{N:Cmax}}$ (Eq. A55) when the N : C ratio becomes too high. The remineralization rates ($\rho_{\text{Si}}^{\text{T}}$), the aggregation (Agg), and the grazing on diatoms ($G_{\text{dia}}^{\text{zoo}}$ and $G_{\text{dia}}^{\text{zoo2}}$) are calculated in Eqs. (A45), (A52), and (A65), respectively. The intracellular ratio between diatom silicon and carbon is defined as $q^{\text{Si:N}} = \text{PhySi}_{\text{dia}}/\text{PhyN}_{\text{dia}}$.

### CaCO$_3$

The coccolithophore fraction of small phytoplankton loses biogenic CaCO$_3$ to the detrital CaCO$_3$ pool along with excretion, aggregation, respiration, and grazing. Dissolution of CaCO$_3$ leads to an increase in DIC and alkalinity.

$$
\begin{aligned}
\text{SMS(DetCalc)} = &\underbrace{\epsilon_{\text{C}}^{\text{phy}} \cdot f_{\text{lim, small}}^{\text{N:Cmax}} \cdot \text{PhyCalc}}_{\text{Small phytoplankton, excretion}} \\
&+ (\underbrace{\text{Agg}}_{\text{Aggregation}} + \underbrace{r_{\text{small}}}_{\text{Respiration}}) \cdot \text{PhyCalc} \\
&+ \underbrace{G_{\text{small}}^{\text{zoo}} \cdot q_{\text{small}}^{\text{CaCO}_3\text{:N}}}_{\text{Grazing loss}} \\
&- \underbrace{G_{\text{small}}^{\text{zoo}} \cdot q_{\text{small}}^{\text{CaCO}_3\text{:N}} \cdot \text{Diss}_{\text{calc-guts}}}_{\text{CaCO}_3 \text{ dissolution in guts}} \\
&- \underbrace{\text{Diss}_{\text{calc}} \cdot \text{DetCalc}}_{\text{CaCO}_3 \text{ dissolution, slow-sinking detritus}} \quad (\text{A35})
\end{aligned}
$$

$$
\begin{aligned}
\text{SMS(DetZ2Calc)} = &\underbrace{G_{\text{small}}^{\text{zoo2}} \cdot q_{\text{small}}^{\text{CaCO}_3\text{:N}}}_{\text{Grazing loss}} \quad (\text{A36}) \\
&- \underbrace{G_{\text{small}}^{\text{zoo2}} \cdot q_{\text{small}}^{\text{CaCO}_3\text{:N}} \cdot \text{Diss}_{\text{calc-guts}}}_{\text{CaCO}_3 \text{ dissolution in guts}} \\
&- \underbrace{\text{Diss}_{\text{calc2}} \cdot \text{DetZ2Calc}}_{\text{CaCO}_3 \text{ dissolution, fast-sinking detritus}} \quad (\text{A37})
\end{aligned}
$$

The TS16 state variables PhyCalc, DetCalc, and DetZ2Calc are listed in Table A1. The constant excretion rate ($\epsilon_{\text{phy}}^{\text{C}}$; Table A8) is downregulated by the limiter factor $f_{\text{lim, small}}^{\text{N:Cmax}}$ (Eq. A55) when the N : C ratio becomes too high. The respiration ($r_{\text{small}}$), the aggregation (Agg), and the grazing on small phytoplankton ($G_{\text{small}}^{\text{zoo}}$ and $G_{\text{small}}^{\text{zoo2}}$) are calculated in Eqs. (A48), (A52), and (A64), respectively. The ratio $q_{\text{small}}^{\text{CaCO}_3\text{:N}} = \text{PhyCalc}/\text{PhyN}_{\text{small}}$.

### Calcite dissolution

As the detritus calcite sinks through the water column, it is subject to dissolution. We follow Yamanaka and Tajika (1996), assuming an exponential decrease in the CaCO$_3$ flux with depth. As we also assume an increasing sinking speed of small detritus with depth, following Kriest and Oschlies (2008), the dissolution rate is scaled with the sinking velocity.

$$
\text{Diss}_{\text{calc}} = \text{Diss}_{\text{calc\_rate}} \cdot w_{\text{det}} \qquad \text{Diss}_{\text{calc2}} = \text{Diss}_{\text{calc\_rate}} \quad (\text{A38})
$$

$\text{Diss}_{\text{calc}}$ and $\text{Diss}_{\text{calc2}}$ are the dissolution rate constants for slow- and fast-sinking detritus classes (Table A5). The reference dissolution rate ($\text{Diss}_{\text{calc\_rate}}$; Table A8) is based on a length scale of 3500 m and velocity of 20 m d$^{-1}$. The sinking speed at depth $z$ ($w_{\text{det}}$; Table A5) is calculated as follows:

$$
w_{\text{det}} = 0.0288 \cdot z + w_0. \quad (\text{A39})
$$

Here, $z$ denotes the depth, and $w_0$ is the sinking speed at the ocean surface (Table A3). The dissolution rate for a fast-sinking detritus class ($\text{Diss}_{\text{calc2}}$) is assumed to be constant throughout the water column and is set to the value of $\text{Diss}_{\text{calc\_rate}}$ (Table A8).

### A1.6 Dissolved oxygen (Oxy)

Oxy concentration increases with the carbon fixation by primary producers. It decreases with the respiration of phytoplankton and zooplankton and the remineralization of dissolved organic carbon. In addition, sea–air flux of O$_2$ leads to an exchange of oxygen with the atmosphere, depending on the partial pressure difference in the O$_2$ between ocean and atmosphere. This exchange is treated separately as a boundary condition. The partial pressure of surface ocean O$_2$ is computed using the mocsy 2.0 routines (Orr and Epitalon, 2015).

$$
\begin{aligned}
\text{SMS(Oxy)} = &\underbrace{(P_{\text{small}} - r_{\text{small}}) \cdot \text{PhyC}_{\text{small}}}_{\text{Net production, small phytoplankton}} \\
&+ \underbrace{(P_{\text{dia}} - r_{\text{dia}}) \cdot \text{PhyC}_{\text{dia}}}_{\text{Net production, diatom}} - \underbrace{\rho_{\text{DOC}} \cdot f_{\text{T}} \cdot \text{DOC}}_{\text{Remineralization of DOC}} \\
&- \underbrace{r_{\text{zoo}} \cdot \text{ZooC}}_{\text{Respiration, small zoo.}} - \underbrace{r_{\text{zoo2}} \cdot \text{Zoo2C}}_{\text{Respiration, macrozoo.}} \quad (\text{A40})
\end{aligned}
$$

The state variables PhyC$_{\text{small}}$, PhyC$_{\text{dia}}$, DOC, ZooC, and Zoo2C are listed in Table A1. Respiration rate constants of small phytoplankton ($r_{\text{small}}$), diatoms ($r_{\text{dia}}$), and zooplankton groups ($r_{\text{zoo}}$ and $r_{\text{zoo2}}$) are computed in Sects. A3.2 and A4.1, respectively. Photosynthesis terms ($P_{\text{small}}$ and $P_{\text{dia}}$) are calculated in Eq. (A46). The remineralization rate constant $\rho_{\text{DOC}}$ is listed in Table A8 and the temperature dependency ($f_{\text{T}}$) is given in Eq. (A43).

### A1.7 Dissolved organic material

Dissolved organic matter in our model is a representation of the semi-labile fraction only; the refractory and labile fractions are not included.

#### Dissolved organic nitrogen (DON)

DON is produced via nitrogen excretion by phytoplankton, by zooplankton, and by the degradation of detrital nitrogen.

DON is turned into DIN by remineralization, which is the only sink term.

$$
\begin{aligned}
\text{SMS(DON)} = &\underbrace{\epsilon_{\text{phy}}^{\text{N}} \cdot f_{\text{lim, small}}^{\text{N:Cmax}} \cdot \text{PhyN}_{\text{small}}}_{\text{Excretion, small phytoplankton}} + \underbrace{\epsilon_{\text{dia}}^{\text{N}} \cdot f_{\text{lim, dia}}^{\text{N:Cmax}} \cdot \text{PhyN}_{\text{dia}}}_{\text{Excretion, diatom}} \\
&+ \underbrace{\epsilon_{\text{zoo}}^{\text{N}} \cdot \text{ZooN}}_{\text{Excretion, small zoo.}} + \underbrace{\epsilon_{\text{zoo2}}^{\text{N}} \cdot \text{Zoo2N}}_{\text{Excretion, macrozoo.}} \\
&+ \underbrace{\rho_{\text{DetN}} \cdot f_{\text{T}} \cdot \text{DetN}}_{\text{Detritus degradation, slow-sinking}} \\
&+ \underbrace{\rho_{\text{Det2ZN}} \cdot f_{\text{T}} \cdot \text{DetZ2N}}_{\text{Detritus degradation, fast sinking}} - \underbrace{\rho_{\text{DON}} \cdot f_{\text{T}} \cdot \text{DON}}_{\text{Remineralization}}
\end{aligned}
\tag{A41}
$$

The state variables $\text{PhyN}_{\text{small}}$, $\text{PhyN}_{\text{dia}}$, ZooN, DetN, Zoo2N, DetZ2N, and DON are listed in Table A1. The constant excretion rate of nitrogen from phytoplankton and zooplankton classes ($\epsilon_{\text{phy}}^{\text{N}}$, $\epsilon_{\text{dia}}^{\text{N}}$, $\epsilon_{\text{zoo}}^{\text{N}}$, and $\epsilon_{\text{zoo2}}^{\text{N}}$), the degradation rate of detritus ($\rho_{\text{DetN}}$ and $\rho_{\text{DetZ2N}}$) and the remineralization rate of DON ($\rho_{\text{DON}}$) are listed in Table A8. The constant excretion rate of phytoplankton is downregulated by the limiter function ($f_{\text{lim, small}}^{\text{N:Cmax}}$ and $f_{\text{lim, dia}}^{\text{N:Cmax}}$; Eq. A55) when the N : C ratio becomes too high. The temperature dependency $f_{\text{T}}$ is calculated in Eq. (A43).

## Dissolved organic carbon (DOC)

DOC is produced via carbon excretion by phytoplankton, by zooplankton, and by the degradation of detrital carbon. DOC is turned into DIC by remineralization, which is the only sink term.

$$
\begin{aligned}
\text{SMS(DOC)} = &\underbrace{\epsilon_{\text{phy}}^{\text{C}} \cdot f_{\text{lim, small}}^{\text{N:Cmax}} \cdot \text{PhyC}_{\text{small}}}_{\text{Excretion, small phytoplankton}} + \underbrace{\epsilon_{\text{dia}}^{\text{C}} \cdot f_{\text{lim, dia}}^{\text{N:Cmax}} \cdot \text{PhyC}_{\text{dia}}}_{\text{Excretion, diatom}} \\
&+ \underbrace{\epsilon_{\text{zoo}}^{\text{C}} \cdot \text{ZooC}}_{\text{Excretion, small zoo.}} + \underbrace{\epsilon_{\text{zoo2}}^{\text{C}} \cdot \text{Zoo2C}}_{\text{Excretion, macrozoo.}} \\
&+ \underbrace{\rho_{\text{DetC}} \cdot f_{\text{T}} \cdot \text{DetC}}_{\text{Detritus degradation, slow sinking}} + \underbrace{\rho_{\text{Det2C}} \cdot f_{\text{T}} \cdot \text{Det2C}}_{\text{Detritus degradation, fast sinking}} \\
&- \underbrace{\rho_{\text{DOC}} \cdot f_{\text{T}} \cdot \text{DOC}}_{\text{Remineralization}}
\end{aligned}
\tag{A42}
$$

The state variables $\text{PhyC}_{\text{small}}$, $\text{PhyC}_{\text{dia}}$, ZooC, DetC, Zoo2C, Det2C, and DOC are listed in Table A1. The constant excretion rate of nitrogen from phytoplankton and zooplankton classes ($\epsilon_{\text{phy}}^{\text{C}}$, $\epsilon_{\text{dia}}^{\text{C}}$, $\epsilon_{\text{zoo}}^{\text{C}}$, and $\epsilon_{\text{zoo2}}^{\text{C}}$), the degradation rate of detritus ($\rho_{\text{DetC}}$ and $\rho_{\text{Det2C}}$), and the remineralization rate of DOC ($\rho_{\text{DOC}}$) are listed in Table A8. The constant excretion rate of phytoplankton is downregulated by the limiter factor ($f_{\text{lim, small}}^{\text{N:Cmax}}$ and $f_{\text{lim, dia}}^{\text{N:Cmax}}$; Eq. A55) when the N : C ratio becomes too high. Temperature dependency $f_{\text{T}}$ is calculated in Eq. (A43).

## A2 Temperature dependence of rates

### Arrhenius function

Most metabolic processes are faster at higher temperatures. This temperature dependence is defined relative to a refer-

ence temperature.

$$
f_{\text{T}} = \exp\left(-4500 \cdot \left(\frac{1}{T} - \frac{1}{T_{\text{ref}}}\right)\right)
\tag{A43}
$$

$T$ and $T_{\text{ref}}$ are the local and reference temperature in Kelvin, respectively (Table A6).

## Macrozooplankton grazing

Macrozooplankton grazing is temperature dependent. A dimensionless exponential temperature function (Butzin and Pörtner, 2016) is used for the parameterization of the temperature dependency ($f_{\text{Tzoo2}}$; Table A5). Specifically, the following parameterization provides an optimum curve with a maximum at 0.5 °C, as described in Karakuş et al. (2021).

$$
f_{\text{Tzoo2}} = \frac{\exp\left(\frac{Q_a}{T_r} - \frac{Q_a}{T}\right)}{1 + \exp\left(\frac{Q_h}{T_h} - \frac{Q_h}{T}\right)}
\tag{A44}
$$

$T_r$ is the intrinsic optimum temperature for development, and $T_h$ is the temperature above which inhibitive processes dominate. $Q_a$ and $Q_h$ are the temperatures for the uninhibited and inhibited reaction kinetics, respectively (Table A9). $T$ is the local temperature in Kelvin.

## Silicon dissolution

The temperature-dependent dissolution rate of silicon ($\rho_{\text{Si}}^{\text{T}}$; Table A5) is calculated following Maerz et al. (2020) but with a minimum dissolution rate.

$$
\rho_{\text{Si}}^{\text{T}} = \max\left(0.023 \cdot 2.6^{\frac{T-10}{10}}, \rho_{\text{Si}}\right)
\tag{A45}
$$

$T$ is the local temperature in degrees Celsius. The minimum dissolution rate ($\rho_{\text{Si}}$) is listed in Table A8.

## A3 Phytoplankton processes

Phytoplankton growth equations are based on Geider et al. (1998), with small modifications for diatom silicon uptake, following Hohn (2009).

### A3.1 Photosynthesis

The rate of the carbon-specific (C-specific from now on) photosynthesis for phytoplankton ($P_{\text{small}}$ and $P_{\text{dia}}$) is parameterized as follows:

$$
\begin{aligned}
P_{\text{small}} &= P_{\text{max}}^{\text{small}} \cdot \left(1.0 - \exp\left(\frac{-\alpha_{\text{small}} \cdot q^{\text{Chl:C}} \cdot \text{PAR}}{P_{\text{max}}^{\text{small}}}\right)\right), \\
P_{\text{dia}} &= P_{\text{max}}^{\text{dia}} \cdot \left(1.0 - \exp\left(\frac{-\alpha_{\text{dia}} \cdot q^{\text{Chl:C}} \cdot \text{PAR}}{P_{\text{max}}^{\text{dia}}}\right)\right).
\end{aligned}
\tag{A46}
$$

The light-harvesting efficiency ($\alpha_{\text{small}}$ and $\alpha_{\text{dia}}$) per chlorophyll is listed in Table A7. PAR is the photosynthetically

available radiation (Table A5). The intracellular Chl to C ratio ($q^{\text{Chl:C}}$) is defined as PhyChl/PhyC and varies as a result of photoacclimation. The apparent maximum photosynthetic rate ($P_{\text{max}}^{\text{small}}$ and $P_{\text{max}}^{\text{dia}}$) is defined below.

$$P_{\text{max}}^{\text{small}} = \mu_{\text{C small}}^{\text{max}} \cdot \min\left(f_{\text{lim, small}}^{\text{Fe}}, f_{\text{lim, small}}^{\text{N:Cmin}}\right) \cdot f_{\text{T}},$$

$$P_{\text{max}}^{\text{dia}} = \mu_{\text{C, dia}}^{\text{max}} \cdot \min\left(f_{\text{lim, dia}}^{\text{Fe}}, f_{\text{lim, dia}}^{\text{N:Cmin}}, f_{\text{lim, dia}}^{\text{Si:Cmin}}\right) \cdot f_{\text{T}} \quad \text{(A47)}$$

The value of $\mu_{\text{C small}}^{\text{max}}$ and $\mu_{\text{C, dia}}^{\text{max}}$ is listed in Table A7. The limitation terms ($f_{\text{lim, small}}^{\text{N:Cmin}}$, $f_{\text{lim, dia}}^{\text{N:Cmin}}$, $f_{\text{lim, dia}}^{\text{Si:Cmin}}$, $f_{\text{lim, small}}^{\text{Fe}}$, and $f_{\text{lim, dia}}^{\text{Fe}}$) are presented in Sect. A3.6, and the temperature dependency ($f_{\text{T}}$) is calculated in Eq. (A43).

## A3.2 Respiration

The phytoplankton respiration rate ($r_{\text{small}}$ and $r_{\text{dia}}$; Table A5) is calculated as a base respiration plus a second term proportional to N assimilation and as a measure of biosynthesis:

$$r_{\text{small}} = \underbrace{\text{res}_{\text{small}} \cdot f_{\text{lim, small}}^{\text{N:Cmax}}}_{\text{Maintenance}} + \underbrace{\zeta \cdot V_{\text{small}}^{\text{N}}}_{\text{N assim}},$$

$$r_{\text{dia}} = \underbrace{\text{res}_{\text{dia}} \cdot f_{\text{lim, dia}}^{\text{N:Cmax}}}_{\text{Maintenance}} + \underbrace{\zeta \cdot V_{\text{dia}}^{\text{N}}}_{\text{N assim}}. \quad \text{(A48)}$$

The values for the maintenance respiration rate ($\text{res}_{\text{small}}$ and $\text{res}_{\text{dia}}$) and the cost of biosynthesis ($\zeta$) are listed in Table A7. Si assimilation is assumed to be inexpensive, so it is not included as additional cost in the respiration (Hohn, 2009). The limiter function ($f_{\text{lim, small}}^{\text{N:Cmax}}$ and $f_{\text{lim, dia}}^{\text{N:Cmax}}$) is described in Eq. (A55), and the N assimilation rate ($V_{\text{small}}^{\text{N}}$ and $V_{\text{dia}}^{\text{N}}$) is calculated in Eq. (A50).

## A3.3 Chlorophyll *a* synthesis

The chlorophyll synthesis rate ($S_{\text{small}}^{\text{chl}}$ and $S_{\text{dia}}^{\text{chl}}$; Table A5) is proportional to N assimilation, with the proportionality factor varying as a function of the C-specific photosynthesis rate, relative to the maximum possible photosynthetic rate at the current Chl : C ratio of the cell, which depends on photosynthetically available radiation and light-harvesting efficiency.

$$S_{\text{small}}^{\text{chl}} = V_{\text{small}}^{\text{N}} \cdot q_{\text{max, small}}^{\text{Chl:N}} \cdot \min\left(1, \frac{P_{\text{small}}}{\alpha_{\text{small}} \cdot q^{\text{Chl:C}} \cdot \text{PAR}}\right),$$

$$S_{\text{dia}}^{\text{chl}} = V_{\text{dia}}^{\text{N}} \cdot q_{\text{max, dia}}^{\text{Chl:N}} \cdot \min\left(1, \frac{P_{\text{dia}}}{\alpha_{\text{dia}} \cdot q^{\text{Chl:C}} \cdot \text{PAR}}\right) \quad \text{(A49)}$$

The N assimilation ($V_{\text{small}}^{\text{N}}$ and $V_{\text{dia}}^{\text{N}}$) is computed in Eq. (A50). The conversion factor of the maximum Chl : N ratio ($q_{\text{max, small}}^{\text{Chl:N}}$ and $q_{\text{max, dia}}^{\text{Chl:N}}$) and the light-harvesting efficiency values ($\alpha_{\text{small}}$ and $\alpha_{\text{dia}}$) are listed in Table A7. The C-specific photosynthesis ($P_{\text{small}}$ and $P_{\text{dia}}$) is given in Eq. (A46). PAR is the photosynthetically available radiation (Table A5), and the intracellular Chl to C ratio ($q^{\text{Chl:C}}$) is defined as PhyChl/PhyC.

## A3.4 N and Si assimilation

### Nitrogen

The C-specific N assimilation rate is a function of the maximum rate of C-specific photosynthesis and DIN concentration. N assimilation depends on the DIN concentration in seawater via Michaelis–Menten kinetics. The maximum photosynthetic rate (SYMBOL $P_{\text{max}}^{\text{small}}$ TS17) is converted to nitrogen units by multiplication with an optimal N : C uptake ratio (SYMBOL $\sigma$ as in equation TS18). Nitrogen uptake rates are further affected by the intracellular nitrogen status $q$ through the limiter functions $f_{\text{lim}}^{\text{N:Cmax}}$ TS19 (see Eq. A55). Nitrogen assimilation is downregulated at high intracellular N : C ratio. CE12

$$V_{\text{small}}^{\text{N}} = V_{\text{cm}}^{\text{small}} \cdot P_{\text{max}}^{\text{small}} \cdot \sigma_{\text{N:C}}^{\text{small}} \cdot f_{\text{lim, small}}^{\text{N:Cmax}} \cdot \frac{\text{DIN}}{K_{\text{small}}^{\text{N}} + \text{DIN}},$$

$$V_{\text{dia}}^{\text{N}} = V_{\text{cm}}^{\text{dia}} \cdot P_{\text{max}}^{\text{dia}} \cdot \sigma_{\text{N:C}}^{\text{dia}} \cdot f_{\text{lim, dia}}^{\text{N:Cmax}} \cdot \frac{\text{DIN}}{K_{\text{dia}}^{\text{N}} + \text{DIN}} \quad \text{(A50)}$$

$V_{\text{cm}}^{\text{small}}$, $V_{\text{cm}}^{\text{dia}}$, $\sigma_{\text{N:C}}^{\text{small}}$, $\sigma_{\text{N:C}}^{\text{dia}}$, $K_{\text{small}}^{\text{N}}$, and $K_{\text{dia}}^{\text{N}}$ are listed in Table A7. The maximum rate of photosynthesis ($P_{\text{max}}^{\text{small}}$ and $P_{\text{max}}^{\text{dia}}$) is given in Eqs. (A47). $f_{\text{lim, small}}^{\text{N:Cmax}}$ and $f_{\text{lim, dia}}^{\text{N:Cmax}}$ are described in Eq. (A55). DIN corresponds to the in situ concentration.

### Silicon

The building of a silica frustule of diatoms requires silicate uptake. The C-specific Si assimilation rate ($V^{\text{Si}}$) is a function of a factor for C-specific N uptake, a rate constant of C-specific photosynthesis, maximum uptake ratio N : C for diatoms, and DSi concentration. The maximum Si : C ratio, temperature, and the scaling factor for the maximum nitrogen uptake further regulate the Si assimilation.

$$V^{\text{Si}} = V_{\text{cm}}^{\text{dia}} \cdot \mu_{\text{C, dia}}^{\text{max}} \cdot f_{\text{T}} \cdot \sigma_{\text{Si:C}} \cdot f_{\text{lim}}^{\text{Si:Cmax}} \cdot f_{\text{lim, dia}}^{\text{N:Cmax}} \cdot \frac{\text{DSi}}{K_{\text{Si}} + \text{DSi}} \quad \text{(A51)}$$

The scaling factor for the N uptake ($V_{\text{cm}}^{\text{dia}}$), the maximum rate constant of C-specific photosynthesis ($\mu_{\text{C, dia}}^{\text{max}}$), the uptake ratio of the maximum Si : C ($\sigma_{\text{Si:C}}$), and half-saturation constant for silicate uptake ($K_{\text{Si}}$) are listed in Table A7. The temperature dependency ($f_{\text{T}}$) is computed in Eq. (A43). The limitation by the intracellular ratios N : C and Si : C ($f_{\text{lim, dia}}^{\text{N:Cmax}}$ and $f_{\text{lim}}^{\text{Si:Cmax}}$) is described in Eqs. (A55) and (A56), respectively. DSi corresponds to the in situ concentration.

## A3.5 Aggregation loss

The aggregation rate (Agg; Table A5) is proportional to the concentration of small phytoplankton, diatoms, and detritus. The effect of increased stickiness of diatoms under nutrient limitation (Waite et al., 1992; Aumont et al., 2015) is taken into account by multiplying the diatom biomass with ($1 -$

$q_{\text{lim}}^{\text{dia}}$). When the nutrient limitation is high (i.e. low $q_{\text{lim}}^{\text{dia}}$), the aggregation rate increases in the model.

$$\text{Agg} = \phi_{\text{phy}} \cdot \left( \text{PhyN}_{\text{small}} + (1 - q_{\text{lim}}^{\text{dia}}) \cdot \text{PhyN}_{\text{dia}} \right)$$
$$+ \phi_{\text{det}} \cdot (\text{DetN} + \text{DetZ2N}) \tag{A52}$$

$$q_{\text{lim}}^{\text{dia}} = \min \left( f_{\text{lim, dia}}^{\text{Fe}}, f_{\text{lim, dia}}^{\text{N:Cmin}}, f_{\text{lim, dia}}^{\text{Si:Cmin}} \right) \tag{A53}$$

The state variables $\text{PhyN}_{\text{small}}$, $\text{PhyN}_{\text{dia}}$, DetN, and DetZ2N are described in Table A1. The values of the maximum aggregation loss parameters ($\phi_{\text{phy}}$ and $\phi_{\text{det}}$) are listed in Table A3. The limitation terms ($f_{\text{lim, dia}}^{\text{N:Cmin}}$, $f_{\text{lim, dia}}^{\text{Si:Cmin}}$, and $f_{\text{lim, dia}}^{\text{Fe}}$) are presented below (Sect. A3.6).

## A3.6 Nutrient limitation

The metabolic processes such as C-specific photosynthesis, respiration rate, and excretion losses are treated as functions of the intracellular nitrogen status (i.e. N : C ratios $q$), following Geider et al. (1998). Intracellular ratios between nutrients and carbon limit the uptake of nitrogen and silicon, which is modelled via a non-linear function, as in Schourup-Kristensen et al. (2014).

$$f_{\text{lim}}(\theta, q_1, q_2) = 1 - \exp(-\theta(|\Delta q| - \Delta q)^2) \tag{A54}$$

Here, $\Delta q = q_1 - q_2$ is the difference between the current intracellular nutrient : C quota and a prescribed maximum or minimum quota. The dimensionless constant $\theta$ controls the limitation.

### $f_{\text{lim}}^{\text{N:Cmax}}$

The limiter $f_{\text{lim}}^{\text{N:Cmax}}$ downregulates the metabolic processes such as nitrogen and Si assimilation, excretion, and maintenance respiration of phytoplankton when the intracellular nitrogen quota ($q^{\text{N:C}}$) becomes too high. $f_{\text{lim}}^{\text{N:Cmax}}$ is one when the current $q^{\text{N:C}} < 0.151$ (i.e. Redfield ratio; 16N : 106C) and zero for $q^{\text{N:C}} > 0.2$ (i.e. 21.2N : 106C). It determines the end of the uptake of nitrogen and silicon in assimilation processes and the cease of carbon and nitrogen release during the respiration and excretion of DON and/or DOC and the CaCO$_3$ processes of phytoplankton (see the discussion of CaCO$_3$ in Sect. A1.5 TS20).

$$f_{\text{lim, small}}^{\text{N:Cmax}} = f_{\text{lim}}(\theta_{\max}^{\text{N}}, q_{\text{small}}^{\text{N:C}}, q_{\text{small}}^{\text{N:Cmax}}),$$
$$f_{\text{lim, dia}}^{\text{N:Cmax}} = f_{\text{lim}}(\theta_{\max}^{\text{N}}, q_{\text{dia}}^{\text{N:C}}, q_{\text{dia}}^{\text{N:Cmax}}) \tag{A55}$$

The limitation function for the quota regulation is calculated with Eq. (A54). $q_{\text{small}}^{\text{N:C}}$ and $q_{\text{dia}}^{\text{N:C}}$ form the current intracellular nitrogen quota for small phytoplankton and diatoms, respectively. Dimensionless constants $\theta_{\max}^{\text{N}}$, $q_{\text{small}}^{\text{N:Cmax}}$, and $q_{\text{dia}}^{\text{N:Cmax}}$ are listed in Table A6.

### $f_{\text{lim}}^{\text{Si:Cmax}}$

The limiter $f_{\text{lim}}^{\text{Si:Cmax}}$ downregulates the Si assimilation of diatoms when the intracellular silicon quota (Si : C) becomes too high. $f_{\text{lim}}^{\text{Si:Cmax}}$ is one when the current $q^{\text{N:C}} < 0.76$ and zero for $q^{\text{N:C}} > 0.8$. It determines the end of the uptake of silicon in the assimilation processes. The limiter function is described in Eq. (A54) and is calculated as follows:

$$f_{\text{lim}}^{\text{Si:Cmax}} = f_{\text{lim}}(\theta_{\max}^{\text{Si}}, q^{\text{Si:C}}, q^{\text{Si:Cmax}}) \tag{A56}$$

Dimensionless constants $\theta_{\max}^{\text{Si}}$ and $q^{\text{Si:Cmax}}$ are listed in Table A6.

### $f_{\text{lim}}^{\text{Si:Cmin}}$

Carbon fixation and aggregation loss in diatoms are further downregulated by a factor ($f_{\text{lim, dia}}^{\text{Si:Cmin}}$; see Eq. A54) when the intracellular silicon quota ($q^{\text{Si:C}}$) approaches a minimum value ($q^{\text{Si:Cmin}}$), mimicking the arrest of cellular division at low cellular Si (Claquin et al., 2002). $f_{\text{lim, dia}}^{\text{Si:Cmin}}$ is zero when the current $q^{\text{Si:C}} < 0.04$ and one for $q^{\text{Si:C}} > 0.08$.

$$f_{\text{lim, dia}}^{\text{Si:Cmin}} = f_{\text{lim}}(\theta_{\min}^{\text{Si}}, q^{\text{Si:Cmin}}, q^{\text{Si:C}}) \tag{A57}$$

The dimensionless constants $\theta_{\min}^{\text{Si}}$ and $q^{\text{Si:Cmin}}$ are listed in Table A6.

### $f_{\text{lim}}^{\text{Fe}}$

Growth limitation by iron is modelled with Michaelis–Menten kinetics, implicitly assuming that all dissolved iron is ultimately bioavailable.

$$f_{\text{lim, small}}^{\text{Fe}} = \frac{\text{DFe}}{K_{\text{small}}^{\text{Fe}} + \text{DFe}}, \qquad f_{\text{lim, dia}}^{\text{Fe}} = \frac{\text{DFe}}{K_{\text{dia}}^{\text{Fe}} + \text{DFe}} \tag{A58}$$

The variable DFe is listed in Table A1. The half-saturation constants ($K_{\text{small}}^{\text{Fe}}$ and $K_{\text{dia}}^{\text{Fe}}$) are given in Table A6.

### $f_{\text{lim}}^{\text{N:Cmin}}$

In addition to iron limitation, photosynthesis is limited by nitrogen in small phytoplankton and diatoms using the Eq. (A54). Nitrogen limitation ($f_{\text{lim, small}}^{\text{N:Cmin}}$ and $f_{\text{lim, dia}}^{\text{N:Cmin}}$) is described as a function of the intracellular nitrogen quota ($q_{\text{small}}^{\text{N:C}}$ and $q_{\text{dia}}^{\text{N:C}}$) with growth ending at a minimum quota ($q_{\text{small}}^{\text{N:Cmin}}$ and $q_{\text{dia}}^{\text{N:Cmin}}$).

$$f_{\text{lim, small}}^{\text{N:Cmin}} = f_{\text{lim}}(\theta_{\min}^{\text{N}}, q_{\text{small}}^{\text{N:Cmin}}, q_{\text{small}}^{\text{N:C}}),$$
$$f_{\text{lim, dia}}^{\text{N:Cmin}} = f_{\text{lim}}(\theta_{\min}^{\text{N}}, q_{\text{dia}}^{\text{N:Cmin}}, q_{\text{dia}}^{\text{N:C}}) \tag{A59}$$

Dimensionless constants $\theta_{\min}^{\text{N}}$, $q_{\text{small}}^{\text{N:Cmin}}$, and $q_{\text{dia}}^{\text{N:Cmin}}$ are listed in Table A6.

## A4 Zooplankton processes

### A4.1 Zooplankton respiration

#### Small zooplankton

When the intracellular C : N ratio in zooplankton exceeds the Redfield ratio, a temperature-dependent respiration ($r_{\text{zoo}}$; Ta-

ble A5) is assumed to drive it back with a timescale $\tau$.

$$r_{zoo} = \frac{q_{zoo}^{C:N} - q_{standard}^{C:N}}{\tau} \cdot f_T \tag{A60}$$

The timescale for respiration ($\tau$) is listed in Table A7. The temperature dependence ($f_T$) is calculated in Eq. (A43). The ratios are defined as $q_{zoo}^{C:N} = ZooC/ZooN$ and $q_{Standard}^{C:N} = 106/16$.

**Macrozooplankton**

The daily respiration rate constant of macrozooplankton ($r_{zoo2}$; Table A5) is modelled following Karakuş et al. (2021).

$$r_{zoo2} = R_s \cdot (1 + R_f + R_a) \tag{A61}$$

The standard respiration rate ($R_s$) is listed in Table A3. The feeding activity factor ($R_f$; Table A5) is defined as the ratio of grazing flux to carbon biomass of macrozooplankton, which increases linearly from zero to one for ratio between 0 % and 10 % and is one otherwise. The respiration activity factor ($R_a$; Table A5) defines the reduced macrozooplankton respiration rate in austral or boreal winters with the value of $-0.5$.

### A4.2 Grazing

In REcoM3, there are two zooplankton classes, namely small zooplankton ($< 2$ cm) and macrozooplankton (2–20 cm). The small zooplankton group grazes on small phytoplankton, diatoms, and on fast- and slow-sinking detrital particles. Similarly, while macrozooplankton grazes on both phytoplankton classes and detritus groups, it further grazes on small zooplankton. The total grazing of both zooplankton groups is based on the Holling type III ingestion function as follows:

$$G_{tot}^{zoo} = \xi_{zoo} \cdot \frac{\left(\sum_i p_i \cdot N_i\right)^2}{\sigma_{zoo} + \left(\sum_i p_i \cdot N_i\right)^2} \cdot f_T \cdot ZooN. \tag{A62}$$

$G_{tot}^{zoo}$ ($G_{tot}^{zoo2}$) is the total grazing flux which is calculated for small (macro) zooplankton. ZooN (Zoo2N) is listed in Table A1. The maximum grazing rate ($\xi_{zoo}$ and $\xi_{zoo2}$) and the half-saturation constants ($\sigma_{zoo}$ and $\sigma_{zoo2}$) are listed in Table A10. The temperature dependency terms ($f_T$ and $f_{Tzoo2}$) are given in Eqs. (A43) and (A44). In the model, relative grazing preferences are implemented following Fasham et al. (1990). Variable relative grazing preferences ($p_i$) are calculated using the nominal preferences for small phytoplankton, diatoms, slow- and fast-sinking detritus, and small zooplankton (Table A10) as follows:

$$p_i = \frac{p_i' \cdot N_i}{\sum_i p_i' \cdot N_i}. \tag{A63}$$

Here, summation $i$ is done over each food source to calculate the relative proportion of the food. Total grazing is used to calculate the grazing of zooplankton groups on individual food source; i.e. small phytoplankton ($i = 1$; PhyN$_{small}$), diatoms ($i = 2$; PhyN$_{dia}$), and both detritus classes ($i = 3$, DetN; $i = 4$, DetZ2N) and ($i = 5$, ZooN) in the case of macrozooplankton as the ratio of each food source to total food source ($G_{small}$, $G_{dia}$, $G_{det}$, $G_{detZ2}$, and $G_{zoo}$).

$$G_{small}^{zoo} = G_{tot}^{zoo} \cdot \frac{p_{small} \cdot PhyN_{small}}{\sum_i p_i \cdot N_i},$$
$$G_{small}^{zoo2} = G_{tot}^{zoo2} \cdot \frac{p_{small} \cdot PhyN_{small}}{\sum_i p_i \cdot N_i} \tag{A64}$$

$$G_{dia}^{zoo} = G_{tot}^{zoo} \cdot \frac{p_{dia} \cdot PhyN_{dia}}{\sum_i p_i \cdot N_i},$$
$$G_{dia}^{zoo2} = G_{tot}^{zoo2} \cdot \frac{p_{dia} \cdot PhyN_{dia}}{\sum_i p_i \cdot N_i} \tag{A65}$$

$$G_{det}^{zoo} = G_{tot}^{zoo} \cdot \frac{p_{det} \cdot DetN}{\sum_i p_i \cdot N_i},$$
$$G_{det}^{zoo2} = G_{tot}^{zoo2} \cdot \frac{p_{det} \cdot DetN}{\sum_i p_i \cdot N_i} \tag{A66}$$

$$G_{detZ2}^{zoo} = G_{tot}^{zoo} \cdot \frac{p_{detZ2} \cdot DetZ2N}{\sum_i p_i \cdot N_i},$$
$$G_{detZ2}^{zoo2} = G_{tot}^{zoo2} \cdot \frac{p_{detZ2} \cdot DetZ2N}{\sum_i p_i \cdot N_i} \tag{A67}$$

$$G_{zoo} = G_{tot}^{zoo2} \cdot \frac{p_{ZooN} \cdot ZooN}{\sum_i p_i \cdot N_i}, \tag{A68}$$

where $G_{zoo}$ is associated with macrozooplankton grazing on small zooplankton. PhyN$_{small}$, PhyN$_{dia}$, ZooN, DetN, and DetZ2N are listed in Table A1.

## A5 Bottom boundary fluxes

The model contains a benthic layer at the sea floor. Within this benthic layer, the total amounts of organic carbon, organic nitrogen, biogenic silica, and CaCO$_3$ are modelled.

**Loss to benthos**

When the slow- and fast-sinking detritus reach the ocean bottom, they continue to sink into the benthic layer with the speeds of $w_{det}$ (Eq. A39) and $w_{detZ2} = 200$ m d$^{-1}$, respectively. This results in a detrital flux (BenF$_{DetN}$, BenF$_{DetZ2N}$, BenF$_{DetC}$, BenF$_{DetZ2C}$, BenF$_{DetSi}$, BenF$_{DetZ2Si}$, BenF$_{DetCalc}$, and BenF$_{DetZ2Calc}$; Table A11) from the water column to the benthos.

$$\text{BenF}_{\text{DetN}} = -w_{\text{det}} \cdot \text{DetN} \tag{A69}$$

$$\text{BenF}_{\text{DetC}} = -w_{\text{det}} \cdot \text{DetC} \tag{A70}$$

$$\text{BenF}_{\text{DetSi}} = -w_{\text{det}} \cdot \text{DetSi} \tag{A71}$$

$$\text{BenF}_{\text{DetCalc}} = -w_{\text{det}} \cdot \text{DetCalc} \tag{A72}$$

$$\text{BenF}_{\text{DetZ2N}} = -w_{\text{detZ2}} \cdot \text{DetZ2N} \tag{A73}$$

$$\text{BenF}_{\text{DetZ2C}} = -w_{\text{detZ2}} \cdot \text{DetZ2C} \tag{A74}$$

$$\text{BenF}_{\text{DetZ2Si}} = -w_{\text{detZ2}} \cdot \text{DetZ2Si} \tag{A75}$$

$$\text{BenF}_{\text{DetZ2Calc}} = -w_{\text{detZ2}} \cdot \text{DetZ2Calc} \tag{A76}$$

These fluxes increase the total amount of the different benthic state variables. The state variables of DetN, DetC, DetSi, DetCalc, DetZ2N, DetZ2C, DetZ2Si, and DetZ2Calc are described in Table A1.

**Input from benthos**

The lowermost ocean layer located next to the benthic layer receives remineralized inorganic matter back from the benthos. At the same time, these fluxes reduce the number of the benthic variables. In addition, a sediment flux of Fe from the sediment is calculated from the nitrogen flux but assuming a Fe : N ratio that is higher than in the biomass. This parameterization models a scenario in which the release of iron from the sediment is driven by redox processes, which are ultimately tied to their remineralization of organic matter.

$$\text{BenF}_{\text{DIN}} = \rho_{\text{ben}}^{\text{N}} \cdot \text{BenthosN} \tag{A77}$$

$$\text{BenF}_{\text{DSi}} = \rho_{\text{ben}}^{\text{Si}} \cdot \text{BenthosSi} \tag{A78}$$

$$\text{BenF}_{\text{DIC}} = \rho_{\text{ben}}^{\text{C}} \cdot \text{BenthosC} + \text{Diss}_{\text{calc}} \cdot \text{BenthosCalc}$$
$$+ \text{Diss}_{\text{calc2}} \cdot \text{BenthosCalc2} \tag{A79}$$

$$\text{BenF}_{\text{Alk}} = (1 + 1/16) \cdot \rho_{\text{ben}}^{\text{N}} \cdot \text{BenthosN} + 2 \cdot \text{Diss}_{\text{calc}}$$
$$\cdot \text{BenthosCalc} \tag{A80}$$

$\text{BenF}_{\text{DIN}}$, $\text{BenF}_{\text{DSi}}$, $\text{BenF}_{\text{DIC}}$, and $\text{BenF}_{\text{Alk}}$ (Table A11) denote the fluxes of DIN, DSi, DIC, and Alk returned into the bottom layer of the ocean. Constant remineralization rates ($\rho_{\text{ben}}^{\text{N}}$, $\rho_{\text{ben}}^{\text{Si}}$, and $\rho_{\text{ben}}^{\text{C}}$) are listed in Table A8. The calcite dissolution rates $\text{Diss}_{\text{calc}}$ and $\text{Diss}_{\text{calc2}}$ are calculated in Eq. (A38). BenthosN, BenthosSi, BenthosC, and BenthosCalc denote the vertically integrated benthos concentration of dissolved nitrogen, silicate, carbon, and calcium carbonate, respectively (Table A2). The alkalinity of the lowermost ocean layer located next to the benthic layer is changed by the remineralization of DIN, dissolved inorganic phosphate converted from DIN with Redfield ratio, and the dissolution of calcite from the benthos.

**Table A1.** List of oceanic state variables in REcoM3.

| Variable | Description | Unit |
|---|---|---|
| DIN | Dissolved inorganic nitrogen | $(\text{mmol N m}^{-3})$ |
| DSi | Dissolved inorganic silicon | $(\text{mmol N m}^{-3})$ |
| DFe | Dissolved inorganic iron | $(\mu\text{mol Fe m}^{-3})$ |
| DIC | Dissolved inorganic carbon | $(\text{mmol C m}^{-3})$ |
| Alk | Alkalinity | $(\text{mmol C m}^{-3})$ |
| $\text{PhyN}_{\text{small}}$ | Intracellular nitrogen concentration in small phytoplankton | $(\text{mmol N m}^{-3})$ |
| $\text{PhyC}_{\text{small}}$ | Intracellular carbon concentration in small phytoplankton | $(\text{mmol C m}^{-3})$ |
| PhyCalc | Intracellular calcite concentration in small phytoplankton | $(\text{mmol CaCO}_3\,\text{m}^{-3})$ |
| $\text{PhyChl}_{\text{small}}$ | Intracellular Chl $a$ concentration in small phytoplankton | $(\text{mg Chl m}^{-3})$ |
| $\text{PhyN}_{\text{dia}}$ | Intracellular nitrogen concentration in diatoms | $(\text{mmol N m}^{-3})$ |
| $\text{PhyC}_{\text{dia}}$ | Intracellular carbon concentration in diatoms | $(\text{mmol C m}^{-3})$ |
| $\text{PhySi}_{\text{dia}}$ | Intracellular silicon concentration in diatoms | $(\text{mmol Si m}^{-3})$ |
| $\text{PhyChl}_{\text{dia}}$ | Intracellular Chl $a$ concentration in diatoms | $(\text{mg Chl m}^{-3})$ |
| ZooN | Small zooplankton nitrogen concentration | $(\text{mmol N m}^{-3})$ |
| Zoo2N | Macrozooplankton nitrogen concentration | $(\text{mmol N m}^{-3})$ |
| ZooC | Small zooplankton carbon concentration | $(\text{mmol C m}^{-3})$ |
| Zoo2C | Macrozooplankton carbon concentration | $(\text{mmol C m}^{-3})$ |
| DetN | Slow-sinking detritus nitrogen concentration | $(\text{mmol N m}^{-3})$ |
| DetZ2N | Fast-sinking detritus nitrogen concentration | $(\text{mmol N m}^{-3})$ |
| DetC | Slow-sinking detritus carbon concentration | $(\text{mmol C m}^{-3})$ |
| DetZ2C | Fast-sinking detritus carbon concentration | $(\text{mmol C m}^{-3})$ |
| DetCalc | Slow-sinking detritus calcite concentration | $(\text{mmol CaCO}_3\,\text{m}^{-3})$ |
| DetZ2Calc | Fast-sinking detritus calcite concentration | $(\text{mmol CaCO}_3\,\text{m}^{-3})$ |
| DetSi | Slow-sinking detritus silicon concentration | $(\text{mmol Si m}^{-3})$ |
| DetZ2Si | Fast-sinking detritus silicon concentration | $(\text{mmol Si m}^{-3})$ |
| DON | Extracellular dissolved organic nitrogen | $(\text{mmol N m}^{-3})$ |
| DOC | Extracellular dissolved organic carbon | $(\text{mmol C m}^{-3})$ |
| Oxy | Dissolved oxygen concentration | $(\text{mmol O m}^{-3})$ |

**Table A2.** List of benthic state variables in REcoM3.

| Variable | Description | Unit |
|---|---|---|
| BenthosN | Vertically integrated N concentration | $(\text{mmol N m}^{-2})$ |
| BenthosC | Vertically integrated C concentration | $(\text{mmol C m}^{-2})$ |
| BenthosSi | Vertically integrated Si concentration | $(\text{mmol Si m}^{-2})$ |
| BenthosCalc | Vertically integrated calcite concentration | $(\text{mmol CaCO}_3\,\text{m}^{-2})$ |

**Table A3.** Parameters for equations of sources minus sinks.

| Parameter | Value | Description | Unit |
|---|---|---|---|
| $\psi$ | 0.02 | Calcite production ratio | (dimensionless) |
| $\gamma_{\text{zoo}}$ | 0.4 | Fraction of grazing flux to small zooplankton pool | (dimensionless) |
| $\gamma_{\text{zoo2}}$ | 0.8 | Fraction of grazing flux to macrozooplankton pool | (dimensionless) |
| $m_{\text{zoo}}$ | 0.05 | Small zooplankton mortality rate | $(\text{m}^3\,\text{mmol N}^{-1}\,\text{d}^{-1})$ |
| $m_{\text{zoo2}}$ | 0.003 | Macrozooplankton mortality rate | $(\text{m}^3\,\text{mmol N}^{-1}\,\text{d}^{-1})$ |
| $\phi_{\text{phy}}$ | 0.015 | Max aggregation loss parameter for phytoplankton N | $(\text{m}^3\,\text{mmol N}^{-1}\,\text{d}^{-1})$ |
| $\phi_{\text{det}}$ | 0.165 | Max aggregation loss parameter for detritus N | $(\text{m}^3\,\text{mmol N}^{-1}\,\text{d}^{-1})$ |
| $w_0$ | 20.0 | Detritus sinking speed at surface | $(\text{m d}^{-1})$ |
| $f_{\text{n}}$ | 0.104 | N fecal pellet production rate constant | $(\text{m}^3\,\text{mmol N}^{-1}\,\text{d}^{-1})$ |
| $f_{\text{c}}$ | 0.236 | C fecal pellet production rate constant | $(\text{m}^3\,\text{mmol C}^{-1}\,\text{d}^{-1})$ |

**Table A4.** Parameters for iron calculations.

| Parameter | Value | Description | Unit |
|---|---|---|---|
| $q^{\text{Fe:N}}$ | 0.033 | Intracellular Fe : N ratio | $(\mu\text{mol Fe mmol N}^{-1})$ |
| $K_{\text{Fe}_L}$ | 100.0 | Iron stability constant | $(\text{m}^{-3}\,\mu\text{mol})$ |
| $L_T$ | 1.0 | Total ligand concentration | $(\mu\text{mol m}^{-3})$ |
| $\kappa_{\text{Fe}}$ | 0.07 | Scavenging rate of iron | $(\text{m}^3\,\text{mmol C}^{-1}\,\text{d}^{-1})$ |
| $q^{\text{Fe:N}}$ | 0.033 | Intracellular Fe : N ratio | $(\mu\text{mol Fe mmol N}^{-1})$ |

**Table A5.** Model variables.

| Variable | Description | Unit |
|---|---|---|
| Agg | Aggregation rate constant | $(\text{d}^{-1})$ |
| $\text{Diss}_{\text{calc}}$ | The dissolution rate constant for slow-sinking detritus | $(\text{d}^{-1})$ |
| $\text{Diss}_{\text{calc2}}$ | The dissolution rate constant for fast-sinking detritus | $(\text{d}^{-1})$ |
| $\text{Diss}_{\text{calc\_guts}}$ | Dissolution of calcium carbonate in guts constant | $(\text{d}^{-1})$ |
| $w_{\text{det}}$ | Sinking velocity of detritus | $(\text{m d}^{-1})$ |
| $f_T$ | Temperature dependence of rates | (dimensionless) |
| $f_{\text{Tzoo2}}$ | Temperature dependence of macrozooplankton grazing rates | (dimensionless) |
| $G_{\text{tot}}$ | Total zooplankton grazing rate | $(\text{mmol N m}^{-3}\,\text{d}^{-1})$ |
| $G_{\text{small}}$ | Small phytoplankton specific zooplankton grazing rate | $(\text{mmol N m}^{-3}\,\text{d}^{-1})$ |
| $G_{\text{dia}}$ | Diatom specific zooplankton grazing rate | $(\text{mmol N m}^{-3}\,\text{d}^{-1})$ |
| PAR | Photosynthetically available radiation | $(\text{W m}^{-2})$ |
| $P_{\text{small}}, P_{\text{dia}}$ | C-specific actual rate constant of photosynthesis | $(\text{d}^{-1})$ |
| $P_{\text{max}}$ | C-specific light saturated rate constant of photosynthesis | $(\text{d}^{-1})$ |
| $r_{\text{small}}$ | Small phytoplankton respiration rate constant | $(\text{d}^{-1})$ |
| $r_{\text{dia}}$ | Diatoms respiration rate constant | $(\text{d}^{-1})$ |
| $r_{\text{zoo}}$ | Small zooplankton respiration rate constant | $(\text{d}^{-1})$ |
| $r_{\text{zoo2}}$ | Macrozooplankton respiration rate constant | $(\text{d}^{-1})$ |
| $R_{\text{f}}$ | Macrozooplankton feeding activity factor | $(\text{d}^{-1})$ |
| $R_{\text{a}}$ | Macrozooplankton respiration activity factor | $(\text{d}^{-1})$ |
| $S_{\text{small}}^{\text{chl}}, S_{\text{dia}}^{\text{chl}}$ | Rate of chlorophyll $a$ synthesis | $(\text{mg Chl mmol C}^{-1}\,\text{d}^{-1})$ |
| $T$ | Local temperature | (K) |
| $V_{\text{small}}^{\text{N}}, V_{\text{dia}}^{\text{N}}$ | N assimilation | $(\text{mmol N mmol C}^{-1}\,\text{d}^{-1})$ |
| $\rho_{\text{Si}}^{\text{T}}$ | Temperature-dependent remineralization rate constant of Si | $(\text{d}^{-1})$ |
| $V^{\text{Si}}$ | Si assimilation | $(\text{mmol Si mmol C}^{-1}\,\text{d}^{-1})$ |

**Table A6.** Parameters for limitation functions.

| Parameter | Value | Description | Unit |
|---|---|---|---|
| $K_{\text{small}}^{\text{Fe}}$ | 0.04 | Half-saturation constant for small phytoplankton Fe uptake | ($\mu$mol Fe m$^{-3}$) |
| $K_{\text{dia}}^{\text{Fe}}$ | 0.12 | Half-saturation constant for diatom Fe uptake | ($\mu$mol Fe m$^{-3}$) |
| $q_{\text{small}}^{\text{N:Cmin}}$ | 0.04 | Minimum intracellular N : C ratio for small phytoplankton | (mmol N mmol C$^{-1}$) |
| $q_{\text{dia}}^{\text{N:Cmin}}$ | 0.04 | Minimum intracellular N : C ratio for diatoms | (mmol N mmol C$^{-1}$) |
| $q_{\text{small}}^{\text{N:Cmax}}$ | 0.2 | Maximum intracellular N : C ratio for small phytoplankton | (mmol N mmol C$^{-1}$) |
| $q_{\text{dia}}^{\text{N:Cmax}}$ | 0.2 | Maximum intracellular N : C ratio for diatoms | (mmol N mmol C$^{-1}$) |
| $q^{\text{Si:Cmin}}$ | 0.04 | Minimum intracellular Si : C ratio for diatoms | (mmol Si mmol C$^{-1}$) |
| $q^{\text{Si:Cmax}}$ | 0.8 | Maximum intracellular Si : C ratio for diatoms | (mmol Si mmol C$^{-1}$) |
| $\theta_{\text{min}}^{\text{N}}$ | 50 | Minimum limiter regulator for N | (mmol C mmol N$^{-1}$) |
| $\theta_{\text{max}}^{\text{N}}$ | 1000 | Maximum limiter regulator for N | (mmol C mmol N$^{-1}$) |
| $\theta_{\text{min}}^{\text{Si}}$ | 1000 | Minimum limiter regulator for Si | (mmol C mmol N$^{-1}$) |
| $\theta_{\text{max}}^{\text{Si}}$ | 1000 | Maximum limiter regulator for Si | (mmol C mmol N$^{-1}$) |
| $T_{\text{ref}}$ | 288.15 | Reference temperature for Arrhenius function | (K) |

**Table A7.** Parameters for phytoplankton processes. TS21

| Parameter | Value | Description | Unit |
|---|---|---|---|
| $\alpha_{\text{small}}$ | 0.14 | Light-harvesting efficiency for small phytoplankton | (mmol C m$^2$ (mg Chl W d)$^{-1}$) |
| $\alpha_{\text{dia}}$ | 0.19 | Light-harvesting efficiency for diatoms | (mmol C m$^2$ (mg Chl W d)$^{-1}$) |
| $\mu_{\text{C,small}}^{\text{max}}$ | 3.0 | Rate constant of C-specific photosynthesis | (d$^{-1}$) |
| $\mu_{\text{C, dia}}^{\text{max}}$ | 3.5 | Rate constant of C-specific photosynthesis | (d$^{-1}$) |
| res$_{\text{small}}$ | 0.01 | Maintenance respiration rate constant | (d$^{-1}$) |
| res$_{\text{dia}}$ | 0.01 | Maintenance respiration rate constant | (d$^{-1}$) |
| $\zeta$ | 2.33 | Cost of biosynthesis of N | (mmol C mmol N$^{-1}$) |
| $q_{\text{max, small}}^{\text{Chl:N}}$ | 3.15 | Maximum Chl : N ratio for phytoplankton | (mg Chl mmol N$^{-1}$) |
| $q_{\text{max, dia}}^{\text{Chl:N}}$ | 4.2 | Maximum Chl : N ratio for phytoplankton | (mg Chl mmol N$^{-1}$) |
| $K_{\text{small}}^{\text{N}}$ | 0.55 | Half-saturation constant for small phytoplankton N uptake | (mmol N m$^{-3}$) |
| $K_{\text{dia}}^{\text{N}}$ | 1.00 | Half-saturation constant for diatom N uptake | (mmol N m$^{-3}$) |
| $V_{\text{cm}}^{\text{small}}$ | 0.7 | Scaling factor for C-specific N uptake for small phytoplankton | (dimensionless) |
| $V_{\text{cm}}^{\text{dia}}$ | 0.7 | Scaling factor for C-specific N uptake for diatoms | (dimensionless) |
| $\sigma_{\text{N:C}}^{\text{small}}$ | 0.2 | Maximum uptake ratio N : C for small phytoplankton | (mmol N mmol C$^{-1}$) |
| $\sigma_{\text{N:C}}^{\text{dia}}$ | 0.2 | Maximum uptake ratio N : C for diatoms | (mmol N mmol C$^{-1}$) |
| $K_{\text{Si}}$ | 4.00 | Half-saturation constant for diatom Si uptake | (mmol Si m$^{-3}$) |
| $\sigma_{\text{Si:C}}$ | 0.2 | Maximum uptake ratio Si : C | (mmol Si mmol C$^{-1}$) |

**Table A8.** Degradation parameters for sources-minus-sinks equations.

| Parameter | Value | Description | Unit |
|---|---|---|---|
| $\epsilon_{\text{phy}}^{\text{N}}$ | 0.05 | Small phytoplankton excretion constant of organic N | $(\text{d}^{-1})$ |
| $\epsilon_{\text{dia}}^{\text{N}}$ | 0.05 | Diatoms excretion constant of organic N | $(\text{d}^{-1})$ |
| $\epsilon_{\text{phy}}^{\text{C}}$ | 0.1 | Small phytoplankton excretion constant of organic C | $(\text{d}^{-1})$ |
| $\epsilon_{\text{dia}}^{\text{C}}$ | 0.1 | Diatoms excretion constant of organic C | $(\text{d}^{-1})$ |
| $\epsilon_{\text{zoo}}^{\text{N}}$ | 0.15 | Small zooplankton excretion constant of organic N | $(\text{d}^{-1})$ |
| $\epsilon_{\text{zoo2}}^{\text{N}}$ | 0.02 | Macrozooplankton excretion constant of organic N | $(\text{d}^{-1})$ |
| $\epsilon_{\text{zoo}}^{\text{C}}$ | 0.15 | Small zooplankton excretion constant of organic C | $(\text{d}^{-1})$ |
| $\epsilon_{\text{zoo2}}^{\text{C}}$ | 0.02 | Macrozooplankton excretion constant of organic C | $(\text{d}^{-1})$ |
| $\rho_{\text{ben}}^{\text{N}}$ | 0.005 | Remineralization rate constant for benthos N | $(\text{d}^{-1})$ |
| $\rho_{\text{ben}}^{\text{Si}}$ | 0.005 | Remineralization rate constant for benthos Si | $(\text{d}^{-1})$ |
| $\rho_{\text{ben}}^{\text{C}}$ | 0.005 | Remineralization rate constant for benthos C | $(\text{d}^{-1})$ |
| $\rho_{\text{DON}}$ | 0.11 | Remineralization constant of DON | $(\text{d}^{-1})$ |
| $\rho_{\text{DOC}}$ | 0.1 | Remineralization constant of DOC | $(\text{d}^{-1})$ |
| $\rho_{\text{DetN}}$ | 0.165 | Degradation constant of DetN | $(\text{d}^{-1})$ |
| $\rho_{\text{DetZ2N}}$ | 0.165 | Degradation constant of DetZ2N | $(\text{d}^{-1})$ |
| $\rho_{\text{DetC}}$ | 0.15 | Degradation constant of DetC | $(\text{d}^{-1})$ |
| $\rho_{\text{DetZ2C}}$ | 0.15 | Degradation constant of DetZ2C | $(\text{d}^{-1})$ |
| $\text{deg}_{\text{small}}^{\text{chl}}$ | 0.2 | Small phytoplankton chlorophyll $a$ degradation rate constant | $(\text{d}^{-1})$ |
| $\text{deg}_{\text{dia}}^{\text{chl}}$ | 0.2 | Diatom chlorophyll $a$ degradation rate constant | $(\text{d}^{-1})$ |
| $\text{Diss}_{\text{calc\_rate}}$ | 0.005714 | Dissolution of calcium carbonate constant | $(\text{d}^{-1})$ |

**Table A9.** Parameters for macrozooplankton grazing.

| Parameter | Value | Description | Unit |
|---|---|---|---|
| $Q_a$ | 28 145 | Temperatures for the uninhibited reaction kinetics | (K TS22) |
| $Q_h$ | 105 234 | Temperatures for the inhibited reaction kinetics | (K) |
| $T_r$ | 272.5 | Intrinsic optimum temperature | (K) |
| $T_h$ | 274.5 | Temperature above which inhibitive processes dominate | (K) |

**Table A10.** Parameters for zooplankton processes.

| Parameter | Value | Description | Unit |
|---|---|---|---|
| $\xi_{zoo}$ | 2.4 | Maximum grazing rate constant, small zooplankton | $(d^{-1})$ |
| $\xi_{zoo2}$ | 0.1 | Maximum grazing rate constant, macrozooplankton | $(d^{-1})$ |
| $\sigma_{zoo}$ | 0.35 | Half-saturation constant, small zooplankton | $((mmol\,N\,m^{-3})^2)$ |
| $\sigma_{zoo2}$ | 0.0144 | Half-saturation constant, macrozooplankton | $((mmol\,N\,m^{-3})^2)$ |
| $\tau$ | 0.01 | Timescale constant for zooplankton respiration | $(d^{-1})$ |
| $R_s$ | 0.0107 | Standard respiration rate constant | $(d^{-1})$ |
| | | Small zooplankton | |
| $p'_{small}$ | 1.0 | Initial grazing preference for small phytoplankton | (dimensionless) |
| $p'_{dia}$ | 0.5 | Initial grazing preference for diatoms | (dimensionless) |
| $p'_{det}$ | 0.5 | Initial grazing preference for slow-sinking detritus | (dimensionless) |
| $p'_{detZ2}$ | 0.5 | Initial grazing preference for fast-sinking detritus | (dimensionless) |
| | | Macrozooplankton | |
| $p'_{small}$ | 0.5 | Initial grazing preference for small phytoplankton | (dimensionless) |
| $p'_{dia}$ | 1.0 | Initial grazing preference for diatoms | (dimensionless) |
| $p'_{zoo}$ | 0.8 | Initial grazing preference for zooplankton | (dimensionless) |
| $p'_{det}$ | 0.5 | Initial grazing preference for slow-sinking detritus | (dimensionless) |
| $p'_{detZ2}$ | 0.5 | Initial grazing preference for fast-sinking detritus | (dimensionless) |

**Table A11.** Benthos variables.

| Variable | Description | Unit |
|---|---|---|
| $BenF_{Alk}$ | Flux of alkalinity from benthos to bottom water | $(mmol\,m^{-2}\,d^{-1})$ |
| $BenF_{DIC}$ | Flux of C from benthos to bottom water | $(mmol\,C\,m^{-2}\,d^{-1})$ |
| $BenF_{DIN}$ | Flux of N from benthos to bottom water | $(mmol\,N\,m^{-2}\,d^{-1})$ |
| $BenF_{DSi}$ | Flux of Si from benthos to bottom water | $(mmol\,Si\,m^{-2}\,d^{-1})$ |
| $BenF_{DetCalc}$ | Flux of slow-sinking detritus calcite from the water to the benthos | $(mmol\,CaCO_3\,m^{-2}\,d^{-1})$ |
| $BenF_{DetC}$ | Flux of slow-sinking detritus C from the water to the benthos | $(mmol\,C\,m^{-2}\,d^{-1})$ |
| $BenF_{DetN}$ | Flux of slow-sinking detritus N from the water to the benthos | $(mmol\,N\,m^{-2}\,d^{-1})$ |
| $BenF_{DetSi}$ | Flux of slow-sinking detritus Si from the water to the benthos | $(mmol\,Si\,m^{-2}\,d^{-1})$ |
| $BenF_{DetZ2Calc}$ | Flux of fast-sinking detritus calcite from the water to the benthos | $(mmol\,CaCO_3\,m^{-2}\,d^{-1})$ |
| $BenF_{DetZ2C}$ | Flux of fast-sinking detritus C from the water to the benthos | $(mmol\,C\,m^{-2}\,d^{-1})$ |
| $BenF_{DetZ2N}$ | Flux of fast-sinking detritus N from the water to the benthos | $(mmol\,N\,m^{-2}\,d^{-1})$ |
| $BenF_{DetZ2Si}$ | Flux of fast-sinking detritus Si from the water to the benthos | $(mmol\,Si\,m^{-2}\,d^{-1})$ |

*Code availability.* The FESOM2.1–REcoM3 source code is available at https://github.com/FESOM/fesom2/tree/fesom2.1_recom (last access: 31 December 2022). The version of FESOM2.1–REcoM3 used for this paper can be found at https://doi.org/10.5281/zenodo.7502419 (Gürses, 2023). A manual is available at https://recom.readthedocs.io/en/latest/ (last access: 23 July 2023).

*Data availability.* The GLODAPv2 mapped product of dissolved inorganic carbon and alkalinity can be found at https://glodap.info/index.php/mapped-data-product/ (last access 26 July 2023) TS23. Dissolved inorganic nitrogen and dissolved inorganic silicon from the World Ocean Atlas 2013 can be found at https://www.ncei.noaa.gov/access/metadata/landing-page/bin/iso?id=gov.noaa.nodc:0114815 (last access 26 July 2023) TS24. The oxygen data set from the World Ocean Atlas 2018 is available at https://www.ncei.noaa.gov/access/world-ocean-atlas-2018/bin/woa18oxnu.pl?parameter=o (last access 26 July 2023) TS25. The OC-CCI data set is available at https://www.oceancolour.org/thredds/catalog-cci.html (last access 26 July 2023) TS26. The Southern Ocean chlorophyll *a* concentration data set is available at https://imos.org.au/facilities/srs/oceancolour (last access 26 July 2023) TS27. CbPM and VGPM products of net primary production are available at http://sites.science.oregonstate.edu/ocean.productivity/inputData.php (last access 26 July 2023) TS28. MAREDAT products can be found at https://doi.org/10.1594/PANGAEA.779970 (Buitenhuis et al., 2012), https://doi.org/10.1594/PANGAEA.785501 (O'Brien and Moriarty, 2012), and https://doi.org/10.1594/PANGAEA.777398 (Moriarty, 2012) TS29. The mixed layer depth data are available at https://github.com/jbsallee-ocean/GlobalMLDchange/tree/main/Databases (last access 26 July 2023) TS30. Global maps of trends and climatological fields are available at https://zenodo.org/record/4073174#.YA_jsC2S3XQ (last access 26 July 2023; DOI: https://doi.org/10.5281/zenodo.4073174, Sallée et al., 2020 TS31). The Surface Ocean $CO_2$ Atlas (SOCAT) was taken from https://socat.info/index.php/previous-versions/ (last access 26 July 2023) TS32. The Global Ocean Surface Carbon is available at https://data.marine.copernicus.eu/product/MULTIOBS_GLO_BIO_CARBON_SURFACE_REP_015_008/description (last access 26 July 2023) TS33. Annual air–sea $CO_2$ flux data is freely available at https://globalcarbonbudgetdata.org (last access 26 July 2023) TS34. The Polar science center Hydrographic Climatology (Steele et al., 2001) TS35 data used for model initialization and the CORE-II atmospheric forcing data (Large and Yeager, 2009) TS36 are freely available online. TS37

*Author contributions.* The conceptualization was done by ÖG, LO, CV, and JH. The data were prepared by ÖG, JH, and OK. All authors analysed the simulations and visualizations and contributed to the writing of the paper.

*Competing interests.* The contact author has declared that none of the authors has any competing interests.

*Acknowledgements.* This research has been funded by the Initiative and Networking Fund of the Helmholtz Association (Helmholtz Young Investigator Group Marine Carbon and Ecosystem Feedbacks in the Earth System, MarESys; grant no. VH-NG-1301) and the ERC-2022-STG OceanPeak (grant no. 101077209). Laurent Oziel and Ying Ye received funding from the European Union's Horizon 2020 research and innovation programme (grant no. 820989; COMFORT project). The work reflects only the authors' views; the European Commission and their executive agency are not responsible for any use that may be made of the information that the work contains. Laurent Oziel acknowledges funding from the Germany Ministry for Education and Research (BMBF; grant no. 03F0918A; nuArctic project). Moritz Zeising has been funded by the Deutsche Forschungsgemeinschaft (DFG, German Research Foundation; grant no. 268020496 – TRR 172) within the Transregional Collaborative Research Center "ArctiC Amplification: Climate Relevant Atmospheric and SurfaCe Processes, and Feedback Mechanisms (AC)[3]" project. Ying Ye and Martin Butzin have received funding by the German Federal Ministry of Education and Research (BMBF; grant no. 01LP1919A; PalMod project). Martin Butzin has additionally been funded through the DFG-ANR project of MARCARA. We thank Sergey Danilov from AWI Bremerhaven, Germany, for his helpful support. We acknowledge the Global Carbon Project, which is responsible for the Global Carbon Budget, and we thank the ocean modelling and TS38 $f\,CO_2$ mapping groups for producing and making available their model and $f\,CO_2$ product output.

*Financial support.* This research has been supported by the NAME OF FUNDER (grant no. GRANT AGREEMENT NO). TS39

The article processing charges for this open-access publication were covered by the Alfred Wegener Institute, Helmholtz Centre for Polar and Marine Research (AWI).

*Review statement.* This paper was edited by Riccardo Farneti and reviewed by two anonymous referees.

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

TS44    Please provide date of last access.

TS45    Please confirm article number