# Peer review of "Ocean biogeochemistry in the coupled ocean-sea ice-biogeochemistry model FESOM2.1-REcoM3"

_Geoscientific Model Development, 2023_

## Author Response (AR1)

In the following, referee comments are in black, while our responses are in green and added material is indicated in blue.

**Reviewer 1**

**Overall comments**
The manuscript "Ocean biogeochemistry in the coupled ocean-sea ice-biogeochemistry model FESOM2.1-REcoM3" is a comprehensive analysis of a new iteration (version 2.1) of the FESOM global ocean model, coupled with the biogeochemical model REcoM3 (also an updated version). The authors set out to evaluate the model for a recent time period (1958-2021) to assess if the model can be used to evaluate climate change effects and $CO_2$ increases on a century timescale.
Overall, I think the authors have presented a clear and thorough manuscript that was generally well-written and easy to read. The authors presented model results that showed relatively good agreement with observations, and illustrated that the new iteration of the FESOM and REcoM models had some improvements for the model's overall performance. I believe this manuscript will be a valuable contribution. I just have some relatively minor comments, which are detailed below.

We would like to thank the referee for the overall positive assessment of our manuscript .

**Specific comments**
1. I thought in general the authors had a really nice and detailed Methods section. For example, the description of the FESOM model grid was very clear and I appreciated the section explicitly listing changes to REcoM. However, I found there were a few details missing from the Methods. See specific lines below:
· Methods: Forcing (line 200ff): I'm curious about what river forcings you used, if any? Perhaps I missed it but it doesn't seem to be detailed in your methods. I think this could be particularly important to note since you comment at line 564-565 that future work is looking at the role of rivers for carbon and nutrient transport.

Good point. We have specified the river (freshwater) forcing. River carbon and nutrient input is not included in these experiments and we have also specified this now. While certainly important, river carbon, alkalinity and nutrient input also needs to be carefully evaluated and sensitivity studies will be needed. This is a full project on its own and already going on in parallel.

We added this information to the text:

"The freshwater supplied by rivers is a climatology and provided by Large and Yeager (2004) as part of the CORE forcing. Nutrient, carbon and alkalinity supply via river discharge is not included in the experiments described here."·

Line 208ff: I think a few more details should be specified about the initialization fields; in particular, I think it would be worthwhile to explicitly state the years that the various climatologies span. Additionally, you point to GLODAP for Alkalinity and DIC initialization but don't describe what years from GLODAP were used?

We slightly changed sentences to include the time period as follows:

Initial fields for temperature and salinity were taken from the winter statistical fields of Polar Science Center Hydrographic Climatology (PHC3, updated from Steele et al., 2001) that ingests observations from the period 1900-1994. Total alkalinity (Alk) and preindustrial dissolved inorganic carbon (DIC) were initialized

from version 2 of the Global Ocean Data Analysis Project (GLODAPv2) climatology centered to the year 2002 (Lauvset et al., 2016) based on data collected between 1972 and 2013. Dissolved inorganic nitrogen (DIN) and dissolved silicic acid (DSi) were started with values from the Levitus World Ocean Atlas climatology of 2013 (Garcia et al., 2014) occupied between 1955 and 2012. We used the Levitus World Ocean Atlas climatology of 2018 for dissolved oxygen (Garcia et al., 2019a) (See Table 2) based on data for the time span 1955-2017.

· Line 246: I found it interesting that you only showed modelled mean fields for 2012-2021, given the long spin-up period as well as the authors stating in other locations (e.g. the abstract) that they are analyzing the period of 1958-2021. Why was this specific period chosen?

> The reviewer is right that we did not justify our choice. We averaged the model results over the period of 2012–2021 to get rid of interannual variability in order to evaluate the mean state in a recent period. Additionally, we used the last decade because we wanted to keep the presented results comparable to the time span where most pCO2 observations were taken. We kept this time period to evaluate other variables for consistency throughout the manuscript.

2. I found section 3.3.3 DIC Inventory Changes  a bit hard to follow (lines 522ff). I think part of it is just some awkward wording and potentially a need for it to be a bit more fleshed out (without adding too much more text!). Additionally, the authors state that "FESOM-2.1-REcoM3 is thus one of the few ocean biogeochemistry models that falls within the range of interior ocean anthropogenic carbon accumulation", which is again emphasized in the Conclusions. I think this should be discussed a bit more here. For example, if the numbers are readily available, I wonder if adding in estimates of C inventory from other models into Table 4 will help illustrate this point and facilitate a bit more discussion.

Thank you for pointing this out. We have fully reworked the section 3.3.3 with the aim to make it more digestible. We now shortly introduce the different carbon components (anthropogenic vs natural) and  have added references so that the interested reader can follow up all the details. We have cleaned Table 4 and also updated the numbers to be calculated for simulations A minus B, i.e. accounting for model drift. Further, we have introduced one other simulation (simulation D), with which we can provide an estimate of the change of the anthropogenic carbon inventory that is coherent with the definition of carbon flux components used in the observation-derived estimate. Simulation D is thus also introduced in the methods (section 2.2 and Table 1).

We have also added the numbers for the anthropogenic DIC inventory change 1994-2007 for the other models to the text. We decided not to add them to the table, because it's not our data and it wouldn't be appropriate to give these numbers for the first time in our model description paper. We will make sure that these numbers will be published in the appendix of the GCB 2023 paper to be accessible to the scientific community.

In regards to Table 4, perhaps it's personal preference, but I think the units should be in the column header as well as in the table caption? I also think it's a bit confusing to report the total inventory in 1994 vs the increase in C inventory for the other periods under the same columns/without any delineation.

We restructured the table to distinguish the total inventory from the differences and introduced subheadings in the table. We also added the units to the column headers.

3. There are a few sentences that require more references/details about references. See specifics below:

· Line 270: What other works? Maybe also include specific numbers of some sort from other studies?

References are added. The text now reads:

"Even though the runs depict large differences in temperature and salinity from the observed climatology, the simulated AMOC shows the canonical picture as known from other stand-alone ocean and coupled climate models (Griffies et al., 2009; Jungclaus et al., 2013; Danabasoglu et al., 2014)"

· Line 277: Please reference what "known other works" you are referring to

References are added and the text is revised as:

"In the Northern Hemisphere, the deepest MLD (>1000 m) is found in the Labrador Sea (LS) as well as in Greenland–Iceland– Norwegian (GIN) seas. The magnitude is larger than in Sallee et al. (2021) but is in the same range as other modelling studies (Griffies et al., 2009; Sidorenko et al., 2011). "

· Figure 19: You reference a "Globally integrated annual air-sea $CO_2$ flux from Global Ocean Biogeochemistry Models" but don't state what these models are. My guess is that the GOBMs used are from the Global Carbon Budget 2022 (?) but as written it's not clear and I think you're missing the appropriate reference to 2022 Carbon Budget here. I also think that it could be useful to explicitly state in the figure caption the number of GOBMs used (10 I believe?) and where in the 2022 Carbon Budget the readers can find information about these models (i.e. reference what table they're listed in). Potentially more details about the pCO2-based data products could also be briefly given.

Thank you for pointing out that this information was missing. The number of GOBMs (10) and pCO2-products (7) is now mentioned in the caption together with the reference to Table 4 of GCB 2022 and all references for the individual models and pCO2-products. We have also amended the discussion on the differences between GOBMs and pCO2-products in the text around Figure 19, referring to model biases in ventilation and carbonate chemistry as well as the uncertainties in the pCO2-products.

**Other minor comments/suggestions**
· Line 545: Is this a spatially averaged oxygen concentration or for a specific region? What were the values presented in Cocco et al. 2013 that you're comparing to?

Good point. The way we wrote this paragraph was indeed misleading. We calculated the global average and standard deviation of oxygen concentrations within the layer 100-600m from both simulations and observations. However, we used a more recent Atlas (WOA 2018 instead of 2009 in Cocco et al. 2013). Therefore, we calculated ourselves the values indicated in the text. The comparison with the Cocco et al. 2013, a valuable model intercomparison of simulated oxygen comparison, was mainly to show that FESOM2.1-REcoM3 showed good results, but this is  since all other models also showed values within the error range of observations. We updated the text as followed:

"FESOM2.1-REcoM3 performed remarkably well with simulated values of about 160±105 mmol m−3, which is very close to the observations from the WOA2018 (158±103 mmol m3) within the 100-600 m layer. A previous model intercomparison study of oxygen concentration concentrations within the 100-600 m layer (Cocco et al., 2013) showed that such performances are, however, common as all evaluated models fell within the error range of observations."

- Line 23: I'm a bit confused here... since a rate of 2.9 PgC per year was given but then the authors state it's in 2021. It's particularly confusing since the previous sentences refer to a 250 year trend in increasing atmospheric $CO_2$

  We have added a sentence on the longer term uptake to link better to the previous sentence, and have then given the numbers for the recent decade 2012-2021. The unit PgC/yr is then correct for the average flux in this period. The text reads now:

  "The ocean has taken up a remarkably constant fraction of 25-30% of human CO2 emissions from fossil fuel burning and land-use change throughout time (Crisp et al., 2022). For the recent decade 2012-2021, the rate of ocean anthropogenic carbon uptake (including effects of climate change) amounted to 2.9 ± 0.4 PgC/yr (26% of total CO2 emissions) (Friedlingstein et al., 2022b)."

  - Line 23-24 (about terrestrial/land fluxes) is also a bit confusing

    We find it important to put the ocean carbon sink in perspective not only to the fossil fuel emissions, but also to the second large carbon sink: the terrestrial biosphere. We have reformulated to link the sentences better as:

    "A similar proportion was taken up by the terrestrial biosphere 3.1 ± 0.6 PgC/yr (2012-2021), but the total air-to-land CO2 flux is substantially lower because of emissions from land-use change, mainly deforestation that amounted to 1.2 ± 0.7 PgC/yr (2012-2021)."

- Line 55ff: I think I would suggest moving this paragraph up to right before introducing REcoM and move the introduction of REcoM down to where you introduce FESOM. That way, you've then fully described the problem you're trying to solve and followed by an explanation of how you're solving it.

  Moved as suggested.

- Line 260: I'm curious if you have any theories as to why FESOM is too saline at the surface compared to climatology that could be commented on in the text?

  Indeed, we speculate that the bias is caused by the imperfections in the river discharge used in the simulations. We prescribed the river runoff as the annual mean runoff from Large and Yeager (2004), which was a standard practice in CORE-type simulations. However, when combined with the relatively low surface salinity restoring, using a piston velocity of 50m/300 days, the SSS bias drifts to the values depicted in Figure 3f. This now added in line :

  " The reason for this bias could be the imperfections in the river discharge from CORE forcing and the relatively low surface salinity restoring, using a piston velocity of 50m/300 days in the simulations."

- Line 305: Any idea why the moderately high silicic acid values in the northern high latitudes is not reproduced well?

  Thank you, this is a good question. We cannot say with certainty, but it may be linked to mixing that is too sluggish, or to overly strong silicic acid draw-down by diatoms, which in turn could be linked to parameter choices or iron limitation that may be too weak. We have added:

  "This may be related to mixing that is too sluggish, or to overly strong silicic acid draw-down by diatoms (Figs. 9 and A1), possibly linked to iron limitation that may be too weak (Fig. 12). ".

- Line 425: I'm not sure what is meant by "A more detailed description of zooplankton can increase NPP by 25%" or even how it fits in with the surrounding sentences

  We have amended the text and added reference to section 3.2.2 where this is explained in some more detail. The text now reads:

  "A more detailed description of zooplankton results in more efficient nutrient recycling and can thus increase NPP by 25% (see also explanation in section 3.2.2, Karakus et al., 2022)."

- Figure 13: I'm confused about what the difference between the orange dots and brown line are for MAREDAT? This should be more clearly defined in the figure caption.

  We have added the information to the figure caption:

  "orange dots for individual observations and solid brown line for the zonal mean of the observations"

- Figure 16: Panel A-B – could be nice to denote on the colourbar (or in the figure caption) what values (positive or negative) are sources/sinks of $CO_2$ for those unfamiliar

  We have added a sentence in the figure caption to clarify that negative indicates a flux into the ocean.

- Line 480: Perhaps outside the scope of this study but comparison to more regional studies could be interesting. For example, the North Atlantic shelves are shown to act as sinks of $CO_2$ (which is consistent with other global studies) but some regional studies have shown that the Scotian Shelf, for example, acts as a source of $CO_2$ (see Shadwick & Thomas 2014: https://doi.org/10.1016/j.marchem.2014.01.009; Rutherford et al. 2021: https://doi.org/10.5194/bg-18-6271-2021)

  Thank you for the comment. Indeed, while certainly interesting, we consider this out of scope here. If discussing specifically the Scotian shelf, then we would need to discuss other regions at the same level of detail. However, we have added a sentence on the patterns in the global coastal ocean in comparison to a recent synthesis based on low- and high-resolution models and pCO2-products (Resplandy et al.).

> "FESOM2.1-REcoM3 also generally captures the large-scale patterns of coastal $CO_2$ fluxes with $CO_2$ uptake in the mid- and high-latitudes (poleward of 25∘N/S) and outgassing in the tropical coastal ocean, as described in a recent synthesis based on low- and high-resolution models and pCO2-products (Resplandy et al., 2023)."

- Lines 502ff: I think this is a super interesting result! One would definitely assume that there is a similar bias in the control and historical simulations…

  Thank you! Yes, it is puzzling, isn't it?

- Line 514: What years of the model simulation are these flux estimates calculated over?

  We have simplified the sentence so that this information is more easily spotted. It now reads:

  > "After accounting for the bias in simulation B, the simulated ocean carbon sink (1990-1999) is 1.74 ± 0.11 PgC yr−1 and 2.17 ± 0.13 PgC yr−1 for FESOM1.4-REcoM2 and FESOM2.1-REcoM3, respectively."

- Line 516: is this an observationally based estimate?

  We refer the reviewer to the summary in Friedlingstein et al 2022: "This is based on indirect observations with seven different methodologies and their uncertainties and further use of the three of these methods that are deemed most reliable for the assessment of this quantity (Denman et al., 2007; Ciais et al., 2013). The observation-based estimates use the ocean–land $CO_2$ sink partitioning from observed atmospheric $CO_2$ and O2/N2 concentration trends (Manning and Keeling, 2006; Keeling and Manning, 2014), an oceanic inversion method constrained by ocean biogeochemistry data (Mikaloff Fletcher et al., 2006), and a method based on penetration timescale for chlorofluorocarbons (McNeil, 2003). The IPCC estimate of 2.2 GtC yr−1 for the 1990s is consistent with a range of methods (Wanninkhof et al., 2013)."

  We refrain from repeating this full information here, and just add:

  > "based on seven different methodologies".

**Minor suggested wording changes**

· Line 4: I think this should be slightly changed to: "Marine biogeochemical models are a useful tool but, as any model, are a simplification and need to be continually improved."

  Rephrased as suggested.

· Line 13; "Dissolved oxygen is **also** added as a new tracer."

  Amended as suggested.

· Line 14: remove comma after 1958-2021

  Amended as suggested.

· Line 21: sentence is a bit awkward

Revised as:

"Since the beginning of the industrial era (year 1750) the concentration of carbon dioxide ($CO_2$) in the air has substantially risen from 277 ppm to 417.2 ppm (year 2022, Friedlingstein et al., 2022b)."

· Line 55: I think remove the colon and replace it with a period

Amended as suggested.

· Line 60: Use either "such as" or "e.g." – you don't need both

We removed "e.g.," from the text.

· Line 75: add comma after "Here"

Amended as suggested.

· Line 85: Add a comma after "FESOM1.4"

Amended as suggested.

· Line 98: suggest adding a colon after "defined" so it reads "A pair of control volumes are defined: the vector control volumes are the prisms based on elements."

Amended as suggested.

· Line 141: maybe add some commas (and remove some of the "ands") to make things clearer?

Amended as suggested.

· Lines 324-325: both sentences are a bit awkward. Line 324, maybe say something like "The larger magnitude of dissolved iron in the model"? In line 325, maybe remove the word "probably"

Rephrased and amended as suggested.

· Line 326-327 is also a bit awkward

Rephrased it.

· Line 384: "The more severe than expected limitation in iron" is awkward

Thanks for spotting this. The strength of iron limitation for phytoplankton is unfortunately hard to compare to observations. We rephrased to:

"severe iron limitation"

· Line 395: awkward sentence

Thank you for spotting this. It now reads:

"Generally, the NPP and chlorophyll differences to satellite-based estimates could also be linked to model deficiencies, such as coarse model resolution and associated weak upwelling, missing complexity in simulated phytoplankton classes, but also the so far unconsidered nutrient input from terrigenous sources."

· Line 422: other **global** modelling studies?

   Amended as suggested.

· Line 424: global **ocean** and the Southern Ocean

   Amended as suggested.

**Reviewer 2**

The study presents an updated ocean-sea ice BGC model with updates to the sea ice component as well as to both the biogeochemical and physical ocean model. The model runs include spinup and pre-industrial and historical-present atmospheric forcing, respectively. The model output is compared to observations and reanalysis products, first for ocean physics (AMOC, SST, SSS) and then for ocean biogeochemistry. The authors do a good job of introducing the model and describing updates. The comparisons to other data products covers a wide range, but I would like to see a bit more on the side of quantitative rather than qualitative comparisons. With that and the corrections or responses to the comments below, I would be happy to recommend for publication.

We thank the reviewer for the overall positive assessment of our manuscript and answer the detailed points of criticism below, point by point.

**General Comments:**

Throughout the paper, there are comparisons between two datasets without any real quantification. They can be found at lines 254, 289, 449, 453. One exception is line 435. Please quantify the other comparisons similarly or in an appropriate way.

We agree with the reviewer that we did not provide statistical metrics for all modeled fields. This is now fixed and we systematically provide at least correlation coefficients and root mean squared errors for all variables (as in mentioned line 435 of the submitted manuscript).

There are also comparisons between model output for 2012-2021 with an observed/reanalysis product that doesn't have a time period: PHC (line 254), WOA DIN and DSi (line 286), GLODAP (line 435), OC-CCI (338, actually noted in Fig caption, but should be included in text),

Reviewer #1 asked the same question about the time span of climatologies and the reviewer is referred to the answer of reviewer #1's specific comment #1.

Figure 3, A map of just SST and SSS (in addition to the difference plots) would be helpful, either for Clim or run A.

We agree with the reviewer. The figure is updated.

262-263: I don't think runs A minus B should be compared to A minus climatology. The former can be considered de-drifted output or anthropogenic signal, while the latter is how well your model matches to observations (or reanalysis product).

We took out the A-B from the figure and rephrased the corresponding sentence in the text.

310-330: discussion of simulated iron. In this discussion, I don't see any reference to the Aeolian iron deposition (other than intro at line 221-222) as an external (atmospheric) forcing. I see that the modelled value is about twice that of the Huang product. Could you please discuss the forcing here briefly?

The only reference of dust input is given in L221-222, as the referee also mentioned. For all our experiments the same field of monthly dust input was used. Albani et al. (2014) provides an improved representation of dust using the Community Atmosphere Model and their products for present-day and the Last Glacial Maximum are widely used as aeolian source of iron in global biogeochemical models (e.g. Kurahashi-Nakamura et al. 2022, Du et al. 2022) or as reference to evaluate presentation of dust fluxes by the CMIP6 (Coupled Model Intercomparison Project) models (Zhao et al. 2022). Recent products of dust deposition from other atmosphere models are available as well (e.g. Myriokefalitakis et al., 2018) and Albani et al. (2014) with a total iron input of 16 Tg Fe more or less represents the average of the total range of those models between 10-30 Tg Fe. Therefore we do agree that there are uncertainties in applied dust input data and added the sentences below in the revised version. However, this might play a minor role compared with other reasons mentioned in the submitted version.
"Furthermore, the intensity and extension of dust plumes vary between modelled dust deposition fluxes (e.g., Myriokefalitakis etal., 2018). The field of dust deposition by Albani et al. (2014), used in our model to calculate aeolian iron input, is within the range of modern estimates but surely contains some uncertainties.

Figure 10, and text 349-365. Fig 9 seems sufficient for your comparison. The high coastal values in OC-CCI of Fig 9, especiallly in the coastal argument, already supports the discussion of the issue with ocean color products and turbid but abiotic signal in coastal waters. I suggest removing Fig 10. I would still keep most of the turbid water discussion, but link it to Fig 9. I also suggest removing the lines 357 (from "Therefore") to 362 up to "(Lee and Marr, 2022)."

We totally agree with the reviewer that the turbid coastal waters represent a challenge for the assessment of both remotely sensed chlorophyll and primary production. Therefore, following reviewer's recommendations, we decided to merge the results and discussions together to avoid repetition, shorten significantly the text and improve readability. However, we preferred to keep Figure 10 showing NPP fields as we believe this is an essential variable.

Fig. 11, Sometimes the PP aren't very limitted at all. Instead of just which is the most limitting factor, maybe use white for regions where all limitting factors are above 0.5 maybe? Also, should MLD be considered in concert with light limitation (i.e., a very deep MLD, despite high light at the surface, could limit PP because the plankton could sink deeper in the ML).

The reviewer is right that this analysis should be seen as a "likely limitation", most often at the beginning or the end of the productive season. But a limitation whatsoever must at some point intervene so that the PP remains finite as in the "real world", either by light or nutrients. In this sense, there is always a factor limiting the PP. In response to the reviewer's comment, we have added shading to the nutrient limitation figure to illustrate where limitation is weak (>0.5).

About the MLD, the classic Sverdrup Theory refers to the MLD as a light limiting factor. Indeed, if plankton is transported too long outside the euphotic layer, then, it will not be exposed sufficiently to sunlight to maintain growth over loss terms. In other words, the MLD effect can be seen as a light limiting effect.

Fig 14. Note that GLODAP is biased towards later in the sampling period when more measurements have been taken. For DIC in particular, that means GLODAP may have a higher value (DIC in surface ocean is increasing with time), but your model output is weighted equally over the time period. This would contribute to model-obs bias, but would not indicate a fault in the model. (This argument does not apply to Alk.)

We thank the reviewer for raising this important point. This also led us to realize that the time-period for DIC comparison was not chosen wisely. As DIC is changing over time, it needs to be compared to the same period as the GLODAP data (normalized to the year 2002). For DIC and alkalinity, we now chose a period centered around the year 2002 (1998-2006). We tested slightly shorter or longer periods, but this didn't affect the comparison substantially. The DIC bias is in fact smaller, and of more mixed (positive and negative) signs than in the previous version where we had compared FESOM-REcoM 2012-2021 to GLODAP 2002.

454: "largely agree": please quantify; "although the magnitude differs": I'm not sure what this means, please clarify

We reformulated the sentence to now read:

"Different pCO2-products largely agree with each other in terms of spatial patterns, although they may differ with respect to amplitude and timing of variability of regionally or globally integrated fluxes (Fay et al., 2021; Fay and McKinley, 2021)."

499-505: note that the magnitude of drift is reduced, but also the sign. Both have drift that leads annual air-sea CO2 flux to increase in magnitude (FESOM1.4 is positive with a positive drift, and FESOm2.1 is neg with a neg drift), therefore, I don't think it is self evident from that information that the flux will tend to zero with a longer spin-up.

The reviewer is right. We think that the flux will not tend to be zero before the deep ocean reaches a steady state. We removed " and could be further reduced towards zero with a longer spin-up" from the sentence.

**Specific/technical comments**

21: "preindustrial" should be "industrial", or "end of preindustrial"

Amended as suggested. "industrial" is used instead of preindustrial.

29: suggest changing "of the seasonal cycle" to "in accurately representing the seasonal cycle"

Amended as suggested.

40: reference needed for "large interbasin gradient between the Pacific and Atlantic"

Thanks - in fact the same reference already used in the sentence applies here. The reference was moved to the end of the sentence.

"The biological carbon pump is responsible for 75% of the natural vertical carbon gradient and for the large-interbasin gradient between the deep Pacific and Atlantic (Sarmiento and Gruber, 2006)."

110: add "off" after parameterization

Amended as suggested.

274: suggest changing "of both AMOC" to ", for both runs, of AMOC"

Amended as suggested.

275: suggest adding to "a nearly constant" with "nearly constant, but with a small increase," (I assume this is from continued model spinup drift?)

Amended as suggested.

280: replace "pursues" with "is in the same area as" - This similarity is not necessarily surprising to me. In general, MLD differences are largest where MLD is largest and where horizontal gradients are largest (i.e. Labrador Sea and Southern Ocean). I assume the same for SST and SSS (I recommended to include those with Fig 3)

Instead of looking at sim A minus sim B, we removed the sim B from our MLD analysis and focused solely on the simA. Therefore, we updated the MLD figure (Fig. 6). We compared March and September maximum mixed layer depth to the most up-to-date climatology from Salle et al. (2021). We revisited the entire MLD paragraph.

280: "Fig1" should be "Fig 3"

Amended as suggested.

320 "quite a bit smaller" quantify please (e.g., "smaller by about a quarter magnitude")

There is no uniform scaling factor for the amplitudes of the patterns between the Huang reconstruction and the model-generated field. The largest discrepancy is in the tropical Atlantic, where the model overestimates the increase in dFe under the Saharan dust plume by more than a factor of two. Thus, we added the following text including a quantitative statement:

"The largest discrepancy in amplitude is found under the Saharan dust plume in the tropical Atlantic, where the model produces maximal dissolved iron values that are almost three times as high as the reconstruction from \citet{Huang2022}. Direct observations in the tropical Atlantic also show dissolved iron concentrations that reach 1.2 nmol L-1 (e.g. Hatta et al., 2015), while modeled maxima are > 3 nmol L-1."

375 "annal" should be "annual"

Amended as suggested.

440: "(too low" please add "(which was too low in"

Amended as suggested.

Fig 16, please add that positive indicates into the ocean in the caption

Good point, a flux indeed into the ocean is negative, not positive. It now reads:

"Negative numbers indicate a flux into the ocean."

Fig. 17, use pCO2 instead of fCO2 to be consistent with rest of paper

Good point, done.

488: suggest replacing "misfits" with "biases" (two times)

Amended as suggested.

490 suggest changing "northern high latitudes" with "northern high latitudes negative bias"

 Amended as suggested.

491: suggest "Pacific" change to "Pacific positive bias"

Amended as suggested.

Fig. 19, I can't really tell the difference between the lines for pCO20 products and for Models. Use a more distinct pair of line-colors please

We use colors that have better contrast.

508 (from 512-513), please move "with a constant atm...forcing (simulation B)" up to line 508, just after "control simulations"

Good idea, done.

524: change "is with 27.7 PgC" to "is 27.7 PgC, "

Amended as suggested.

525: I suggest removing "best"

Amended as suggested.

533: remove one "also"

Amended as suggested.

535: suggest change "too weak" to "too weak of uptake or too strong release"

Amended as suggested. The sentence now reads:

"If the observation-based assessment of DIC inventory changes in North, Tropics and South is correct, this may indicate a transport of anthropogenic carbon from the Southern Ocean into the tropics that is too weak, or an air-sea CO2 flux in the tropics, with too little ocean uptake (or too much release) of CO2."

545: Replace "Compared ... compared" with "Compared to a model intercomparison study of", and also, what was the value of Cocco et al?

We agree with the reviewer. The first reviewer also raised that point. We therefore refer the reviewer #2 to our answer to the reviewer #1 above. It now reads:

"FESOM2.1-REcoM3 performed remarkably well with simulated values of about 160±105 mmol m−3, which is very close to the observations from the WOA2018 (158±103 mmol m3) within the 100-600 m layer. A previous model intercomparison study of oxygen concentration concentrations within the 100-600 m layer (Cocco et al., 2013) showed that such performances are, however, common as all evaluated models fell within the error range of observations."

553: Change "sensitivities of" to "sensitivities to"

Changed to:

"the sensitivity of phytoplankton growth to rising $CO_2$"